# A Stochastic Polynomial Expansion for Uncertainty Propagation through Networks

**Songhan Zhang**                                                          *songhan.z@wustl.edu*
*Department of Electrical and Systems Engineering*
*Washington University in St. Louis*

**ShiNung Ching**                                                          *shinung@wustl.edu*
*Department of Electrical and Systems Engineering*
*Washington University in St. Louis*

**Reviewed on OpenReview:** *https://openreview.net/forum?id=lyDRBhUjhv*

## Abstract

Network-based machine learning constructs are becoming more prevalent in sensing and decision-making systems. As these systems are implemented in safety-critical environments such as pedestrian detection and power management, it is crucial to evaluate confidence in their decisions. At the heart of this problem is a need to understand and characterize how errors at the input of networks become progressively expanded or contracted as signals move through layers, especially in light of the non-trivial nonlinearities manifest throughout modern machine learning architectures. When sampling methods become expensive due to network size or complexity, approximation is needed and popular methods include Jacobian (first order Taylor) linearization and stochastic linearization. However, despite computational tractability, the accuracy of these methods can break down in situations with moderate to high input uncertainty. Here, we present a generalized method of propagating variational multivariate Gaussian distributions through neural networks. We propose a modified Taylor expansion function for nonlinear transformation of Gaussian distributions, with an additional approximation in which the polynomial terms act on independent Gaussian random variables (which are identically distributed). With these approximated higher order terms (HOTs), we obtain significantly more accurate estimation of layer-wise distributions. Despite the introduction of the HOTs, this method can propagate a full covariance matrix with a complexity of $O(n^2)$ (and $O(n)$ if only propagating marginal variance), comparable to Jacobian linearization. Thus, our method finds a balance between efficiency and accuracy. We derived the closed form solutions for this approximate Stochastic Taylor expansion for seven commonly used nonlinearities and verified the effectiveness of our method in deep residual neural networks, Bayesian neural networks, and variational autoencoders. This general method can be integrated into use-cases such as Kalman filtering, adversarial training, and variational learning.

## 1 Introduction

A fundamental problem in uncertainty estimation and verification is to characterize how a given input distribution becomes transformed by the operant function (succinctly, $Y = f(X)$). When $f$ takes the form of a modern machine learning (ML) architecture, this problem quickly becomes analytically intractable, necessitating either sampling methods or approximation. The predominant approximation technique remains Jacobian linearization (JL), i.e., deterministic first order Taylor expansion around the mean of input distribution. To give a few examples in ML contexts, Gandhi et al. (2018); Dera et al. (2021); Petersen et al. (2024) used Jacobian linearization for uncertainty propagation through networks, and Beiu et al. (1994); Abdelaziz et al. (2015) used a piece-wise linear approximation of **sigmoid** functions prior to JL. Perhaps

unsurprisingly, these methods work best in low-uncertainty regimes, because taking derivate at input mean ignores the uncertainty of the derivative itself.

In another line of research, Bayesian neural networks try to do uncertainty awareness training by fitting Gaussians to their weights, so they will naturally output distributions rather than a deterministic results, avoiding overconfidence. In exact Bayesian inference, one needs to solve the posterior distribution of the network parameters $\theta$ given data $D$, i.e. $p(\theta|D) = \frac{p(D|\theta)p(\theta)}{\int p(D|\theta')p(\theta')d\theta'}$. The marginal distribution is an integral of the likelihood $p(D|\theta')$ over all possible combinations of network parameters. Since the likelihood is intractable due to the nonlinearity, this integral has to be approximated.

Early approaches attempted to precompute posterior mean and variance lookup tables to avoid Monte Carlo sampling at runtime (Hinton and van Camp, 1993), but this necessitated discretizing continuous variables, introducing imprecision. Monte Carlo-based methods remained popular for some time—Graves (2011) proposed Monte Carlo Variational Inference (MCVI) to approximate the evidence lower bound, though gradient estimation via sampling remained a limitation.

Between fully sampling and fully deterministic methods lies a class of numerical integration techniques that seek greater efficiency than naive Monte Carlo. Gauss-Hermite quadrature is a classical example: instead of drawing random samples, it deterministically chooses quadrature points and weights based on Hermite polynomials to efficiently approximate integrals of Gaussian-weighted functions. This approach underpins algorithms such as the cubature Kalman filter (Arasaratnam and Haykin, 2009; Särkkä, 2013), widely used for nonlinear state estimation in signal processing and control. In Section A.13, we analyze the relationship between our method and Gauss-Hermite quadrature, and demonstrate experimentally that our approach—which is fully deterministic and does not rely on quadrature sampling—achieves faster convergence.

At the fully deterministic end of the spectrum are moment-matching approaches, which propagate distributions by analytically computing the first and second moments of the output. The exact moments of Gaussian distributions transformed by **ReLU** were first derived in Frey and Hinton (1999) and later reused in several works, such as Gast and Roth (2018). Spiegelhalter and Lauritzen (1990) and MacKay (1992b) provided approximations for **sigmoid** and **tanh** functions using different reformulations, while Wang et al. (2016) presented another similar approach. Shekhovtsov and Flach (2019) developed an analytical method for propagating uncertainty through **argmax** and softmax, assuming logistic and Gumbel priors.

To integrate these ideas into BNN training, Jylänki et al. (2014) introduced expectation propagation. A more scalable alternative, probabilistic backpropagation (PBP), was later proposed by Hernández-Lobato and Adams (2015), which propagates moments forward through the network to estimate the marginal likelihood, then backpropagates gradients with respect to approximated posterior parameters. However, PBP only propagates the diagonal of covariance. To address this limitation, Wu et al. (2019) introduced a formulation that enables full covariance propagation through ReLU and Heaviside activations.

**Contributions**    To summarize, current uncertainty propagation of variational Gaussian distributions through nonlinear layers relies on one of the following (1) Monte Carlo sampling (2) local linearization of the nonlinearities or (3) direct derivation (or approximation) of the first two moments of output distributions under the assumption of zero correlation. The first is plausible thanks to powerful tensor calculation with GPUs, but is prone to undersampling, especially in high dimensional settings. The second allows propagating full covariance matrices, but introduces errors from ignoring higher order moments' contribution to the covariance calculation. The last one introduces errors from ignoring the correlations' contributions to the variance calculation.

To address these shortcomings, we here postulate a generalized framework of using a stochastic polynomial expansion as a surrogate nonlinearity, and derive the closed-form solutions of the mean, covariance, and cross-covariance of propagated multi-variate distributions, for seven nonlinearities that are ubiquitous in modern ML network constructs. This is achieved with a computational complexity of $\mathcal{O}(n^2)$, comparable to that of first-order Taylor expansion. Our methodology is inspired by stochastic linearization (SL), which uses expected value of the first derivative as gain and mean of output as bias, or $\hat{Y} = \mathbb{E}[\nabla_x f(X)] \circ (X - \mathbb{E}[X]) + \mathbb{E}[f(X)]$. Stochastic linearization minimizes mean square of the residual (Booton, 1953; Kazakov, 1954), and has been

used in the context of feedback control systems (see, e.g., (Zhang et al., 2025; Ching et al., 2010; Elishakoff and Crandall, 2017)).

## 2 Theory

To ensure consistency and clarity, we explicitly define the notation and convention used throughout this paper. All vectors are vertical. All instances of the $\circ$ operation denote element-wise (Hadamard) operations, including both element-wise multiplication and exponentiation. For example, given a vector $\mathbf{X}$, we define $\mathbf{X}^{\circ 2} = \mathbf{X} \circ \mathbf{X}$, whereas $\mathbf{X}^2 = \mathbf{X}\mathbf{X}^\top$, which represents the outer product. This convention extends to differntial operators as well. Specifically, the element-wise second derivative of a function $f$ is given by:

$$\nabla^{\circ 2} f = \left( \frac{\partial^2 f}{\partial x_1^2} \,, \, \cdots \,, \, \frac{\partial^2 f}{\partial x_n^2} \right)^\top = \mathrm{diag}\left( \nabla^2 f \right)$$

Let $\mathbf{X} := (\mathrm{X}_1, \mathrm{X}_2, \cdots, \mathrm{X}_n)^\top$ be a Gaussian random vector following $\mathcal{N}(\tilde{\boldsymbol{\mu}}, \tilde{\boldsymbol{\Sigma}})$, and $\boldsymbol{\Xi} := \mathbf{X} - \mathbb{E}[\mathbf{X}]$. Let $\bar{f}(\cdot)$ be a smooth, univariate function. We define the vector function $f(\mathbf{X}) = (\bar{f}(\mathrm{X}_1), \bar{f}(\mathrm{X}_2), \cdots, \bar{f}(\mathrm{X}_n))^\top$. Here, we have in mind that $f(\cdot)$ describes the activation function at the output of a feedforward layer. Now, let us define a set of i.i.d. surrogate random vectors $\boldsymbol{\Xi}_{(s)}$, which are mutually independent as random vectors, but the components within each vector may be correlated.

$$\boldsymbol{\Xi}_{(s)} \sim \text{i.i.d. } \mathcal{N}(\mathbf{0}, \tilde{\boldsymbol{\Sigma}}) \quad, \quad s = 1, 2, ..., S$$

We now propose the central construct in this paper, the **pseudo-Taylor polynomial expansion (PTPE)** of $f(X)$ as:

$$g(\mathbf{X}) = \mathbb{E}\left[ f(\mathbf{X}) \right] + \sum_{s=1}^{S} \frac{\mathbb{E}\left[ \nabla_{\mathbf{x}}^{\circ s} f(\mathbf{X}) \right]}{s!} \circ \left( \boldsymbol{\Xi}_{(s)}^{\circ s} - \mathbb{E}\left[ \boldsymbol{\Xi}_{(s)}^{\circ s} \right] \right) \tag{1}$$

We postulate and later show that this expansion provides a tractable and accurate approximation of $f(X)$ for the purposes of propagating uncertainty through feedforward network architectures.

In the above, we use the form of a Taylor polynomial expansion to describe the behavior (in expectation) of the function $f(\mathbf{X})$ subject to the stochastic input $\mathbf{X}$. The choice of the i.i.d. surrogate polynomials, $\{\mathbf{1}, \boldsymbol{\Xi}_{(1)}, \boldsymbol{\Xi}_{(2)}^{\circ 2}, \boldsymbol{\Xi}_{(3)}^{\circ 3}, \cdots\}$, is made to simplify the ensuing derivations. Note that if taking only the first two terms, this expansion is equivalent to stochastic linearization, because $\boldsymbol{\Xi}_{(1)} - \mathbb{E}[\boldsymbol{\Xi}_{(1)}]$ is equivalent to $\mathbf{X} - \mathbb{E}[\mathbf{X}]$. It is straightforward to observe that $g(\mathbf{X})$ has the same first moment as $f(\mathbf{X})$ because all terms after the first one are designed to have zero mean. In the following, we will provide empirical evidence that the second moment is well-captured for many common activation functions.

First, we derive the solution for covariance and cross-covariance using the proposed stochastic polynomial expansion.

**Lemma 1.** *Define*

$$\boldsymbol{A}_0 = \mathbb{E}\left[ f(\mathbf{X}) \right] \qquad \boldsymbol{A}_1 = \frac{\mathbb{E}\left[ \nabla_{\mathbf{x}} f(\mathbf{X}) \right]}{1!} \qquad \boldsymbol{A}_2 = \frac{\mathbb{E}\left[ \nabla_{\mathbf{x}}^{\circ 2} f(\mathbf{X}) \right]}{2!} \qquad \cdots$$

*Then, the covariance matrix of $g(\mathbf{X})$ is*

$$\Sigma_{g(\mathbf{X})} = \sum_{s=1}^{S} \boldsymbol{A}_s \circ \mathrm{cov}(\mathbf{X}^{\circ s}) \circ \boldsymbol{A}_s^\top \tag{2}$$

*for an $S$-th order expansion. For $S = 3$,*

$$\begin{aligned} \Sigma_{g(\mathbf{X})} = & \boldsymbol{A}_1 \circ \tilde{\boldsymbol{\Sigma}} \circ \boldsymbol{A}_1^\top + \\ & \boldsymbol{A}_2 \circ \left( 2\tilde{\boldsymbol{\Sigma}}^{\circ 2} \right) \circ \boldsymbol{A}_2^\top + \\ & \boldsymbol{A}_3 \circ \left[ 6\tilde{\boldsymbol{\Sigma}}^{\circ 3} + 9\, \mathrm{diag}(\tilde{\boldsymbol{\Sigma}}) \circ \tilde{\boldsymbol{\Sigma}} \circ \mathrm{diag}(\tilde{\boldsymbol{\Sigma}})^\top \right] \circ \boldsymbol{A}_3^\top \end{aligned}$$

*Proof.* The expected values can be solved using central moments of Gaussian distributions and Isserlis' theorem. All power operations are element-wise. Note that since $\boldsymbol{A}_s$ are $n$ dimensional vector, and all the power and product operations are element-wise, the complexity of calculating covariance is $\mathcal{O}(n^2)$. For detailed derivation, see Appendix A.1.2. □

It is useful to note that an addition (residual or recurrent) layer sums the activation of two (or more) layers, e.g. $\mathbf{X}$ and $g(\mathbf{Y})$. In thise case, the covariance of $\mathbf{X} + g(\mathbf{Y})$ is the sum of their covariances and cross-covariances, i.e.

$$\boldsymbol{\Sigma}_{\mathbf{X}+g(\mathbf{Y})} = \boldsymbol{\Sigma}_{\mathbf{X}} + \boldsymbol{\Sigma}_{g(\mathbf{Y})} + \boldsymbol{\Sigma}_{\mathbf{X}g(\mathbf{Y})} + \boldsymbol{\Sigma}_{g(\mathbf{Y})\mathbf{X}}$$

It is thus helpful to postulate an additional lemma for the purpose of calculating covariance after addition.

**Lemma 2.** *Let* $\mathbf{Y} \coloneqq (\mathrm{Y}_1, \mathrm{Y}_2, \cdots, \mathrm{Y}_n)^\top$ *be another Gaussian random vector that is cross-correlated to* $\mathbf{X}$ *with* $\boldsymbol{\Sigma}_{\mathbf{YX}}$. *Then, the cross-covariance matrix between* $\mathbf{Y}$ *and* $\mathbf{Z} \coloneqq g(\mathbf{X})$ *is*

$$\begin{aligned}
\boldsymbol{\Sigma}_{\mathbf{YZ}} &= \sum_{t=1,t\ is\ odd}^{S} \boldsymbol{A}_t^\top \circ \mathrm{cov}(\mathbf{Y}, \mathbf{X}^{\circ t}) \\
\boldsymbol{\Sigma}_{\mathbf{ZY}} &= \sum_{s=1,s\ is\ odd}^{S} \boldsymbol{A}_s \circ \mathrm{cov}(\mathbf{X}^{\circ t}, \mathbf{Y})
\end{aligned} \tag{3}$$

*for an S-th order expansion. For* $S = 3$,

$$\begin{aligned}
\boldsymbol{\Sigma}_{\mathbf{YZ}} &= \boldsymbol{A}_1^\top \circ \boldsymbol{\Sigma}_{\mathbf{YX}} + 3\boldsymbol{A}_3^\top \circ \boldsymbol{\Sigma}_{\mathbf{YX}} \circ \mathrm{diag}(\boldsymbol{\Sigma}_{\mathbf{X}})^\top \\
\boldsymbol{\Sigma}_{\mathbf{ZY}} &= \boldsymbol{A}_1 \circ \boldsymbol{\Sigma}_{\mathbf{XY}} + 3\boldsymbol{A}_3 \circ \boldsymbol{\Sigma}_{\mathbf{XY}} \circ \mathrm{diag}(\boldsymbol{\Sigma}_{\mathbf{X}})
\end{aligned}$$

*Proof.* The expected value can be calculated using Isserlis' theorem. Note that this term is nonzero only if $t$ and $s$ are odd. For details of derivation, see Appendix A.1.3. □

With these results, to find the covariance of the output of a nonlinear layer, assuming the input follows a multi-variate normal distribution, one needs to derive the coefficients of the PTPE, i.e., $\boldsymbol{A}_0$, $\boldsymbol{A}_1$, $\boldsymbol{A}_2$, etc., for the nonlinearity of interest. Note that these coefficients only depend on mean $\tilde{\boldsymbol{\mu}}$ and variance $\tilde{\boldsymbol{\sigma}}^2 = \mathrm{diag}(\tilde{\boldsymbol{\Sigma}})$, not correlations, rendering the computational complexity $\mathcal{O}(n)$. We briefly discuss some of the techniques we adopted to solve for these polynomial coefficients and list the final results in Table Table 1 and Table 4. For detailed derivation for all nonlinearities, see Appendix A.2 - A.7.

**Tanh, Sigmoid, and Softplus.** Because the integral $\int \nabla \mathbf{tanh}(x)p(x)dx$ is not tractable analytically, so we make a further approximation by substituting **tanh** with the error function which is very similar but more tractable. Specifically, we propose

$$\mathbf{tanh}(x) \approx \frac{1}{p} \sum_{j=1}^{p} \mathbf{erf}[\gamma_j x]$$

where $\{\gamma_1, \cdots, \gamma_p\}$ is a set of scaling factors obtained by numerical optimization (see Eq.7 in Appendix), and the relationship between approximation accuracy and the number of scaling factors is discussed in A.12. The error function is defined as

$$\mathbf{erf}(x) = \frac{2}{\sqrt{\pi}} \int_0^x \exp(-t^2)dt$$

Then, the integral $\int \nabla \mathbf{tanh}(x)p(x)dx$ can be approximated as $\frac{1}{p}\sum_{j=1}^{p} \int \nabla \mathbf{erf}(\gamma_j x)p(x)dx$, which is tractable analytically. The higher order derivatives of the error function are simply derivatives of Gaussian functions $\varphi(x)$, which are related to Hermite polynomials $\mathbf{H}_s(x)$ through

$$\frac{d^s}{dx^s}\left[\frac{1}{\sigma}\varphi\left(\frac{x}{\sigma}\right)\right] = \left(\frac{-1}{\sqrt{2\sigma^2}}\right)^s \mathbf{H}_s\left(\frac{x}{\sqrt{2\sigma^2}}\right)\frac{1}{\sigma}\varphi\left(\frac{x}{\sigma}\right) \tag{4}$$

where

$$\mathbf{H}_0(x) = 1 \qquad\qquad \mathbf{H}_1(x) = 2x \qquad\qquad \mathbf{H}_2(x) = 4x^2 - 2 \qquad\qquad \cdots$$

We show the pseudo-Taylor coefficients are convolutions of Gaussian derivatives and Gaussian pdf, which are analytically tractable (see Appendix A.2). We used a similar treatment for **sigmoid** function (see Appendix A.3). Using error functions as an approximation was first suggested by Spiegelhalter and Lauritzen (1990), but we use a linear combination, which is easily parallelizable and enhanced approximation accuracy. The derivation for **softplus** can reuse the results of **sigmoid**, because the derivative of **softplus** is just **sigmoid** with a scaling factor $\beta$ (A.4).

**ReLU and LeakyReLU.** It is obvious that we cannot apply our method directly on **ReLU**, because it is not continuously differentiable. Hence, we modified the results for **softplus** at the limit of $\beta \to \infty$, considering the relationship (A.5)

$$\lim_{\beta \to \infty} \frac{1}{\beta} \mathbf{log} \left( 1 + e^{\beta x} \right) = \max\{0, x\}$$

Similarly, **leaky ReLU** and any piece-wise linear activation function can be described as a combination of **ReLU** functions with different scaling, shifting, and/or mirroring.

**GELU and SiLU.** The derivatives of **GELU** can be expressed using derivatives of Gaussian cdf $\mathbf{\Phi}(x)$

$$\frac{\partial^s}{\partial x^s} \mathbf{GELU}(x) = s \frac{\partial^{s-1}}{\partial x^{s-1}} \mathbf{\Phi}(x) + x \frac{\partial^s}{\partial x^s} \mathbf{\Phi}(x)$$

The expected value of the **GELU** derivative involves integrating the product of Hermite polynomials and Gaussian functions, which is analytically tractable (A.6). Using normal cdf to approximate a **sigmoid** function, the derivations for a **SiLU** function becomes similar to that of the **GELU** (A.7).

Before presenting the final expression, we introduce a few element-wise operators to simplify the formulation. Let $\tilde{\boldsymbol{\sigma}}^2 = \mathrm{diag}(\tilde{\boldsymbol{\Sigma}})$ denote the input variance and $\hat{\sigma}_j^2$ represent a constant dependent on the nonlinearity (see Table 1 column 3). Defining $\hat{\boldsymbol{\sigma}}_j^2 = \tilde{\boldsymbol{\sigma}}^2 + \hat{\sigma}_j^2$, we then have

$$\mathcal{L}(\tilde{\mu} \,; \hat{\sigma}_j) \coloneqq \frac{1}{\hat{\sigma}_j} \, \varphi\left( \frac{x}{\hat{\sigma}_j} \right)\Bigg|_{x=\tilde{\mu}} \qquad\qquad \text{Likelihood of observing } x = \tilde{\mu} \text{ if } X \sim \mathcal{N}(0, \hat{\sigma}_j^2)$$

$$\mathcal{F}(\tilde{\mu} \,; \hat{\sigma}_j) \coloneqq \mathbf{\Phi}\left( \frac{x}{\hat{\sigma}_j} \right)\Bigg|_{x=\tilde{\mu}} \qquad\qquad \text{Cumulative likelihood of observing } x \le \tilde{\mu}$$

$$\mathcal{I}(\tilde{\mu} \,; \hat{\sigma}_j) \coloneqq \int_{-\infty}^{\tilde{\mu}} \mathbf{\Phi}\left( \frac{x}{\hat{\sigma}_j} \right) \mathrm{d}x = \tilde{\mu} \, \mathcal{F}(\tilde{\mu} \,; \hat{\sigma}_j) + \hat{\sigma}_j^2 \, \mathcal{L}(\tilde{\mu} \,; \hat{\sigma}_j) \qquad \text{Expected value of excess } (\tilde{\mu} - X)$$
$$\text{(only when } \tilde{\mu} > x, \text{ and } X \sim \mathcal{N}(0, \hat{\sigma}_j^2))$$

We also define an element-wise derivative operator $\boldsymbol{\mathcal{D}}^s$ such that

$$\boldsymbol{\mathcal{D}}^s \boldsymbol{\mathcal{L}}(\tilde{\boldsymbol{\mu}} \,; \hat{\boldsymbol{\sigma}}_j) \coloneqq \nabla_{\boldsymbol{x}}^{\circ s} \boldsymbol{\mathcal{L}}(\boldsymbol{x} \,; \hat{\boldsymbol{\sigma}}_j)\big|_{\boldsymbol{x}=\tilde{\boldsymbol{\mu}}} = \left( \frac{-1}{\sqrt{2\hat{\boldsymbol{\sigma}}_j^2}} \right)^s \mathbf{H}_s\left( \frac{\tilde{\boldsymbol{\mu}}}{\sqrt{2\hat{\boldsymbol{\sigma}}_j^2}} \right) \boldsymbol{\mathcal{L}}(\tilde{\boldsymbol{\mu}} \,; \hat{\boldsymbol{\sigma}}_j) \qquad\qquad \dagger$$

Notice that

$$\boldsymbol{\mathcal{D}} \, \boldsymbol{\mathcal{I}}(\tilde{\boldsymbol{\mu}} \,; \hat{\boldsymbol{\sigma}}_j) = \boldsymbol{\mathcal{F}}(\tilde{\boldsymbol{\mu}} \,; \hat{\boldsymbol{\sigma}}_j)$$
$$\boldsymbol{\mathcal{D}}^2 \, \boldsymbol{\mathcal{I}}(\tilde{\boldsymbol{\mu}} \,; \hat{\boldsymbol{\sigma}}_j) = \boldsymbol{\mathcal{D}} \boldsymbol{\mathcal{F}}(\tilde{\boldsymbol{\mu}} \,; \hat{\boldsymbol{\sigma}}_j) = \boldsymbol{\mathcal{L}}(\tilde{\boldsymbol{\mu}} \,; \hat{\boldsymbol{\sigma}}_j)$$

Then, it becomes evident that all of these pseudo-Taylor coefficients $\boldsymbol{A}_s$ can be written as derivatives of $\boldsymbol{\mathcal{I}}$, $\boldsymbol{\mathcal{F}}$, and $\boldsymbol{\mathcal{L}}$ (Table. 1). This is a result of the Gaussian assumption and the reformulation of **tanh** and **sigmoid**.

---

$\dagger$All operations in the right hand side of the equation are element-wise.

Table 1: General solutions for the pseudo-Taylor coefficients. For notational simplicity, all the product, division, and exponentiation operations are element-wise. For expanded solutions of the first four coefficients, see Table 4 in the Appendix.

| Nonlinearity | General Solution | $\hat{\boldsymbol{\sigma}}_j^2 = \tilde{\boldsymbol{\sigma}}^2 + \acute{\sigma}_j^2$ | Definition of $\gamma_j$ |
|---|---|---|---|
| Tanh | $\boldsymbol{A}_s = \dfrac{1}{s!}\dfrac{1}{p}\displaystyle\sum_{j=1}^{p}\boldsymbol{\mathcal{D}}^s\left[2\boldsymbol{\mathcal{F}}(\tilde{\boldsymbol{\mu}}\,;\hat{\boldsymbol{\sigma}}_j)-1\right]$ | $\acute{\sigma}_j^2 = \dfrac{1}{2\gamma_j^2}$ | $\tanh(x) \approx \dfrac{1}{p}\displaystyle\sum_{j=1}^{p}\operatorname{erf}(\gamma_j x)$ |
| Sigmoid | $\boldsymbol{A}_s = \dfrac{1}{s!}\dfrac{1}{p}\displaystyle\sum_{j=1}^{p}\boldsymbol{\mathcal{D}}^s\boldsymbol{\mathcal{F}}(\tilde{\boldsymbol{\mu}}\,;\hat{\boldsymbol{\sigma}}_j)$ | $\acute{\sigma}_j^2 = \dfrac{1}{2\gamma_j^2}$ | $\operatorname{sigmoid}(x) \approx \dfrac{1}{p}\displaystyle\sum_{j=1}^{p}\boldsymbol{\Phi}\left(\sqrt{2\gamma_j^2}\,x\right)$ |
| Softplus | $\boldsymbol{A}_s = \dfrac{1}{s!}\dfrac{1}{p}\displaystyle\sum_{j=1}^{p}\boldsymbol{\mathcal{D}}^s\boldsymbol{\mathcal{I}}(\tilde{\boldsymbol{\mu}}\,;\hat{\boldsymbol{\sigma}}_j)$ | $\acute{\sigma}_j^2 = \dfrac{1}{2\gamma_j^2\beta^2}$ | $\dfrac{\mathrm{d}}{\mathrm{d}x}\operatorname{softplus}(x;\beta) \approx \dfrac{1}{p}\displaystyle\sum_{j=1}^{p}\boldsymbol{\Phi}\left(\sqrt{2\gamma_j^2\beta^2}\,x\right)$ |
| ReLU | $\boldsymbol{A}_s = \dfrac{1}{s!}\,\boldsymbol{\mathcal{D}}^s\boldsymbol{\mathcal{I}}(\tilde{\boldsymbol{\mu}}\,;\tilde{\boldsymbol{\sigma}})$ | $\acute{\sigma}^2 = 0$ | |
| LeakyReLU($\theta$) | $\boldsymbol{A}_s = \dfrac{1}{s!}\,\boldsymbol{\mathcal{D}}^s\left[\theta\boldsymbol{x}+\boldsymbol{\mathcal{I}}(\tilde{\boldsymbol{\mu}}\,;\tilde{\boldsymbol{\sigma}})\right]$ | $\acute{\sigma}^2 = 0$ | |
| GELU | $\boldsymbol{A}_s = \dfrac{1}{s!}\,\boldsymbol{\mathcal{D}}^s\left[\boldsymbol{\mathcal{I}}(\tilde{\boldsymbol{\mu}}\,;\hat{\boldsymbol{\sigma}})-\boldsymbol{\mathcal{L}}(\tilde{\boldsymbol{\mu}}\,;\hat{\boldsymbol{\sigma}})\right]$ | $\acute{\sigma}^2 = 1$ | |
| SiLU | $\boldsymbol{A}_s = \dfrac{1}{s!}\dfrac{1}{p}\displaystyle\sum_{j=1}^{p}\boldsymbol{\mathcal{D}}^s\left[\boldsymbol{\mathcal{I}}(\tilde{\boldsymbol{\mu}}\,;\hat{\boldsymbol{\sigma}}_j)-\acute{\sigma}_j^2\boldsymbol{\mathcal{L}}(\tilde{\boldsymbol{\mu}}\,;\hat{\boldsymbol{\sigma}}_j)\right]$ | $\acute{\sigma}_j^2 = \dfrac{1}{2\gamma_j^2}$ | same as Sigmoid |

## 3 Results

### 3.1 PTPE significantly improves estimation accuracy when exposed to higher input variance

As an initial empirical test and demonstration of concept, we applied PTPE to a single, univariate nonlinearity subject to a parameterized normally distributed input. We varied the input mean and variance and examined how the output mean and variance compared to those predicted by PTPE. For this comparison, the true output statistics were obtained through $10^7$ Monte Carlo sampling across all input parameters. As expected, PTPE far outstrips Jacobian linearization, and this effect is prominent especially when input variance is high. With up-to third order PTPE, the estimated variance by our method is already very close to the ground truth (Fig. 1 col 4).

### 3.2 PTPE accurately quantifies uncertainty in canonical network architectures

To benchmark PTPE for uncertainty estimation in neural networks, we trained 9 residual neural networks (He et al. (2016)) with three depths (13, 33, and 65 layers) and 3 three typical nonlinearities (**Tanh**, **ReLU**, **GELU**) on CIFAR10 (Krizhevsky (2009)). We corrupted each input image with additive Gaussian noise to simulate noise in low light conditions (first type of corruption in Hendrycks and Dietterich (2019)), then compared the PTPE-predicted and reference (via $10^7$ Monte Carlo sampling) logits distributions. Four levels of corruption, with noise variance values of [1, 10, 100, 1000], were applied to RBG values ([0, 255]) of the input image. If z-scored, the corresponding noise variance scales are [1e-5, 1e-4. 1e-3, 1e-2]. The visualization of the corrupted images are shown in (Fig. 2). The layerwise application of PTPE is outlined in Algorithm 1 with accompanying pseudo code.

We measure the estimation accuracy of moments in three ways: the Euclidean distance from the reference mean to the predicted mean, $||\boldsymbol{\mu}_{\text{est}} - \boldsymbol{\mu}_{\text{ref}}||_2$, the Frobenius norm of the covariance residuals $||\boldsymbol{\Sigma}_{\text{est}} - \boldsymbol{\Sigma}_{\text{ref}}||_{\text{fro}}$, and the 2-Wasserstein distance (or Kantorovich-Rubinstein metric) between the reference and estimated distributions, assuming both distributions were Gaussian. This 2-Wasserstein distance is defined as

$$W_2 = \sqrt{||\boldsymbol{\mu}_{\text{est}} - \boldsymbol{\mu}_{\text{ref}}||_2^2 + \operatorname{trace}\left(\boldsymbol{\Sigma}_{\text{est}} + \boldsymbol{\Sigma}_{\text{ref}} - 2\left(\boldsymbol{\Sigma}_{\text{est}}^{1/2}\boldsymbol{\Sigma}_{\text{ref}}\boldsymbol{\Sigma}_{\text{est}}^{1/2}\right)^{1/2}\right)}$$

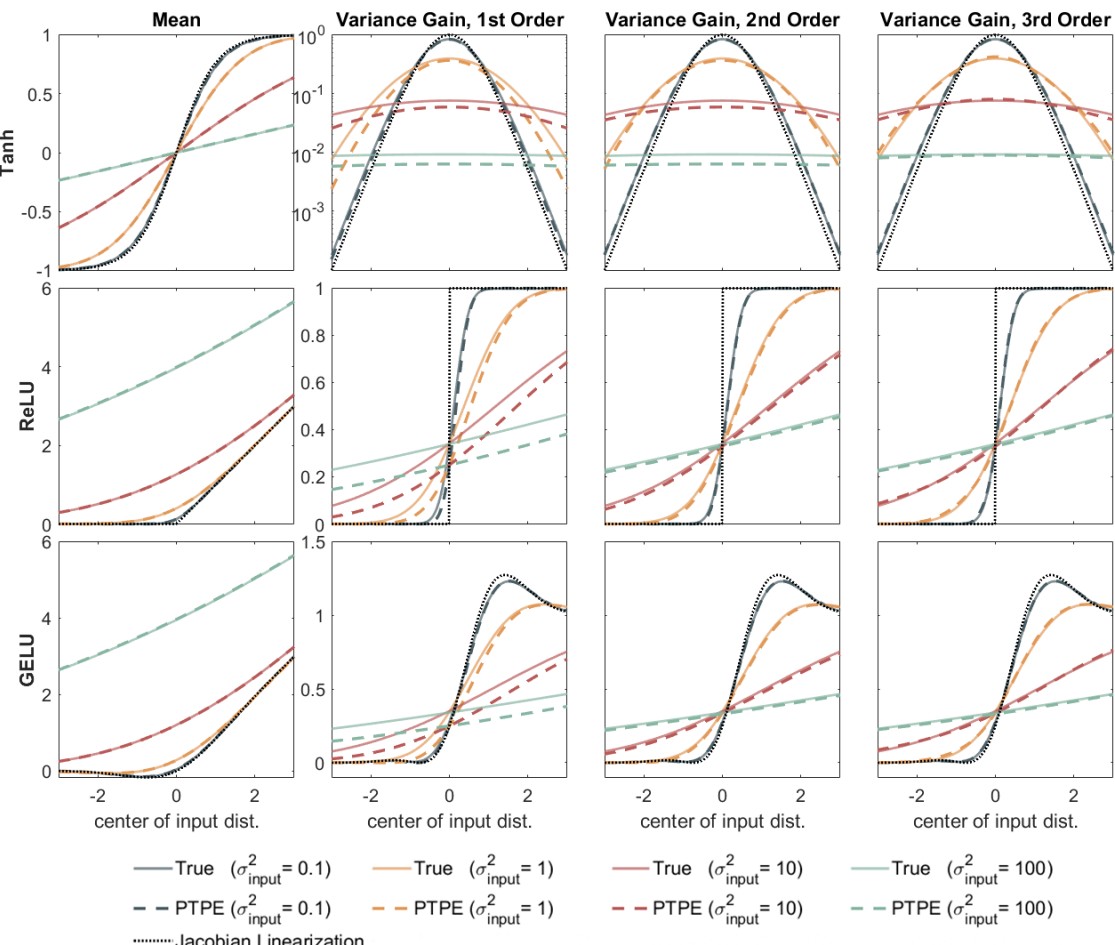

Figure 1: **Solid lines**: mean (column 1) and variance gain (output variance divided by input variance) (column 2-5) obtained by sampling 1e7 datapoints from Gaussian distributions with centers ranging from -3 to 3. **Dashed lines**: approximated mean and variance gain predicted with 1st, 2nd, and 3rd order pseudo Taylor polynomial expansion (column 2 - 4). Colors correspond to different input variances (blue: 0.1, yellow: 1, red: 10, green: 100). **Dotted lines**: approximated mean and variance gain using Jacobian linearization (first order deterministic Taylor expansion around input mean, e.g. Petersen et al. (2024)). For other nonlinearities, see Fig. 8.

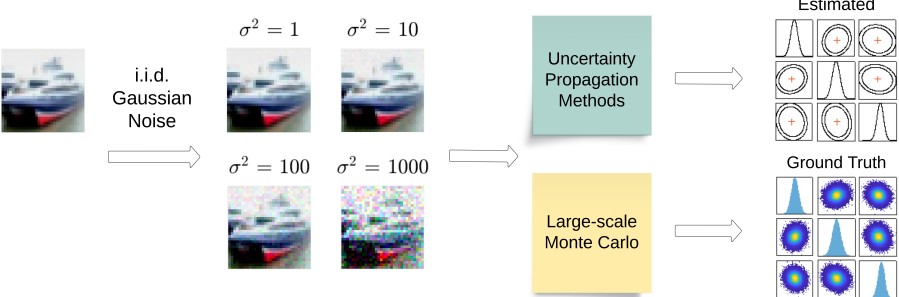

Figure 2: Schematic of experiment setup. We inject i.i.d. Gaussian noise to input image to simulate sensor noises, then compare the estimated output distribution to the ground truth obtained by large-scale simulation (sampling $10^7$ noisy images).

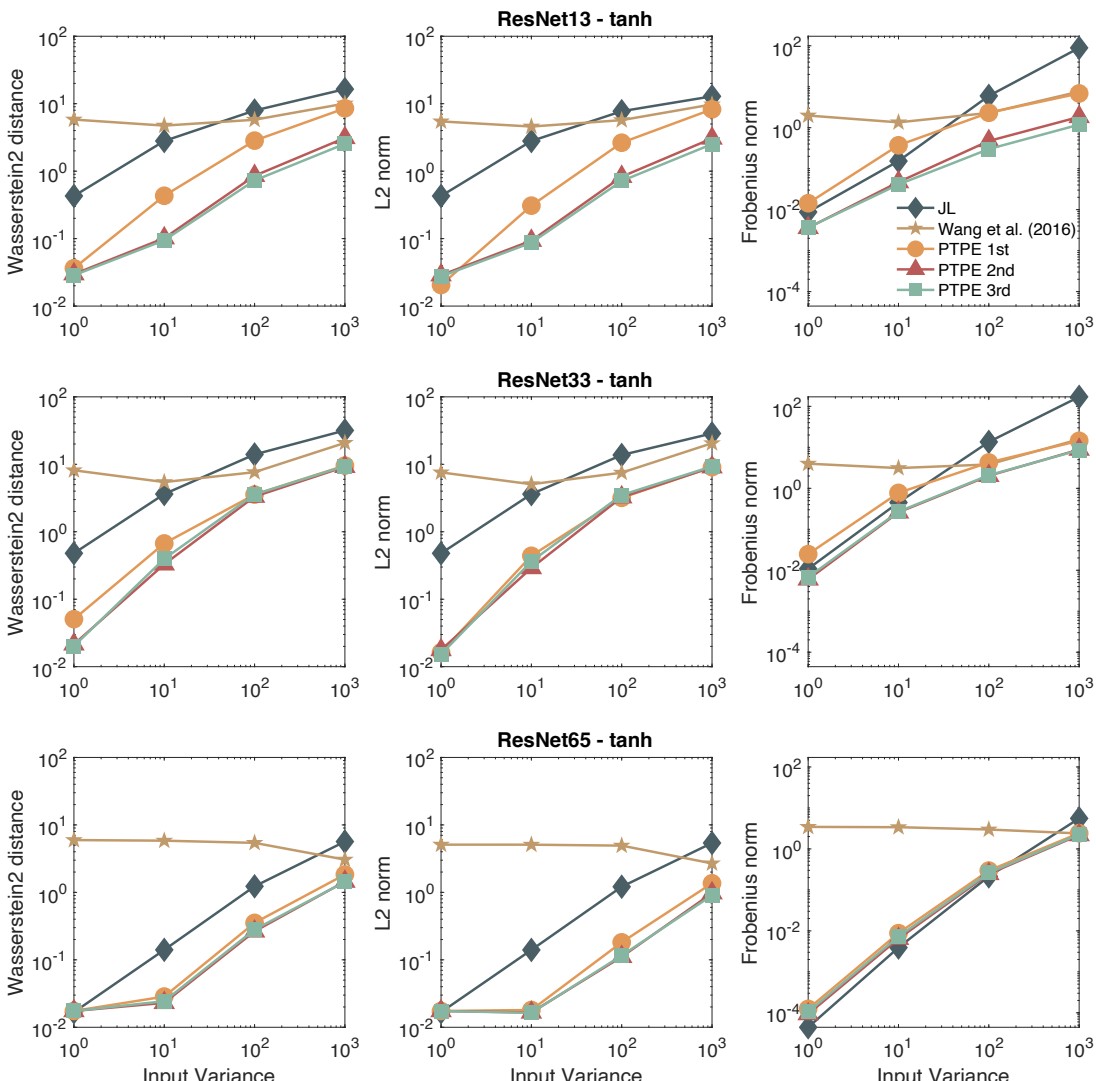

Figure 3: Estimation accuracies of expectation propagation methods evaluated on residual networks of three different depth, and four different input variance. The smaller the better. We chose Petersen et al. (2024) as an example of JL and Wang et al. (2016) as an alternative **Tanh** approximation. For non-linearities **ReLU** and **GELU**, see Fig. 9 and 10 in the Appendix. Differences between methods are systematic rather than random; we assessed variability by repeating experiments 10 times and computing error bars, but the resulting variances were too small to be visible at this scale. For transparency and reproducibility, all data and code used to generate these results are publicly available in our GitHub repository (Sec. 6).

We summarize the results in Fig. 3, 9, and 10. Overall, the experimental results align with expectations: (1) Jacobian linearization degrades dramatically in moderate to high variance regime. (2) Direct derivation is not suitable for this task due to the assumption of independence, since the overlapping convolution kernels and residual layers introduce substantial correlation. (3) Introducing up-to the third order PTPE typically outperformes stochastic and Jacobian linearization by a large margin. We also compared using 4 scaling factors versus using 8, and the results showed no significant difference (Fig. 12), justifying our choice of four scaling factors rather than a larger number in this case.

While higher-order approximations generally yield better performance, they can occasionally underperform in practice (e.g., for the L2 norm in ResNet13-tanh with input variance = 1), due to two factors:

- **Numerical Instability at Low Variance:** When input variance is small (for example, due to batch normalization), higher-order terms become increasingly sensitive to numerical instability. This instability can introduce errors that outweigh the potential benefits of higher-order corrections.
- **Imprecise Reference Distributions:** Our evaluation relies on reference distributions estimated using $10^7$ samples. While increasing the sample size from $10^6$ to $10^7$ led to lower Wasserstein-2, L2, and Frobenius distance metrics, it was not feasible to further increase the sample size to $10^8$ due to prohibitive runtime and memory requirements. We expect that such an increase would further reduce these metrics, but this could not be tested directly. Notably, estimating the statistics (e.g., mean and covariance) of a high-dimensional distribution (e.g., 10D or more) ideally requires an exponentially larger number of samples (on the order of $10^{70}$ for 10 dimensions), which is computationally infeasible. Thus, some of the observed underperformance of higher-order terms may be attributable to imperfections in the estimated reference.

While the second issue is inherent to the problem setup and cannot be fully eliminated, the first issue is, in principle, addressable. In practice, we recommend adaptively choosing the order of approximation based on input variance; for example, using second-order PTPE when variance is low to avoid instability, and higher orders when the input variance is larger.

### 3.3 PTPE addresses the limitations of DVI by incorporating non-piecewise-linear activations.

One major contribution of this work is to address the lack of accurate deterministic moment estimation for general nonlinearities in the field of variational inference. The method of deterministic variational inference (DVI) proposed by Wu et al. (2019) shares the same goal, which is to find a deterministic method to approximate moments in neural networks, thus eliminating gradient variance. However, closed form solutions of posterior mean and covariance were solved only for piecewise linear activations such as ReLU and Heaviside. We know $\text{Var}(f(X)) = \mathbb{E}[f(X)^2] - \mathbb{E}[f(X)]^2$, and the first term, $\mathbb{E}[f(X)^2] = \int f(x)^2 p_X(x)dx$, becomes arduous to solve for more complex nonlinearities $f(\cdot)$. Our approach circumvents this issue by taking derivatives inside the integrals, which provides tractability for general nonlinearities.

Table 2: Averaged test performance in RMSE. The smaller the better. The best value is highlighted in bold.

| Dataset | Concrete Strength | Energy Efficiency | Kin8nm | Naval Propulsion | Power Plant | Protein Structure | Wine Quality (Red) | Yacht Hydro-dynamics |
|---|---|---|---|---|---|---|---|---|
| Data points | 1030 | 768 | 8192 | 11934 | 9568 | 45730 | 1599 | 308 |
| Dimensions | 8 | 8 | 8 | 16 | 4 | 9 | 11 | 6 |
| MCVI | $7.128 \pm 0.123$ | $2.646 \pm 0.081$ | $0.099 \pm 0.001$ | $0.005 \pm 0.001$ | $4.327 \pm 0.035$ | $4.842 \pm 0.031$ | $0.646 \pm 0.008$ | $6.887 \pm 0.675$ |
| PBP | $5.667 \pm 0.093$ | $1.804 \pm 0.048$ | $0.098 \pm 0.001$ | $0.006 \pm 0.000$ | $4.124 \pm 0.035$ | $4.732 \pm 0.013$ | $0.635 \pm 0.008$ | $1.015 \pm 0.054$ |
| Dropout | $5.23 \pm 0.12$ | $1.66 \pm 0.04$ | $0.10 \pm 0.00$ | $0.01 \pm 0.00$ | $4.02 \pm 0.04$ | $\mathbf{4.36 \pm 0.01}$ | $\mathbf{0.62 \pm 0.01}$ | $1.11 \pm 0.09$ |
| Ensemble | $6.03 \pm 0.58$ | $2.09 \pm 0.29$ | $0.09 \pm 0.00$ | $\mathbf{0.00 \pm 0.00}$ | $4.11 \pm 0.17$ | $4.71 \pm 0.06$ | $0.64 \pm 0.04$ | $1.58 \pm 0.48$ |
| PTPE ReLU | $5.196 \pm 0.206$ | $0.615 \pm 0.024$ | $0.072 \pm 0.001$ | $\mathbf{0.003 \pm 0.000}$ | $3.925 \pm 0.025$ | $4.445 \pm 0.042$ | $0.633 \pm 0.010$ | $0.640 \pm 0.057$ |
| PTPE GELU | $\mathbf{5.068 \pm 0.153}$ | $\mathbf{0.570 \pm 0.021}$ | $\mathbf{0.071 \pm 0.000}$ | $0.004 \pm 0.000$ | $\mathbf{3.915 \pm 0.024}$ | $4.415 \pm 0.043$ | $0.634 \pm 0.010$ | $\mathbf{0.623 \pm 0.049}$ |
| PTPE Tanh | $5.574 \pm 0.148$ | $0.580 \pm 0.022$ | $0.076 \pm 0.001$ | $0.005 \pm 0.000$ | $4.073 \pm 0.028$ | $\mathbf{4.364 \pm 0.036}$ | $0.628 \pm 0.010$ | $1.678 \pm 0.193$ |

Table 3: Averaged test performance in average log-likelihood. The larger the better. The best value is highlighted in bold, and the second best is underlined.

| Dataset | Concrete Strength | Energy Efficiency | Kin8nm | Naval Propulsion | Power Plant | Protein Structure | Wine Quality (Red) | Yacht Hydro-dynamics |
|---|---|---|---|---|---|---|---|---|
| MCVI | $-3.391 \pm 0.017$ | $-2.391 \pm 0.029$ | $0.897 \pm 0.010$ | $3.734 \pm 0.116$ | $-2.890 \pm 0.010$ | $-2.992 \pm 0.006$ | $-0.980 \pm 0.013$ | $-3.439 \pm 0.163$ |
| PBP | $-3.161 \pm 0.019$ | $-2.042 \pm 0.019$ | $0.896 \pm 0.006$ | $3.731 \pm 0.006$ | $-2.837 \pm 0.009$ | $-2.973 \pm 0.003$ | $-0.968 \pm 0.014$ | $-1.634 \pm 0.016$ |
| Dropout | $\underline{-3.04 \pm 0.02}$ | $-1.99 \pm 0.02$ | $0.95 \pm 0.01$ | $3.80 \pm 0.01$ | $-2.80 \pm 0.01$ | $-2.89 \pm 0.01$ | $\underline{-0.93 \pm 0.01}$ | $-1.55 \pm 0.03$ |
| Ensemble | $-3.06 \pm 0.18$ | $-1.38 \pm 0.22$ | $1.20 \pm 0.02$ | $5.63 \pm 0.05$ | $\underline{-2.79 \pm 0.04}$ | $-2.83 \pm 0.02$ | $-0.94 \pm 0.12$ | $-1.18 \pm 0.21$ |
| DVI | $-3.06 \pm 0.01$ | $-1.01 \pm 0.06$ | $1.13 \pm 0.00$ | $\mathbf{6.29 \pm 0.04}$ | $-2.80 \pm 0.00$ | $-2.85 \pm 0.01$ | $\mathbf{-0.90 \pm 0.01}$ | $\underline{-0.47 \pm 0.03}$ |
| PTPE ReLU | $\mathbf{-3.010 \pm 0.037}$ | $-1.045 \pm 0.044$ | $1.251 \pm 0.009$ | $5.751 \pm 0.086$ | $\underline{-2.789 \pm 0.007}$ | $-2.821 \pm 0.024$ | $-0.966 \pm 0.029$ | $-0.910 \pm 0.044$ |
| PTPE GELU | $-3.092 \pm 0.056$ | $\underline{-0.789 \pm 0.039}$ | $\underline{1.278 \pm 0.007}$ | $5.858 \pm 0.135$ | $\mathbf{-2.780 \pm 0.006}$ | $\underline{-2.801 \pm 0.019}$ | $-0.982 \pm 0.029$ | $\mathbf{-0.236 \pm 0.052}$ |
| PTPE Tanh | $-3.159 \pm 0.039$ | $-0.827 \pm 0.043$ | $1.234 \pm 0.008$ | $6.050 \pm 0.028$ | $-2.825 \pm 0.006$ | $\underline{-2.802 \pm 0.013}$ | $-0.939 \pm 0.016$ | $-0.699 \pm 0.067$ |
| LL Tanh | $-3.07 \pm 0.07$ | $\mathbf{-0.65 \pm 0.05}$ | $\mathbf{1.29 \pm 0.01}$ | $\mathbf{6.29 \pm 0.19}$ | $\underline{-2.79 \pm 0.01}$ | $\mathbf{-2.79 \pm 0.00}$ | $-0.98 \pm 0.01$ | $-0.92 \pm 0.03$ |

We provide additional context and quantification by replacing the forward-passing functions of Gaussian moments in DVI with PTPE and conducting regression experiments on eight UCI datasets. Following the methodology suggested by Hernández-Lobato and Adams (2015), we search over MLPs with up to four layers containing 50 hidden units (100 for the larger Protein Structure dataset) and report the best test performance. We randomly set aside 10% of the data as test samples, and the error bars reflect the results from 20 random splits.

We evaluate RMSE and log-likelihood on the held-out data and summarize the results in Tables 2 and 3. For comparison, we include reported statistics (where available) from Monte Carlo Variational Inference (MCVI) (Graves, 2011), Probabilistic Backpropagation (PBP) (Hernández-Lobato and Adams, 2015), Dropout (Gal and Ghahramani, 2016), Ensemble (Lakshminarayanan et al., 2017), DVI (Wu et al., 2019), and Linearized Laplace (MacKay, 1992a; Foong et al., 2019). The results indicate that PTPE-DVI achieves competent accuracy, demonstrating the effectiveness of PTPE.

We extend the evaluation to out-of-distribution (OOD) detection in MNIST. Here, we test how models trained with PTPE respond to rotated and OOD images, using FashionMNIST (Xiao et al., 2017) as OOD data. The format of our analysis is similar to that of Ovadia et al. (2019).

The results (Fig. 4) show two main findings. First, the DVI+PTPE model achieves the highest accuracy on shifted images and is the least overconfident among all models tested, though this comes at the cost of a small drop in performance on undistorted images. One possible explanation is that the DVI+PTPE model may have learned a more robust representation of handwritten digits, which helps it generalize better to rotated images; however, we leave a more detailed investigation of this effect to future work. Second, all models perform similarly in terms of OOD detection capability.

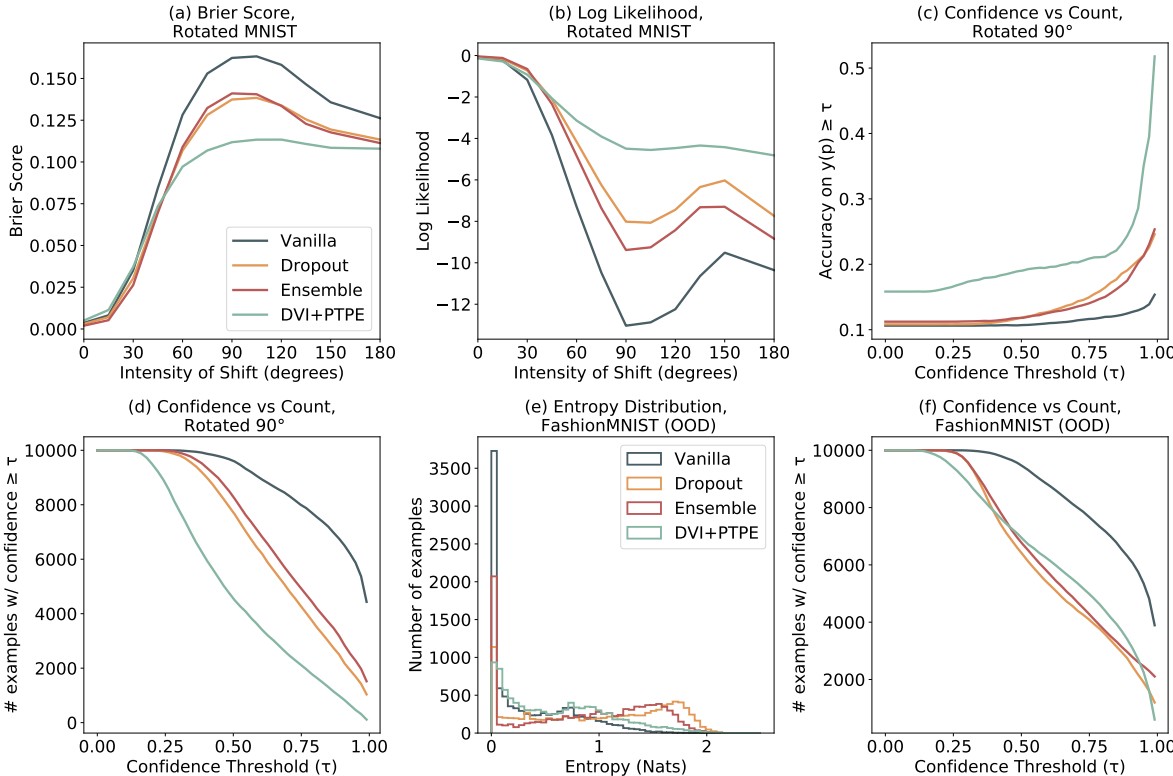

Figure 4: (a) and (b) show accuracy measured by Brier score and log likelihood as the test images are increasingly rotated. We explore the predictive distributions in 90 degree rotation of each method by looking at the confidence of the predictions in (c) and (d). We also explore the entropy and confidence of each method on entirely OOD data in (e) and (f).

### 3.4 PTPE enabled Variational Autoencoder demonstrates improved performance with $\mathcal{O}(n)$ complexity.

Since PTPE is fundamentally designed to accurately propagate Gaussian moments through nonlinearities, it can be incorporated into various applications, one of which is the decoder of a Variational Autoencoder (VAE) (Kingma and Welling, 2014). In the decoding stage, a VAE propagates the Gaussian means and variances encoded by the encoder through layers of nonlinearity in the decoder. The original model, which we refer to as the "vanilla VAE," accomplishes this by propagating Monte Carlo samples. Instead, we replace the decoder with PTPE while keeping the trainable parameters unchanged and train the model to reconstruct MNIST handwritten digits (LeCun et al., 1998). As shown in Fig. 5, the PTPE-enabled VAE achieves a higher Evidence Lower Bound (ELBO) and improved reconstruction accuracy compared to the vanilla VAE.

A key bottleneck in applying many well-established Bayesian methods to VAEs is scalability. Many Bayesian approaches require sampling, which suffers from curse of dimensionality. PTPE offers an alternative solution: the PTPE-VAE shown in Fig. 5 propagates only the diagonal of the covariance, resulting in a computational complexity of $\mathcal{O}(n)$. Moreover, the training procedure remains identical to that of the vanilla VAE.

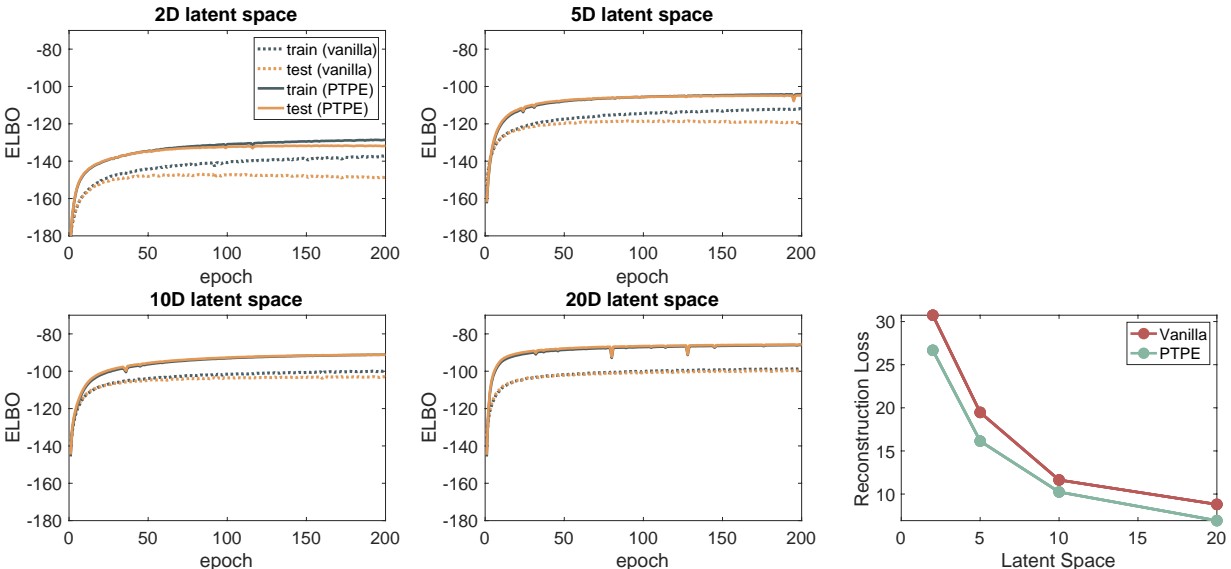

Figure 5: (Left four) Training and testing objective, measured as ELBO, for both vanilla and PTPE implemented VAE. (Right) Reconstruction loss (MSE) of the two types of models.

## 4 Discussion

One immediate potential limitation of PTPE is its reliance on the assumption that inputs are Gaussian. It has been well-established that at the limit of infinite width, a deep neural network with Gaussian input is equivalently a Gaussian process (Neal (1994); Williams (1996); de G. Matthews et al. (2018); Lee et al. (2018); Gao et al. (2023)), and a similar phenomenon is also reported in Bayesian neural networks with Gaussian weights (Goulet et al. (2021); Nguyen and Goulet (2022)). Based on this observation, we assume a "wide enough" neural network will have approximate Gaussianity in each layer, so that the error of using variational Gaussian distributions to approximate layer-wise distributions becomes negligible. We verify this assumption through simulation (see e.g., Fig. 6).

In this paper, we focus on the propagation of Gaussian distributions. This choice is due to their prevalence in machine learning and their convenient property of being Lévy alpha-stable, meaning a linear combination of Gaussian random variables remains Gaussian. This makes Gaussian distributions pertinent to our objectives. Consequently, our method could potentially be extended to other types within the Lévy alpha-stable family.

For instance, Petersen et al. (2024) demonstrated the propagation of Cauchy distributions through neural networks. A more comprehensive survey is provided in Wang et al. (2016), where the authors examined the propagation of exponential family distributions (including Beta, Rayleigh, Gamma, Poisson, and Gaussian), though this requires more intricate derivations.

A strength of PTPE is its generality. As mentioned in the introduction, several immediate motivating use-cases are in the training of robust networks including probabilistic network models. Furthermore, our proposed method may also find application in safety-critical engineering systems that require estimates on uncertainty. Recently, researchers combined an LSTM and Kalman filtering to monitor the states of plasma inside a nuclear fusion device Pavone et al. (2023). The Kalman filter, by construction, requires statistics on the output of the LSTM in order to generate control signals. Such statistics were generated by using a probabalistic architcture within the LSTM, i.e., where parameters are specified by a learned distribution. PTPE provides a potential alternative path for such problems (we discuss in A.13), but enabling uncertainty propagation through deterministic learned architectures.

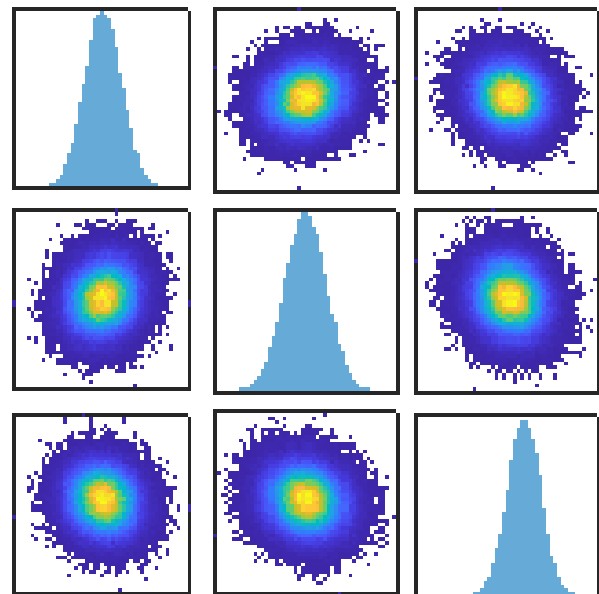

Figure 6: Empirical distributions of the first three logits of a resnet, obtained by Monte Carlo sampling $10^5$ images with additive Gaussian noises. For distributions of all 10 logits, see Fig. 7 in Appendix.

## 5 Conclusion

In this article, we developed a stochastic polynomial expansion approach, PTPE, to perform uncertainty propagation in neural networks. Our method offers significant advantages in accurately propagating the full covariance matrix of an input distribution compared to state-of-the-art methods, without substantially sacrificing computational efficiency. We derived analytical solutions for the first two moments of the output distributions for seven commonly used nonlinearities, demonstrating remarkable accuracy in predicting univariate mean and variance, particularly under high uncertainty. Additionally, we assessed its multivariate accuracy in deep residual neural networks trained on image categorization tasks. By incorporating up to third-order polynomial expansion, our method generally outperformed others, except in scenarios with minimal uncertainty in which the performance of competing methods is comparable. Overall, our proposed method provides a tractable framework for solving uncertainty propagation problems. It can potentially be effectively applied in various domains, including adversarial training, Bayesian inference, generative models, and safety-critical applications, offering a versatile tool for enhancing the reliability and robustness of neural networks.

## 6 Code Availability

All code for reproducing the experiments and figures is publicly available at `https://github.com/songhanz/Stochastic_Polynomial_Expansion`.

## 7 Acknowledgement

This research was, in part, funded by the U.S. Government (award no. HR00112290113). The views and conclusions contained in this document are those of the authors and should not be interpreted as representing the official policies, either expressed or implied, of the U.S. Government.

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

# A  Appendix

## A.1  Derivation of mean, covariance, and cross-covariance propagated through a univariate nonlinear function

Revisit the definitions of notations.

$$
\begin{aligned}
\mathbf{U} \quad & \text{multivariate Gaussian input} \quad \sim \mathcal{N}^n(\boldsymbol{\mu}, \boldsymbol{\Sigma}) \\
\boldsymbol{W} \quad & \text{weight matrix (constant)} \\
\boldsymbol{b} \quad & \text{bias vector (constant)} \\
\mathbf{X} \quad & = \boldsymbol{W}^\top \mathbf{U} + \boldsymbol{b} \quad \sim \mathcal{N}^n(\tilde{\boldsymbol{\mu}}, \tilde{\boldsymbol{\Sigma}}) \\
\tilde{\boldsymbol{\mu}} \quad & = \boldsymbol{W}^\top \boldsymbol{\mu} + \boldsymbol{b} \\
\tilde{\boldsymbol{\Sigma}} \quad & = \boldsymbol{W}^\top \boldsymbol{\Sigma} \boldsymbol{W} \\
\boldsymbol{\Xi} \quad & = \mathbf{X} - \tilde{\boldsymbol{\mu}} \quad \sim \mathcal{N}^n(\mathbf{0}, \tilde{\boldsymbol{\Sigma}}) \\
\tilde{\boldsymbol{\sigma}}^2 \quad & = \operatorname{diag}\left(\tilde{\boldsymbol{\Sigma}}\right) \\
\{\gamma_1, \dots, \gamma_p\} \quad & \text{positive scaling factor set obtained through numerical optimizations}
\end{aligned}
$$

All $\circ$ oprations are element-wise, including the Hadamard product and Hadamard exponentiation. However, for notationaly simplicity, all product, division, and power operations are element-wise starting from A.2.

In the context of machine learning, all non-linearities are applied element-wise – they are univarite. Thus, the off-diagonal entries of their Hessian matrices (second order partial derivatives) are zero, and it has similar effect on higher order partial derivatives. This makes PTPE on multivariate input easier to write down.

Given a smooth nonlinear function $\bar{f}(\cdot)$ of univariate random variables, define vector operation $f(\mathbf{X}) := (\bar{f}(X_1), \bar{f}(X_2), \cdots, \bar{f}(X_n))^\top$. We define an approximation $g(\cdot)$, which stochastically expands $f(\cdot)$ under the Taylor scheme, such that $f(\cdot)$ and $g(\cdot)$ have approximately the same first and second moment. This expansion uses i.i.d. surrogate polynomials, $\{\mathbf{1}, \boldsymbol{\Xi}_{(1)}, \boldsymbol{\Xi}_{(2)}^{\circ 2}, \boldsymbol{\Xi}_{(3)}^{\circ 3}, \cdots\}$, and such choice reduces computational complexity of covariance. Notably, the i.i.d. assumption is not a concession, but rather yields more accurate estimates than the non-i.i.d. alternative within our framework.

$$
g\left(\mathbf{X}\right) = \mathbb{E}\left[f\left(\mathbf{X}\right)\right] + \frac{\mathbb{E}\left[\nabla_{\mathbf{x}} f\left(\mathbf{X}\right)\right]}{1!} \circ \left(\boldsymbol{\Xi}_{(1)} - \mathbb{E}\left[\boldsymbol{\Xi}_{(1)}\right]\right) + \frac{\mathbb{E}\left[\nabla_{\mathbf{x}}^{\circ 2} f\left(\mathbf{X}\right)\right]}{2!} \circ \left(\boldsymbol{\Xi}_{(2)}^{\circ 2} - \mathbb{E}\left[\boldsymbol{\Xi}_{(2)}^{\circ 2}\right]\right) + \cdots
$$

and denote

$$
\begin{aligned}
\boldsymbol{A}_0 &= \mathbb{E}\left[f\left(\mathbf{X}\right)\right] \\
\boldsymbol{A}_1 &= \frac{\mathbb{E}\left[\nabla_{\mathbf{x}} f\left(\mathbf{X}\right)\right]}{1!} \\
\boldsymbol{A}_2 &= \frac{\mathbb{E}\left[\nabla_{\mathbf{x}}^{\circ 2} f\left(\mathbf{X}\right)\right]}{2!} \\
\boldsymbol{A}_3 &= \frac{\mathbb{E}\left[\nabla_{\mathbf{x}}^{\circ 3} f\left(\mathbf{X}\right)\right]}{3!} \\
&\cdots
\end{aligned}
$$

such that

$$
g\left(\mathbf{X}\right) = \mathbb{E}\left[f\left(\mathbf{X}\right)\right] + \underbrace{\sum_{s=1}^{\infty} \boldsymbol{A}_s \circ \left(\boldsymbol{\Xi}_{(s)}^{\circ s} - \mathbb{E}\left[\boldsymbol{\Xi}_{(s)}^{\circ s}\right]\right)}_{\text{zero mean}}
$$

### A.1.1 Mean

$\boldsymbol{A}_0$ is simply the mean of the output. In the later sections we show this value can either be analytically solved or approximated using similarly behaving nonlinear functions.

### A.1.2 Covariance

For clarity of reading, we omit the subscript of $\xi$, but will revisit the independence of polynomial basis. The covariance function of $f(\mathbf{X})$ with S-th order expansion is

$$
\begin{aligned}
\mathrm{cov}\left(g(\mathbf{X})\right) &= \mathbb{E}\left[\left(\sum_{s=1}^{S}\boldsymbol{A}_s \circ \left(\boldsymbol{\Xi}_{(s)}^{\circ s} - \mathbb{E}[\boldsymbol{\Xi}_{(s)}^{\circ s}]\right)\right)\left(\sum_{t=1}^{S}\boldsymbol{A}_t \circ \left(\boldsymbol{\Xi}_{(t)}^{\circ t} - \mathbb{E}[\boldsymbol{\Xi}_{(t)}^{\circ t}]\right)\right)^{\top}\right] \\
&= \sum_{s=1}^{S}\sum_{t=1}^{S}\left(\boldsymbol{A}_s\boldsymbol{A}_t^{\top}\right)\circ\mathbb{E}\left[\left(\boldsymbol{\Xi}_{(s)}^{\circ s} - \mathbb{E}[\boldsymbol{\Xi}_{(s)}^{\circ s}]\right)\left(\boldsymbol{\Xi}_{(t)}^{\circ t} - \mathbb{E}[\boldsymbol{\Xi}_{(t)}^{\circ t}]\right)^{\top}\right] \\
&= \sum_{s=1}^{S}\sum_{t=1}^{S}\left(\boldsymbol{A}_s\boldsymbol{A}_t^{\top}\right)\circ\left(\mathbb{E}\left[\boldsymbol{\Xi}_{(s)}^{\circ s}(\boldsymbol{\Xi}_{(t)}^{\circ t})^{\top}\right] - \mathbb{E}[\boldsymbol{\Xi}_{(s)}^{\circ s}]\mathbb{E}[\boldsymbol{\Xi}_{(t)}^{\circ t}]^{\top}\right)
\end{aligned}
$$

which is an $n \times n$ matrix, and the $\{i, j\}$-th entry is

$$
\sum_{s=1}^{S}\sum_{t=1}^{S}A_{s,i}A_{t,j}\left(\mathbb{E}\left[\Xi_{(s),i}^{s}\Xi_{(t),j}^{t}\right] - \mathbb{E}\left[\Xi_{(s),i}^{s}\right]\mathbb{E}\left[\Xi_{(t),j}^{t}\right]\right)
$$

Since $\boldsymbol{\Xi}_{(1)}, \boldsymbol{\Xi}_{(2)}^{\circ 2}, \boldsymbol{\Xi}_{(3)}^{\circ 3}, \cdots$ are independent, the off-diagonal entries $(s \neq t)$ are zero, then the $\{i, j\}$-th entry becomes

$$
\sum_{s=1}^{S}A_{s,i}A_{s,j}\left(\mathbb{E}\left[\Xi_{(s),i}^{s}\Xi_{(s),j}^{s}\right] - \mathbb{E}\left[\Xi_{(s),i}^{s}\right]\mathbb{E}\left[\Xi_{(s),j}^{s}\right]\right)
$$

Rewrite in matrix form

$$
\begin{aligned}
\mathrm{cov}\left(g(\mathbf{X})\right) &= \sum_{s=1}^{S}\boldsymbol{A}_s \circ \left(\mathbb{E}\left[\boldsymbol{\Xi}_{(s)}^{\circ s}\,\boldsymbol{\Xi}_{(s)}^{\circ s\top}\right] - \mathbb{E}\left[\boldsymbol{\Xi}_{(s)}^{\circ s}\right]\mathbb{E}\left[\boldsymbol{\Xi}_{(s)}^{\circ s}\right]^{\top}\right)\circ\boldsymbol{A}_s^{\top} \\
&= \sum_{s=1}^{S}\boldsymbol{A}_s \circ \mathrm{cov}(\mathbf{X}^{\circ s})\circ\boldsymbol{A}_s^{\top}
\end{aligned}
\tag{5}
$$

where $\boldsymbol{A}_s$ and $\boldsymbol{\Xi}_{(s)}$ are both n dimensional vertical vectors. Using central moments of normal distributions,

$$
\mathbb{E}\left[\Xi_{(s),i}^{s}\right] = \begin{cases} 0 & \text{if } s \text{ is odd} \\ \tilde{\sigma}_i^s(s-1)!! & \text{if } s \text{ is even} \end{cases}
$$

With application of Isserlis' theorem (Isserlis, 1918),

$$
\mathbb{E}\left[\Xi_{(s),i}^{s}\Xi_{(s),j}^{s}\right] = \sum_{p \in P_{\mathcal{B}}^2}\prod_{\{c,d\}\in p}\tilde{\rho}_{cd}\tilde{\sigma}_c\tilde{\sigma}_d
$$

where $c, d \in \{i, j\}$, and $\tilde{\rho}_{cd}$ is correlation. The sum is over all the pairings of the set $\mathcal{B} = \{\underbrace{i, i, \cdots, i}_{s}, \underbrace{j, j, \cdots, j}_{s}\}$,

i.e. all distinct (suppose each i or j is different from other i's or j's) ways of partitioning $\mathcal{B}$ into pairs $\{c, d\}$,

and the product is over the pairs contained in $p$ (Janson, 1997; Michalowicz et al., 2011), so there exists $(2s-1)!!$ pairs in the partition, or $(2s-1)!!$ terms in the sum. For example, the first four terms of Eqn. 5 are

$$\boldsymbol{A}_1 \circ \tilde{\boldsymbol{\Sigma}} \circ \boldsymbol{A}_1^\top$$
$$\boldsymbol{A}_2 \circ \left(2\tilde{\boldsymbol{\Sigma}}^{\circ 2}\right) \circ \boldsymbol{A}_2^\top$$
$$\boldsymbol{A}_3 \circ \left[6\tilde{\boldsymbol{\Sigma}}^{\circ 3} + 9 \operatorname{diag}(\tilde{\boldsymbol{\Sigma}}) \circ \tilde{\boldsymbol{\Sigma}} \circ \operatorname{diag}(\tilde{\boldsymbol{\Sigma}})^\top\right] \circ \boldsymbol{A}_3^\top$$
$$\boldsymbol{A}_4 \circ \left[24\tilde{\boldsymbol{\Sigma}}^{\circ 4} + 72 \operatorname{diag}(\tilde{\boldsymbol{\Sigma}}) \circ \tilde{\boldsymbol{\Sigma}}^{\circ 2} \circ \operatorname{diag}(\tilde{\boldsymbol{\Sigma}})^\top\right] \circ \boldsymbol{A}_4^\top$$

$\boldsymbol{A}_s$ and $\operatorname{diag}(\tilde{\boldsymbol{\Sigma}})$ are both n dimensional vertical vector. With this result, to find the covariance of the output of a nonlinear layer, assuming the input follows a multi-variate normal distribution, one just needs to derive the factors of Taylor polynomial, $\boldsymbol{A}_0$, $\boldsymbol{A}_1$, $\boldsymbol{A}_2$, etc., for the nonlinearity.

### A.1.3 Cross-covariance

Let $\mathbf{Y} := (Y_1, Y_2, \cdots, Y_n)^\top$ be a Gaussian random vector that is cross-correlated to $\mathbf{X}$, and $\boldsymbol{\Omega} = \mathbf{Y} - \mathbb{E}[\mathbf{Y}]$. If $\mathbf{X}$ undergoes a non-linear transformation via function $f(\cdot)$,

$$\mathbf{Z} = f(\mathbf{X})$$

The cross-covariance between $\mathbf{Y}$ and $\mathbf{Z}$ can be written as

$$\boldsymbol{\Sigma}_{\mathbf{YZ}} = \mathbb{E}\left[\boldsymbol{\Omega}\left(\sum_{t=1}^{S} \boldsymbol{A}_t \circ \left(\boldsymbol{\Xi}_{(t)}^{\circ t} - \mathbb{E}\left[\boldsymbol{\Xi}_{(t)}^{\circ t}\right]\right)\right)^\top\right]$$

which is an $n \times n$ matrix, and the $\{i, j\}$-th entry is

$$\sum_{t=1}^{S} A_{t,j}\left(\mathbb{E}\left[\Omega_i \Xi_{(t),j}^t\right] - \mathbb{E}[\Omega_i]\mathbb{E}\left[\Xi_{(t),j}^t\right]\right)$$

$\mathbb{E}[\Omega_i]$ is zero by definition. The terms with product of odd number of Gaussian random variables are zero by Isserlis' theorem. It can be simplified as

$$\boldsymbol{\Sigma}_{\mathbf{YZ}}(i, j) = \sum_{t=1, t \text{ is odd}}^{S} A_{t,j}\mathbb{E}\left[\Omega_i \Xi_{(t),j}^t\right]$$

$$\boldsymbol{\Sigma}_{\mathbf{ZY}}(i, j) = \sum_{s=1, s \text{ is odd}}^{S} A_{s,i}\mathbb{E}\left[\Xi_{(s),i}^s \Omega_j\right]$$

Rewrite in matrix form

$$\boldsymbol{\Sigma}_{\mathbf{YZ}} = \sum_{t=1, t \text{ is odd}}^{S} \boldsymbol{A}_t^\top \circ \mathbb{E}\left[\boldsymbol{\Omega}\, \boldsymbol{\Xi}_{(t)}^{\circ t^\top}\right] = \sum_{t=1, t \text{ is odd}}^{S} \boldsymbol{A}_t^\top \circ \operatorname{cov}(\mathbf{Y}, \mathbf{X}^{\circ t})$$

$$\boldsymbol{\Sigma}_{\mathbf{ZY}} = \sum_{s=1, s \text{ is odd}}^{S} \boldsymbol{A}_s \circ \mathbb{E}\left[\boldsymbol{\Xi}_{(s)}^{\circ s}\, \boldsymbol{\Omega}^\top\right] = \sum_{s=1, s \text{ is odd}}^{S} \boldsymbol{A}_s \circ \operatorname{cov}(\mathbf{X}^{\circ t}, \mathbf{Y})$$

$$(6)$$

and the expected value can be calculated using Isserlis' theorem mentioned above. Note that this term is nonzero only if $t$ and $s$ are odd, so the first two terms are

$$\boldsymbol{\Sigma}_{\mathbf{YZ}} \approx \boldsymbol{A}_1^\top \circ \boldsymbol{\Sigma}_{\mathbf{YX}} + 3\boldsymbol{A}_3^\top \circ \boldsymbol{\Sigma}_{\mathbf{YX}} \circ \operatorname{diag}(\boldsymbol{\Sigma}_{\mathbf{X}})^\top$$
$$\boldsymbol{\Sigma}_{\mathbf{ZY}} \approx \boldsymbol{A}_1 \circ \boldsymbol{\Sigma}_{\mathbf{XY}} + 3\boldsymbol{A}_3 \circ \boldsymbol{\Sigma}_{\mathbf{XY}} \circ \operatorname{diag}(\boldsymbol{\Sigma}_{\mathbf{X}})$$

An addition (e.g. residual) layer outputs the summation of activation of two (or more) layers, $\mathbf{Y}$ and $\mathbf{Z}$. Thus the covariance of $\mathbf{Y} + \mathbf{Z}$ is the sum of their covariance and cross-covariance.

$$\boldsymbol{\Sigma}(\mathbf{Y} + \mathbf{Z}) = \boldsymbol{\Sigma}_{\mathbf{Y}} + \boldsymbol{\Sigma}_{\mathbf{YZ}} + \boldsymbol{\Sigma}_{\mathbf{ZY}} + \boldsymbol{\Sigma}_{\mathbf{Z}}$$

### A.2 Tanh layers †

We use a linear combination of **independent** error functions with different scaling factors to approximate **tanh** function. In our experiments, we choose a set of four scaling parameters, $\{0.5583, 0.8596, 0.8596, 1.2612\}$, using `fmincon` in MATLAB. In practice, one can add more terms for even higher accuracy without losing efficiency (depending on the computing resources), because the extra terms can be easily paralleled. We define a variance term considering the relation between error function and Gaussian cdf, such that

$$\mathbf{tanh}(\mathbf{X}) \approx \frac{1}{p} \sum_{j=1}^{p} \mathbf{erf}\left(\gamma_j \mathbf{X}\right)$$
$$\acute{\sigma}_j^2 = \frac{1}{2\gamma_j^2} \tag{7}$$

Thus, the factors of the pseudo Taylor polynomials are

$$\boldsymbol{A}_0 = \mathbb{E}\left[\frac{1}{p} \sum_{j=1}^{p} \mathbf{erf}\left(\frac{\mathbf{X}}{\sqrt{2\acute{\sigma}_j^2}}\right)\right]$$

$$\boldsymbol{A}_1 = \mathbb{E}\left[\nabla_{\mathbf{x}}\left(\frac{1}{p} \sum_{j=1}^{p} \mathbf{erf}\left(\frac{\mathbf{X}}{\sqrt{2\acute{\sigma}_j^2}}\right)\right)\right]$$

$$\boldsymbol{A}_2 = \frac{1}{2!}\mathbb{E}\left[\nabla_{\mathbf{x}}^{\circ 2}\left(\frac{1}{p} \sum_{j=1}^{p} \mathbf{erf}\left(\frac{\mathbf{X}}{\sqrt{2\acute{\sigma}_j^2}}\right)\right)\right]$$

$$\cdots$$

Since all the operations in $\boldsymbol{A}_0, \boldsymbol{A}_1, \boldsymbol{A}_2, \cdots$ are element-wise, we only show the derivation for univariate case for notational simplicity in the following sections

#### A.2.1 Find $A_0$

$$\mathbb{E}\left[\mathbf{erf}\left(\frac{\mathrm{X}}{\sqrt{2\acute{\sigma}_j^2}}\right)\right] = \int_{-\infty}^{\infty} \mathbf{erf}\left(\frac{x}{\sqrt{2\acute{\sigma}_j^2}}\right) \frac{1}{\tilde{\sigma}} \varphi\left(\frac{x-\tilde{\mu}}{\tilde{\sigma}}\right) \mathrm{d}x$$

This is a known integral Ng and Geller (1969)

$$= \mathbf{erf}\left(\frac{\tilde{\mu}}{\sqrt{2\tilde{\sigma}^2 + 2\acute{\sigma}_j^2}}\right)$$

We define

$$\hat{\sigma}_j^2 = \tilde{\sigma}^2 + \acute{\sigma}_j^2 \tag{8}$$

Thus,

$$\boxed{A_0 = \frac{1}{p} \sum_{j=1}^{p} \mathbf{erf}\left(\frac{\tilde{\mu}}{\sqrt{2\hat{\sigma}_j^2}}\right)} \tag{9}$$

---

† For notational simplicity, all the product, division, and power operations that appear in and after this section are all element-wise.

The usage of error function instead of Gaussian cdf may give $A_0$ a very distinctive form from those of the other factors. The reasons behind are purely out of considerations of numerical computing: calculating Gaussian cdf is computationally demanding, while the approximation algorithm of the error function is available Cody (1969).

### A.2.2  Find $A_1$

Notice that

$$\frac{\partial}{\partial \mathrm{x}}\mathbf{erf}\left(\frac{\mathrm{X}}{\sqrt{2\acute{\sigma}_j^2}}\right) = \frac{\partial}{\partial x}\left(\int_0^{x/\sqrt{2\acute{\sigma}_j^2}}\frac{2}{\sqrt{\pi}}\mathbf{exp}(-t^2)\mathrm{d}t\right)$$

by Leibniz integral rule

$$= \frac{2}{\sqrt{\pi}}\frac{1}{\sqrt{2\acute{\sigma}_j^2}}\mathbf{exp}\left(-\frac{x^2}{2\acute{\sigma}_j^2}\right)$$

$$= \frac{2}{\acute{\sigma}_j}\varphi\left(\frac{x}{\acute{\sigma}_j}\right)$$

where $\varphi$ is the standard normal pdf. With the identity that the convolution of two Gaussians is still a Gaussian. (Bromiley (2003))

$$\mathbb{E}\left[\frac{\partial}{\partial \mathrm{x}}\mathbf{erf}\left(\frac{\mathrm{X}}{\sqrt{2\acute{\sigma}_j^2}}\right)\right] = \int_{-\infty}^{\infty}\frac{2}{\acute{\sigma}_j}\varphi\left(\frac{x}{\acute{\sigma}_j}\right)\frac{1}{\tilde{\sigma}}\varphi\left(\frac{x-\tilde{\mu}}{\tilde{\sigma}}\right)\mathrm{d}x$$

$$= \frac{2}{\sqrt{\tilde{\sigma}^2+\acute{\sigma}_j^2}}\,\varphi\left(\frac{\tilde{\mu}}{\sqrt{\tilde{\sigma}^2+\acute{\sigma}_j^2}}\right)$$

Therefore,

$$\boxed{A_1 = \frac{1}{p}\sum_{j=1}^{p}\frac{2}{\hat{\sigma}_j}\,\varphi\left(\frac{\tilde{\mu}}{\hat{\sigma}_j}\right)} \tag{10}$$

and each term of the summation is a Gaussian function written in its standardized form.

### A.2.3  Find $A_2$ and beyond

In previous section, we show that the first derivative of the error function is a Gaussian, thus the expected value of which is the convolution of two Gaussians. Similarly, we can obtain $A_2$, $A_3$, etc. by convolving the second, third, and higher order Gaussian derivatives with another Gaussian.

Gaussian derivatives can be represented by Hermite polynomial $\mathbf{H}_s(x)$.

$$\frac{d^s}{dx^s}\left[\frac{1}{\sigma}\varphi\left(\frac{x}{\sigma}\right)\right] = \left(\frac{-1}{\sqrt{2\sigma^2}}\right)^s\mathbf{H}_s\left(\frac{x}{\sqrt{2\sigma^2}}\right)\frac{1}{\sigma}\varphi\left(\frac{x}{\sigma}\right)$$

For examples,

$$\mathbf{H}_0(x) = 1$$
$$\mathbf{H}_1(x) = 2x$$
$$\mathbf{H}_2(x) = 4x^2 - 2$$
$$\mathbf{H}_3(x) = 8x^3 - 12x$$
$$\cdots$$

There are implemented functions for this from various scientific computing tools, such as hermiteH() from MATLAB and scipy.special.hermite() from SciPy.

Hence,

$$
\mathbb{E}\left[\frac{\partial^s}{\partial \mathrm{x}^s}\mathbf{erf}\left(\frac{\mathrm{X}}{\sqrt{2\acute{\sigma}_j^2}}\right)\right]
$$

$$
= \int_{-\infty}^{\infty}\left[\frac{\partial^s}{\partial x^s}\mathbf{erf}\left(\frac{x}{\sqrt{2\acute{\sigma}_j^2}}\right)\right]p(x)\mathrm{d}x
$$

$$
= 2\int_{-\infty}^{\infty}\frac{\partial^{s-1}}{\partial x^{s-1}}\left[\frac{1}{\acute{\sigma}_j}\varphi\left(\frac{x}{\acute{\sigma}_j}\right)\right]\frac{1}{\tilde{\sigma}}\varphi\left(\frac{x-\tilde{\mu}}{\tilde{\sigma}}\right)\mathrm{d}x
$$

$$
= 2\left(\frac{-1}{\sqrt{2\acute{\sigma}_j^2}}\right)^{s-1}\int_{-\infty}^{\infty}\mathbf{H}_{s-1}\left(\frac{x}{\sqrt{2\acute{\sigma}_j^2}}\right)\frac{1}{\acute{\sigma}_j}\varphi\left(\frac{x}{\acute{\sigma}_j}\right)\frac{1}{\tilde{\sigma}}\varphi\left(\frac{x-\tilde{\mu}}{\tilde{\sigma}}\right)dx
$$

$$
= 2\left(\frac{-1}{\sqrt{2\acute{\sigma}_j^2}}\right)^{s-1}\frac{1}{\hat{\sigma}_j}\varphi\left(\frac{\tilde{\mu}}{\hat{\sigma}_j}\right)\int_{-\infty}^{\infty}\mathbf{H}_{s-1}\left(\frac{x}{\sqrt{2\acute{\sigma}_j^2}}\right)\frac{1}{\bar{\sigma}_j}\varphi\left(\frac{x-\bar{\mu}}{\bar{\sigma}_j}\right)dx
$$

where $\bar{\mu} = \tilde{\mu}\frac{\acute{\sigma}_j^2}{\hat{\sigma}_j^2}$ $\bar{\sigma}_j^2 = \tilde{\sigma}^2\frac{\acute{\sigma}_j^2}{\hat{\sigma}_j^2}$. The convolution of a Hermite polynomial and a Gaussian pdf is a known integral Gradshteyn and Ryzhik (2015)

$$
= 2\left(\frac{-1}{\sqrt{2\acute{\sigma}_j^2}}\right)^{s-1}\frac{1}{\hat{\sigma}_j}\varphi\left(\frac{\tilde{\mu}}{\hat{\sigma}_j}\right)\left(1 - 2\bar{\sigma}^2\frac{1}{2\acute{\sigma}_j^2}\right)^{\frac{s-1}{2}}\mathbf{H}_{s-1}\left(\frac{\bar{\mu}/\sqrt{2\acute{\sigma}_j^2}}{\left(1 - 2\bar{\sigma}^2\frac{1}{2\acute{\sigma}_j^2}\right)^{\frac{1}{2}}}\right)
$$

$$
= 2\left(\frac{-1}{\sqrt{2\hat{\sigma}_j^2}}\right)^{s-1}\mathbf{H}_{s-1}\left(\frac{\tilde{\mu}}{\sqrt{2\hat{\sigma}_j^2}}\right)\frac{1}{\hat{\sigma}_j}\varphi\left(\frac{\tilde{\mu}}{\hat{\sigma}_j}\right)
$$

Therefore, we can write the formula of $A_s$ for $s \geq 1$

$$
A_s(s \geq 1) = \frac{1}{s!}\frac{1}{p}\sum_{j=1}^{p}2\left(\frac{-1}{\sqrt{2\hat{\sigma}_j^2}}\right)^{s-1}\mathbf{H}_{s-1}\left(\frac{\tilde{\mu}}{\sqrt{2\hat{\sigma}_j^2}}\right)\frac{1}{\hat{\sigma}_j}\varphi\left(\frac{\tilde{\mu}}{\hat{\sigma}_j}\right) \tag{11}
$$

To give a few examples,

$$
\begin{aligned}
A_2 &= \frac{1}{2!\,p}\sum_{j=1}^{p} -2\frac{\tilde{\mu}}{\hat{\sigma}_j^2}\,\frac{1}{\hat{\sigma}_j}\varphi\left(\frac{\tilde{\mu}}{\hat{\sigma}_j}\right) \\
A_3 &= \frac{1}{3!\,p}\sum_{j=1}^{p} 2\frac{\tilde{\mu}^2-\hat{\sigma}_j^2}{\hat{\sigma}_j^4}\,\frac{1}{\hat{\sigma}_j}\varphi\left(\frac{\tilde{\mu}}{\hat{\sigma}_j}\right) \\
A_4 &= \frac{1}{4!\,p}\sum_{j=1}^{p} 2\left(\frac{-\tilde{\mu}^3+3\tilde{\mu}\hat{\sigma}_j^2}{\hat{\sigma}_j^6}\right)\frac{1}{\hat{\sigma}_j}\varphi\left(\frac{\tilde{\mu}}{\hat{\sigma}_j}\right) \\
&\cdots
\end{aligned}
\tag{12}
$$

Note that we will reuse this relation in the following section

$$
\int_{-\infty}^{\infty}\left(\frac{-1}{\sqrt{2\acute{\sigma}_j^2}}\right)^s \mathbf{H}_s\left(\frac{x}{\sqrt{2\acute{\sigma}_j^2}}\right)\frac{1}{\hat{\sigma}_j}\varphi\left(\frac{x}{\hat{\sigma}_j}\right)\frac{1}{\tilde{\sigma}}\varphi\left(\frac{x-\tilde{\mu}}{\tilde{\sigma}}\right)dx = \left(\frac{-1}{\sqrt{2\hat{\sigma}_j^2}}\right)^s \mathbf{H}_s\left(\frac{\tilde{\mu}}{\sqrt{2\hat{\sigma}_j^2}}\right)\frac{1}{\hat{\sigma}_j}\varphi\left(\frac{\tilde{\mu}}{\hat{\sigma}_j}\right)
\tag{13}
$$

### A.3 Sigmoid layers

We can apply the same framework on **sigmoid** layers, with modifications

$$
\mathbf{sigmoid}(x) = \frac{1}{1+\mathbf{exp}(-x)} \approx \frac{1}{2}+\frac{1}{2p}\sum_{j=1}^{p}\mathbf{erf}\left(\gamma_j x\right)
$$

Using `fmincon` in MATLAB, we find a set of $\gamma = (0.2791, 0.4298, 0.4298, 0.6306)^{\top}$. Then the first four factors of the Taylor polynomials are listed below. $A_0$ is represented in complementary error function **erfc** to avoid subtractive cancellation that leads to inaccuracy in the tails. Note that except for $A_0$, all $A_s$ of **sigmoid** layers are just $1/2$ of those of **tanh** layers.

$$
\begin{aligned}
A_0 &= \frac{1}{p}\sum_{j=1}^{p}\frac{1}{2}\mathbf{erfc}\left(-\frac{\tilde{\mu}}{\sqrt{2\hat{\sigma}_j^2}}\right) \\
A_1 &= \frac{1}{p}\sum_{j=1}^{p}\frac{1}{\hat{\sigma}_j}\,\varphi\left(\frac{\tilde{\mu}}{\hat{\sigma}_j}\right) \\
A_2 &= \frac{1}{2!\,p}\sum_{j=1}^{p}-\frac{\tilde{\mu}}{\hat{\sigma}_j^2}\,\frac{1}{\hat{\sigma}_j}\varphi\left(\frac{\tilde{\mu}}{\hat{\sigma}_j}\right) \\
A_3 &= \frac{1}{3!\,p}\sum_{j=1}^{p}\frac{\tilde{\mu}^2-\hat{\sigma}_j^2}{\hat{\sigma}_j^4}\,\frac{1}{\hat{\sigma}_j}\varphi\left(\frac{\tilde{\mu}}{\hat{\sigma}_j}\right) \\
A_4 &= \frac{1}{4!\,p}\sum_{j=1}^{p}\left(\frac{-\tilde{\mu}^3+3\tilde{\mu}\hat{\sigma}_j^2}{\hat{\sigma}_j^6}\right)\frac{1}{\hat{\sigma}_j}\varphi\left(\frac{\tilde{\mu}}{\hat{\sigma}_j}\right) \\
&\cdots
\end{aligned}
\tag{14}
$$

### A.4 Softplus layers

The derivation of pseudo-Taylor polynomials for a **softplus** layer is related to that for a **sigmoid** layer, since the derivative of the **softplus** function is the **sigmoid** function with scaling factor $\beta$, and the latter can be approximated with a linear combination of Gaussian cdf (or error functions like we did in the previous

section). We have

$$\mathbf{softplus}(x) = \frac{1}{\beta}\mathbf{log}\left(1 + e^{\beta x}\right)$$

Then we use the approximation of sum of **independent** standard Gaussian cdf $\mathbf{\Phi}$

$$\frac{\partial}{\partial x}\mathbf{softplus}(x) = \frac{1}{1 + e^{-\beta x}} \approx \frac{1}{p}\sum_{j=1}^{p}\mathbf{\Phi}\left(\frac{x}{\acute{\sigma}_j}\right) \tag{15}$$

where we re-define

$$\acute{\sigma}_j^2 = \frac{1}{2\gamma_j^2\beta^2} \tag{16}$$

Note that $\acute{\sigma}_j^2$ changes definition and should not be confused with that in the **tanh** and **sigmoid** sections.

### A.4.1 Find $A_0$

First we apply substitution of variables $X = \tilde{\mu} + \Xi$, then

$$\mathbf{softplus}(x) = \mathbf{softplus}(\tilde{\mu}, \xi) = \frac{1}{\beta}\mathbf{log}\left(1 + e^{\beta(\tilde{\mu}+\xi)}\right)$$

Notice that $\dfrac{\partial}{\partial x} = \dfrac{\partial}{\partial \xi}$ since $\tilde{\mu}$ is constant, then

$$A_0 = \int_{-\infty}^{\infty}\mathbf{softplus}(\tilde{\mu}, \xi)\ p(\xi)\ \mathrm{d}\xi$$

$$= \int_{-\infty}^{\infty}p(\xi)\mathrm{d}\xi\int_{-\infty}^{\tilde{\mu}}\frac{\partial}{\partial\zeta}\mathbf{softplus}(\zeta, \xi)\ \mathrm{d}\zeta$$

by Fubini's theorem (Fubini, 1907)

$$= \int_{-\infty}^{\tilde{\mu}}\mathrm{d}\zeta\int_{-\infty}^{\infty}\frac{\partial}{\partial\zeta}\mathbf{softplus}(\zeta, \xi)\ p(\xi)\mathrm{d}\xi$$

$$\approx \frac{1}{p}\sum_{j=1}^{p}\int_{-\infty}^{\tilde{\mu}}\mathrm{d}\zeta\int_{-\infty}^{\infty}\mathbf{\Phi}\left(\frac{\zeta+\xi}{\acute{\sigma}_j}\right)\frac{1}{\tilde{\sigma}}\varphi\left(\frac{\xi}{\tilde{\sigma}}\right)\mathrm{d}\xi$$

using the known Gaussian integral identity $\int_{-\infty}^{\infty}\mathbf{\Phi}(ax+b)\varphi(x)dx = \mathbf{\Phi}\left(\frac{b}{\sqrt{1+a^2}}\right)$

$$= \frac{1}{p}\sum_{j=1}^{p}\int_{-\infty}^{\tilde{\mu}}\mathbf{\Phi}\left(\frac{\zeta}{\hat{\sigma}_j}\right)\mathrm{d}\zeta$$

$$= \frac{1}{p}\sum_{j=1}^{p}\left[\tilde{\mu}\ \mathbf{\Phi}\left(\frac{\tilde{\mu}}{\hat{\sigma}_j}\right) + \hat{\sigma}_j\varphi\left(\frac{\tilde{\mu}}{\hat{\sigma}_j}\right)\right]$$

$$= \frac{1}{p}\sum_{j=1}^{p}\left[\frac{\tilde{\mu}}{2}\ \mathbf{erfc}\left(-\frac{\tilde{\mu}}{\sqrt{2\hat{\sigma}_j^2}}\right) + \hat{\sigma}_j\varphi\left(\frac{\tilde{\mu}}{\hat{\sigma}_j}\right)\right]$$

Or, with simplification

$$A_0 = A_1\tilde{\mu} + \frac{1}{p}\sum_{j=1}^{p}\hat{\sigma}_j\varphi\left(\frac{\tilde{\mu}}{\hat{\sigma}_j}\right) \tag{17}$$

### A.4.2   Find $A_1$

Since the first derivative of the **softplus** function is just a **sigmoid** function with scaling factor $\beta$, we can immediately write $A_1$ using previous results

$$A_1 = \frac{1}{p} \sum_{j=1}^{p} \frac{1}{2} \mathbf{erfc} \left( -\frac{\tilde{\mu}}{\sqrt{2\hat{\sigma}_j^2}} \right) \tag{18}$$

### A.4.3   Find $A_2$ and beyond

In previous section, we find that $\nabla \mathbf{softplus}(x)$ is approximately a Gaussian cdf. Subsequently, $\nabla^2 \mathbf{softplus}(x)$ is approximately a Gaussian. Since Gaussian function is infinitely differentiable, all $A_s(s > 2)$ can be found using Gaussian derivatives, which can be represented by Hermite polynomial $\mathbf{H}_s(x)$ introduced above.

$$
\begin{aligned}
A_s &= \frac{1}{s!} \mathbb{E}\left[ \frac{\partial^s}{\partial x^s} \mathbf{softplus}(x) \right] \\
&\approx \frac{1}{s!\, p} \sum_{j=1}^{p} \int_{-\infty}^{\infty} \frac{\partial^{s-2}}{\partial x^{s-2}} \left[ \frac{1}{\acute{\sigma}_j} \varphi\left( \frac{x}{\acute{\sigma}_j} \right) \right] p(x)\ \mathrm{d}x \\
&= \frac{1}{s!\, p} \sum_{j=1}^{p} \left( \frac{-1}{\sqrt{2\acute{\sigma}_j^2}} \right)^{s-2} \int_{-\infty}^{\infty} \mathbf{H}_{s-2}\left( \frac{x}{\sqrt{2\acute{\sigma}_j^2}} \right) \frac{1}{\acute{\sigma}_j}\ \varphi\left( \frac{x}{\acute{\sigma}_j} \right) \frac{1}{\tilde{\sigma}} \varphi\left( \frac{x - \tilde{\mu}}{\tilde{\sigma}} \right) \mathrm{d}x
\end{aligned}
$$

we solved this integral in **tanh** section

$$= \frac{1}{s!\, p} \sum_{j=1}^{p} \left( \frac{-1}{\sqrt{2\hat{\sigma}_j^2}} \right)^{s-2} \mathbf{H}_{s-2}\left( \frac{\tilde{\mu}}{\sqrt{2\hat{\sigma}_j^2}} \right) \frac{1}{\hat{\sigma}_j} \varphi\left( \frac{\tilde{\mu}}{\hat{\sigma}_j} \right)$$

To summarize, $A_s(s \geq 2)$ can be expressed as

$$A_s(s \geq 2) = \frac{1}{s!\, p} \sum_{j=1}^{p} \left( \frac{-1}{\sqrt{2\hat{\sigma}_j^2}} \right)^{s-2} \mathbf{H}_{s-2}\left( \frac{\tilde{\mu}}{\sqrt{2\hat{\sigma}_j^2}} \right) \frac{1}{\hat{\sigma}_j} \varphi\left( \frac{\tilde{\mu}}{\hat{\sigma}_j} \right) \tag{19}$$

For examples,

$$
\begin{aligned}
A_2 &= \frac{1}{2!\, p} \sum_{j=1}^{p} \frac{1}{\hat{\sigma}_j}\ \varphi\left( \frac{\tilde{\mu}}{\hat{\sigma}_j} \right) \\
A_3 &= \frac{1}{3!\, p} \sum_{j=1}^{p} -\frac{\tilde{\mu}}{\hat{\sigma}_j^2}\ \frac{1}{\hat{\sigma}_j} \varphi\left( \frac{\tilde{\mu}}{\hat{\sigma}_j} \right) \\
A_4 &= \frac{1}{4!\, p} \sum_{j=1}^{p} \frac{\tilde{\mu}^2 - \hat{\sigma}_j^2}{\hat{\sigma}_j^4}\ \frac{1}{\hat{\sigma}_j} \varphi\left( \frac{\tilde{\mu}}{\hat{\sigma}_j} \right) \\
\dots &
\end{aligned}
\tag{20}
$$

### A.5   ReLU, Leaky ReLU, and Piece-wise Linear layers

Since ReLU function is only first-order differentiable $(x > 0)$, we cannot do PTPE directly. However, given its relation to **softplus** function,

$$\lim_{\beta \to \infty} \frac{1}{\beta} \mathbf{log}\left( 1 + e^{\beta x} \right) = \max\{0, x\}$$

we can reuse the results for **softplus** layers by applying the limit

$$\lim_{\beta\to\infty}\acute{\sigma}_j^2 = 0 \qquad \text{and} \qquad \lim_{\beta\to\infty}\hat{\sigma}_j^2 = \tilde{\sigma}^2$$

Therefore,

$$
\begin{aligned}
A_0 &= A_1\tilde{\mu} + \tilde{\sigma}\boldsymbol{\varphi}\left(\frac{\tilde{\mu}}{\tilde{\sigma}}\right) \\[4pt]
A_1 &= \frac{1}{2}\mathbf{erfc}\left(-\frac{\tilde{\mu}}{\sqrt{2\tilde{\sigma}^2}}\right) \\[4pt]
A_2 &= \frac{1}{2!}\frac{1}{\tilde{\sigma}}\,\varphi\left(\frac{\tilde{\mu}}{\tilde{\sigma}}\right) \\[4pt]
A_3 &= \frac{1}{3!} - \frac{\tilde{\mu}}{\tilde{\sigma}^2}\frac{1}{\tilde{\sigma}}\varphi\left(\frac{\tilde{\mu}}{\tilde{\sigma}}\right) \\[4pt]
A_4 &= \frac{1}{4!}\frac{\tilde{\mu}^2 - \tilde{\sigma}^2}{\tilde{\sigma}^4}\frac{1}{\tilde{\sigma}}\varphi\left(\frac{\tilde{\mu}}{\tilde{\sigma}}\right) \\[4pt]
\dots
\end{aligned}
\tag{21}
$$

and for $s \geq 2$ we have the general form of

$$A_s(s \geq 2) = \frac{1}{s!}\left(\frac{-1}{\sqrt{2\tilde{\sigma}_j^2}}\right)^{s-2}\mathbf{H}_{s-2}\left(\frac{\tilde{\mu}}{\sqrt{2\tilde{\sigma}_j^2}}\right)\frac{1}{\tilde{\sigma}_j}\varphi\left(\frac{\tilde{\mu}}{\tilde{\sigma}_j}\right) \tag{22}$$

On the other hand, leaky ReLU can be considered as superposition of two ReLU functions - consider a leaky ReLU with negative slope of $\theta$

$$\mathbf{LeakyReLU}(x;\theta) = \mathbf{ReLU}(x) - \theta\,\mathbf{ReLU}(-x) \tag{23}$$

which can also be written as

$$\lim_{\beta\to\infty}\mathbf{softplus}(x) - \theta\,\mathbf{softplus}(-x)$$

Therefore,

$$
\begin{aligned}
A_0 &= \lim_{\beta\to\infty}\frac{1}{p}\sum_{j=1}^{p}\left[\tilde{\mu}\boldsymbol{\Phi}\left(\frac{\tilde{\mu}}{\hat{\sigma}_j}\right) + \hat{\sigma}_j\boldsymbol{\varphi}\left(\frac{\tilde{\mu}}{\hat{\sigma}_j}\right)\right] - \theta\left[-\tilde{\mu}\boldsymbol{\Phi}\left(-\frac{\tilde{\mu}}{\hat{\sigma}_j}\right) + \hat{\sigma}_j\boldsymbol{\varphi}\left(\frac{\tilde{\mu}}{\hat{\sigma}_j}\right)\right] \\[4pt]
&= \theta\tilde{\mu} + (1-\theta)\left[\tilde{\mu}\boldsymbol{\Phi}\left(\frac{\tilde{\mu}}{\tilde{\sigma}}\right) + \tilde{\sigma}\boldsymbol{\varphi}\left(\frac{\tilde{\mu}}{\tilde{\sigma}}\right)\right]
\end{aligned}
$$

To find the expected value of the derivative of **LeakyReLU**, first we find the derivative

$$
\begin{aligned}
\frac{\partial}{\partial x}\mathbf{LeakyReLU}(x\,;\,\theta) &= \lim_{\beta\to\infty}\frac{\partial}{\partial x}\mathbf{softplus}(x) - \theta\frac{\partial}{\partial x}\mathbf{softplus}(-(x)) \\[4pt]
&= \lim_{\beta\to\infty}\frac{1}{1+e^{-\beta(x)}} + \frac{\theta}{1+e^{\beta(x)}} \\[4pt]
&\approx \frac{1}{p}\sum_{j=1}^{p}\boldsymbol{\Phi}\left(\frac{x}{\acute{\sigma}_j}\right) + \theta\boldsymbol{\Phi}\left(\frac{-x}{\acute{\sigma}_j}\right) \\[4pt]
&= \lim_{\beta\to\infty}\theta + \frac{1-\theta}{p}\sum_{j=1}^{p}\boldsymbol{\Phi}\left(\frac{x}{\acute{\sigma}_j}\right)
\end{aligned}
$$

Then we can write $A_1$ for **LeakyReLU** as

$$
\begin{aligned}
A_1 &= \lim_{\beta \to \infty} \int_{-\infty}^{\infty} \left[ \theta + \frac{1-\theta}{p} \sum_{j=1}^{p} \mathbf{\Phi}\left( \frac{x}{\acute{\sigma}_j} \right) \right] \frac{1}{\tilde{\sigma}} \varphi\left( \frac{x - \tilde{\mu}}{\tilde{\sigma}} \right) \mathrm{d}x \\
&= \theta + \lim_{\beta \to \infty} \frac{1-\theta}{p} \sum_{j=1}^{p} \int_{-\infty}^{\infty} \mathbf{\Phi}\left( \frac{x}{\acute{\sigma}_j} \right) \frac{1}{\tilde{\sigma}} \varphi\left( \frac{x - \tilde{\mu}}{\tilde{\sigma}} \right) \mathrm{d}x \\
&= \theta + \lim_{\beta \to \infty} \frac{1-\theta}{p} \sum_{j=1}^{p} \mathbf{\Phi}\left( \frac{\tilde{\mu}}{\hat{\sigma}_j} \right) \\
&= \theta + (1-\theta)\mathbf{\Phi}\left( \frac{\tilde{\mu}}{\tilde{\sigma}} \right)
\end{aligned}
$$

Rewrite in complementary error function

$$
A_1 = \theta + \frac{1-\theta}{2}\mathbf{erfc}\left( -\frac{\tilde{\mu}}{\sqrt{2\tilde{\sigma}^2}} \right) \tag{24}
$$

Note that we can also rewrite $A_0$ using the result of $A_1$ to improve computational efficiency.

$$
A_0 = A_1 \tilde{\mu} + (1-\theta)\tilde{\sigma}\varphi\left( \frac{\tilde{\mu}}{\tilde{\sigma}} \right) \tag{25}
$$

Note that starting from the second order, the derivative of **LeakyReLU** is just that of **ReLU** scaled by $1-\theta$. Therefore,

$$
\begin{aligned}
A_2 &= \frac{1-\theta}{2} \frac{1}{\tilde{\sigma}} \varphi\left( \frac{\tilde{\mu}}{\tilde{\sigma}} \right) \\
A_3 &= -\frac{1-\theta}{3!} \frac{\tilde{\mu}}{\tilde{\sigma}^3} \varphi\left( \frac{\tilde{\mu}}{\tilde{\sigma}} \right) \\
A_4 &= \frac{1-\theta}{4!} \frac{\tilde{\mu}^2 - \tilde{\sigma}^2}{\tilde{\sigma}^5} \varphi\left( \frac{\tilde{\mu}}{\tilde{\sigma}} \right) \\
&\dots
\end{aligned} \tag{26}
$$

and for $s \geq 2$, we have the general form of

$$
A_s(s \geq 2) = \frac{1-\theta}{s!} \left( \frac{-1}{\sqrt{2\tilde{\sigma}^2}} \right)^{s-2} \mathbf{H}_{s-2}\left( \frac{\tilde{\mu}}{\sqrt{2\tilde{\sigma}^2}} \right) \frac{1}{\tilde{\sigma}}\varphi\left( \frac{\tilde{\mu}}{\tilde{\sigma}} \right) \tag{27}
$$

Similarly, any piece-wise linear activation function can be described as a combination of ReLU functions with different scaling, shifting, and/or mirroring. Thus, their pseudo Taylor coefficients can be found using the same methodology.

### A.6 GELU layers

**GELU** (Gaussian Error Linear Unit) is defined as the product of input and a standard Gaussian cdf

$$
\mathbf{GELU}(x) = x \ \mathbf{\Phi}(x)
$$

and we can write the derivatives (with order $s \geq 1$) of **GELU** as

$$
\frac{\partial^s}{\partial x^s}\mathbf{GELU}(x) = s\frac{\partial^{s-1}}{\partial x^{s-1}}\mathbf{\Phi}(x) + x\frac{\partial^s}{\partial x^s}\mathbf{\Phi}(x)
$$

### A.6.1 Find $A_0$

$$A_0 = \mathbb{E}\left[\mathbf{GELU}(x)\right]$$
$$= \int_{-\infty}^{\infty} x\mathbf{\Phi}(x)\frac{1}{\tilde{\sigma}}\varphi\left(\frac{x-\tilde{\mu}}{\tilde{\sigma}}\right)\mathrm{d}x$$
$$= \int_{-\infty}^{\infty} (\tilde{\mu}+\xi)\int_{-\infty}^{\tilde{\mu}}\varphi\left(\zeta+\xi\right)\mathrm{d}\zeta\,\frac{1}{\tilde{\sigma}}\varphi\left(\frac{\xi}{\tilde{\sigma}}\right)\mathrm{d}\xi$$
$$= \tilde{\mu}\int_{-\infty}^{\tilde{\mu}}\int_{-\infty}^{\infty}\varphi\left(\zeta+\xi\right)\frac{1}{\tilde{\sigma}}\varphi\left(\frac{\xi}{\tilde{\sigma}}\right)\mathrm{d}\xi\,\mathrm{d}\zeta + \int_{-\infty}^{\tilde{\mu}}\int_{-\infty}^{\infty}\xi\,\varphi\left(\zeta+\xi\right)\frac{1}{\tilde{\sigma}}\varphi\left(\frac{\xi}{\tilde{\sigma}}\right)\mathrm{d}\xi\,\mathrm{d}\zeta$$
$$= \tilde{\mu}\int_{-\infty}^{\tilde{\mu}}\frac{1}{\sqrt{1+\tilde{\sigma}^2}}\varphi\left(\frac{\zeta}{\sqrt{1+\tilde{\sigma}^2}}\right)\mathrm{d}\zeta + \int_{-\infty}^{\tilde{\mu}}\frac{1}{\sqrt{1+\tilde{\sigma}^2}}\varphi\left(\frac{\zeta}{\sqrt{1+\tilde{\sigma}^2}}\right)\frac{-\zeta\tilde{\sigma}^2}{1+\tilde{\sigma}^2}\mathrm{d}\zeta$$
$$= \tilde{\mu}\mathbf{\Phi}\left(\frac{\tilde{\mu}}{\sqrt{1+\tilde{\sigma}^2}}\right) + \frac{\tilde{\sigma}^2}{\sqrt{1+\tilde{\sigma}^2}}\varphi\left(\frac{\tilde{\mu}}{\sqrt{1+\tilde{\sigma}^2}}\right)$$

We re-define $\hat{\sigma}^2$

$$\hat{\sigma}^2 = 1+\tilde{\sigma}^2 \tag{28}$$

and re-write the result with complementary error function

$$\boxed{A_0 = \frac{\tilde{\mu}}{2}\mathbf{erfc}\left(-\frac{\tilde{\mu}}{\sqrt{2\hat{\sigma}^2}}\right) + \frac{\tilde{\sigma}^2}{\hat{\sigma}}\varphi\left(\frac{\tilde{\mu}}{\hat{\sigma}}\right)} \tag{29}$$

### A.6.2 Find $A_1$

$$A_1 = \mathbb{E}\left[\frac{\partial}{\partial x}\mathbf{GELU}(x)\right]$$
$$= \int_{-\infty}^{\infty}\mathbf{\Phi}(x)\frac{1}{\tilde{\sigma}}\varphi(\frac{x-\tilde{\mu}}{\tilde{\sigma}})\mathrm{d}x + \int_{-\infty}^{\infty}x\varphi(x)\frac{1}{\tilde{\sigma}}\varphi(\frac{x-\tilde{\mu}}{\tilde{\sigma}})\mathrm{d}x$$

using results of previous section

$$= \mathbf{\Phi}\left(\frac{\tilde{\mu}}{\hat{\sigma}}\right) + \frac{\tilde{\mu}}{\hat{\sigma}^2}\frac{1}{\hat{\sigma}}\varphi\left(\frac{\tilde{\mu}}{\hat{\sigma}}\right)$$

Therefore,

$$\boxed{A_1 = \frac{1}{2}\mathbf{erfc}\left(-\frac{\tilde{\mu}}{\sqrt{2\hat{\sigma}^2}}\right) + \frac{\tilde{\mu}}{\hat{\sigma}^2}\frac{1}{\hat{\sigma}}\varphi\left(\frac{\tilde{\mu}}{\hat{\sigma}}\right)} \tag{30}$$

### A.6.3 Find $A_2$ and beyond

Higher order coefficients $(A_s(s \geq 2))$ all consist of two parts: (i) a term of expected value of a Gaussian derivative, (ii) a term of expected value of the product of $x$ and a Gaussian derivative. We have already found a general form of the first term in the **tanh** section

$$\mathbb{E}\left[s\frac{\partial^{s-2}}{\partial x^{s-2}}\varphi(x)\right] = s\left(\frac{-1}{\sqrt{2\hat{\sigma}^2}}\right)^{s-2}\mathbf{H}_{s-2}\left(\frac{\tilde{\mu}}{\sqrt{2\hat{\sigma}^2}}\right)\frac{1}{\hat{\sigma}}\varphi\left(\frac{\tilde{\mu}}{\hat{\sigma}}\right)$$

To solve the second part, we need to use the Hermite polynomial recurrence relation:

$$x\,\mathbf{H}_{s-1}(x) = \frac{1}{2}\mathbf{H}_s(x) + s\,\mathbf{H}_{s-2}(x)$$

$$\mathbb{E}\left[x\frac{\partial^{s-1}}{\partial x^{s-1}}\varphi(x)\right]$$

$$= \left(\frac{-1}{\sqrt{2}}\right)^{s-1}\int_{-\infty}^{\infty} x\ \mathbf{H}_{s-1}\left(\frac{x}{\sqrt{2}}\right)\varphi(x)\frac{1}{\tilde{\sigma}}\varphi\left(\frac{x-\tilde{\mu}}{\tilde{\sigma}}\right)\mathrm{d}x$$

$$= \left(\frac{-1}{\sqrt{2}}\right)^{s-1}\sqrt{2}\int_{-\infty}^{\infty}\frac{x}{\sqrt{2}}\ \mathbf{H}_{s-1}\left(\frac{x}{\sqrt{2}}\right)\ \varphi(x)\frac{1}{\tilde{\sigma}}\varphi\left(\frac{x-\tilde{\mu}}{\tilde{\sigma}}\right)\mathrm{d}x$$

$$= \left(\frac{-1}{\sqrt{2}}\right)^{s-1}\sqrt{2}\int_{-\infty}^{\infty}\left[\frac{1}{2}\mathbf{H}_s\left(\frac{x}{\sqrt{2}}\right) + (s-1)\mathbf{H}_{s-2}\left(\frac{x}{\sqrt{2}}\right)\right]\varphi(x)\frac{1}{\tilde{\sigma}}\varphi\left(\frac{x-\tilde{\mu}}{\tilde{\sigma}}\right)\mathrm{d}x$$

$$= -\left(\frac{-1}{\sqrt{2}}\right)^{s}\int_{-\infty}^{\infty}\mathbf{H}_s\left(\frac{x}{\sqrt{2}}\right)\varphi(x)\frac{1}{\tilde{\sigma}}\varphi\left(\frac{x-\tilde{\mu}}{\tilde{\sigma}}\right)\mathrm{d}x\ \cdots$$

$$-\ (s-1)\left(\frac{-1}{\sqrt{2}}\right)^{s-2}\int_{-\infty}^{\infty}\mathbf{H}_{s-2}\left(\frac{x}{\sqrt{2}}\right)\varphi(x)\frac{1}{\tilde{\sigma}}\varphi\left(\frac{x-\tilde{\mu}}{\tilde{\sigma}}\right)\mathrm{d}x$$

by equation 13

$$= \frac{1}{\hat{\sigma}}\varphi\left(\frac{\tilde{\mu}}{\hat{\sigma}}\right)\left[-\left(\frac{-1}{\sqrt{2\hat{\sigma}^2}}\right)^{s}\mathbf{H}_s\left(\frac{\tilde{\mu}}{\sqrt{2\hat{\sigma}^2}}\right) - (s-1)\left(\frac{-1}{\sqrt{2\hat{\sigma}^2}}\right)^{s-2}\mathbf{H}_{s-2}\left(\frac{\tilde{\mu}}{\sqrt{2\hat{\sigma}^2}}\right)\right]$$

Sum the two integral together, we get the general form of $A_s(s \geq 2)$

$$A_s(s \geq 2) = \frac{1}{s!}\left[\left(\frac{-1}{\sqrt{2\hat{\sigma}^2}}\right)^{s-2}\mathbf{H}_{s-2}\left(\frac{\tilde{\mu}}{\sqrt{2\hat{\sigma}^2}}\right) - \left(\frac{-1}{\sqrt{2\hat{\sigma}^2}}\right)^{s}\mathbf{H}_s\left(\frac{\tilde{\mu}}{\sqrt{2\hat{\sigma}^2}}\right)\right]\frac{1}{\hat{\sigma}}\varphi\left(\frac{\tilde{\mu}}{\hat{\sigma}}\right) \tag{31}$$

For examples,

$$\begin{aligned}
A_2 &= \frac{1}{2!}\left[1 + \frac{1}{\hat{\sigma}^2} - \frac{\tilde{\mu}^2}{\hat{\sigma}^4}\right]\frac{1}{\hat{\sigma}}\varphi\left(\frac{\tilde{\mu}}{\hat{\sigma}}\right) \\
A_3 &= -\frac{1}{3!}\left[\frac{\tilde{\mu}}{\hat{\sigma}^2} + \frac{3\tilde{\mu}}{\hat{\sigma}^4} - \frac{\tilde{\mu}^3}{\hat{\sigma}^6}\right]\frac{1}{\hat{\sigma}}\varphi\left(\frac{\tilde{\mu}}{\hat{\sigma}}\right) \\
A_4 &= \frac{1}{4!}\left[-\frac{1}{\hat{\sigma}^2} + \frac{\tilde{\mu}^2-3}{\hat{\sigma}^4} + \frac{6\tilde{\mu}^2}{\hat{\sigma}^6} - \frac{\tilde{\mu}^4}{\hat{\sigma}^8}\right]\frac{1}{\hat{\sigma}}\varphi\left(\frac{\tilde{\mu}}{\hat{\sigma}}\right) \\
\cdots
\end{aligned} \tag{32}$$

## A.7 SiLU layers

**SiLU** (Sigmoid Linear Unit), equivalent to **Swish** when $\beta = 1$, is defined as the product of input and a **sigmoid** function

$$\mathbf{SiLU}(x) = x\ \mathbf{Sigmoid}(x)$$

In the previous section, we approximate **Sigmoid** function with error functions so that we can reuse derivations from the **Tanh** section. Here we approximate **Sigmoid** function with Gaussian cdf's in order to reuse derivations from the **GELU** section. With $\gamma$ as a numerically optimized scalar vector, let

$$\acute{\sigma}_j^2 = \frac{1}{2\gamma_j^2}\quad,\quad j \in \{1,\cdots,p\}$$

Then, we approximate **SiLU** as

$$\mathbf{SiLU}(x) \approx \frac{x}{p} \sum_{j=1}^{p} \mathbf{\Phi}\left(\frac{x}{\acute{\sigma}_j}\right)$$

and we can write the derivatives (with order $s \geq 1$) of **SiLU** as

$$\frac{\partial^s}{\partial u^s}\mathbf{SiLU}(x) = \frac{s}{p} \sum_{j=1}^{p} \frac{\partial^{s-1}}{\partial u^{s-1}} \mathbf{\Phi}\left(\frac{x}{\acute{\sigma}_j}\right) + \frac{x}{p} \sum_{j=1}^{p} \frac{\partial^s}{\partial u^s} \mathbf{\Phi}\left(\frac{x}{\acute{\sigma}_j}\right)$$

The rest of the derivation is very similar to that of **GELU**, so we only list the final results. With

$$\hat{\sigma}_j^2 = \tilde{\sigma}^2 + \acute{\sigma}_j^2$$

$$
\begin{aligned}
A_0 &= \frac{1}{p} \sum_{j=1}^{p} \frac{\tilde{\mu}}{2} \mathbf{erfc}\left(-\frac{\tilde{\mu}}{\sqrt{2\hat{\sigma}_j^2}}\right) + \frac{\tilde{\sigma}^2}{\hat{\sigma}_j} \boldsymbol{\varphi}\left(\frac{\tilde{\mu}}{\hat{\sigma}_j}\right) \\[2mm]
A_1 &= \frac{1}{p} \sum_{j=1}^{p} \frac{1}{2} \mathbf{erfc}\left(-\frac{\tilde{\mu}}{\sqrt{2\hat{\sigma}_j^2}}\right) + \tilde{\mu}\frac{\acute{\sigma}_j^2}{\hat{\sigma}_j^2}\frac{1}{\hat{\sigma}_j} \boldsymbol{\varphi}\left(\frac{\tilde{\mu}}{\hat{\sigma}_j}\right) \\[2mm]
A_2 &= \frac{1}{2!\,p} \sum_{j=1}^{p} \left[1 + \frac{\acute{\sigma}_j^2}{\hat{\sigma}_j^2} - \frac{\tilde{\mu}^2\acute{\sigma}_j^2}{\hat{\sigma}_j^4}\right] \frac{1}{\hat{\sigma}_j} \boldsymbol{\varphi}\left(\frac{\tilde{\mu}}{\hat{\sigma}_j}\right) \\[2mm]
A_3 &= -\frac{1}{3!\,p} \sum_{j=1}^{p} \left[\frac{\tilde{\mu}}{\hat{\sigma}_j^2} + \frac{3\tilde{\mu}\acute{\sigma}_j^2}{\hat{\sigma}_j^4} - \frac{\tilde{\mu}^3\acute{\sigma}_j^2}{\hat{\sigma}_j^6}\right] \frac{1}{\hat{\sigma}_j} \boldsymbol{\varphi}\left(\frac{\tilde{\mu}}{\hat{\sigma}_j}\right) \\[2mm]
A_4 &= \frac{1}{4!\,p} \sum_{j=1}^{p} \left[-\frac{1}{\hat{\sigma}_j^2} + \frac{\tilde{\mu}^2 - 3\acute{\sigma}_j^2}{\hat{\sigma}_j^4} + \frac{6\tilde{\mu}^2\acute{\sigma}_j^2}{\hat{\sigma}_j^6} - \frac{\tilde{\mu}^4\acute{\sigma}_j^2}{\hat{\sigma}_j^8}\right] \frac{1}{\hat{\sigma}_j} \boldsymbol{\varphi}\left(\frac{\tilde{\mu}}{\hat{\sigma}_j}\right) \\
&\cdots
\end{aligned}
\tag{33}
$$

and the general form of $A_s(s \geq 2)$ is

$$A_s(s \geq 2) = \frac{1}{s!\,p} \sum_{j=1}^{p} \left[\left(\frac{-1}{\sqrt{2\hat{\sigma}^2}}\right)^{s-2} \mathbf{H}_{s-2}\left(\frac{\tilde{\mu}}{\sqrt{2\hat{\sigma}^2}}\right) - \left(\frac{-1}{\sqrt{2\hat{\sigma}^2}}\right)^{s} \mathbf{H}_s\left(\frac{\tilde{\mu}}{\sqrt{2\hat{\sigma}^2}}\right)\right] \frac{1}{\hat{\sigma}} \boldsymbol{\varphi}\left(\frac{\tilde{\mu}}{\hat{\sigma}}\right) \tag{34}$$

Table 4: First four coefficients of the polynomials for seven commonly used nonlinearities. For notational simplicity, all the product, division, and power operations are element-wise. The method for determining $\gamma_j$ is outlined in Table 1.

| | $A_0$ | $A_1$ | $A_2$ | $A_3$ | $\hat{\boldsymbol{\sigma}}_j^2 = \tilde{\boldsymbol{\sigma}}^2 + \acute{\sigma}_j^2$ |
|---|---|---|---|---|---|
| Tanh | $\dfrac{1}{p}\sum_{j=1}^{p}\left(2\mathcal{F}-1\right)$ | $\dfrac{1}{p}\sum_{j=1}^{p}2\mathcal{L}$ | $\dfrac{1}{2\,p}\sum_{j=1}^{p}-2\mathcal{L}\dfrac{\tilde{\boldsymbol{\mu}}}{\hat{\boldsymbol{\sigma}}_j^2}$ | $\dfrac{1}{3!\,p}\sum_{j=1}^{p}2\mathcal{L}\dfrac{\tilde{\boldsymbol{\mu}}^2-\hat{\boldsymbol{\sigma}}_j^2}{\hat{\boldsymbol{\sigma}}_j^4}$ | $\acute{\sigma}_j^2=\dfrac{1}{2\gamma_j^2}$ |
| Sigmoid | $\dfrac{1}{p}\sum_{j=1}^{p}\mathcal{F}$ | $\dfrac{1}{p}\sum_{j=1}^{p}\mathcal{L}$ | $\dfrac{1}{2\,p}\sum_{j=1}^{p}-\mathcal{L}\dfrac{\tilde{\boldsymbol{\mu}}}{\hat{\boldsymbol{\sigma}}_j^2}$ | $\dfrac{1}{3!\,p}\sum_{j=1}^{p}\mathcal{L}\dfrac{\tilde{\boldsymbol{\mu}}^2-\hat{\boldsymbol{\sigma}}_j^2}{\hat{\boldsymbol{\sigma}}_j^4}$ | $\acute{\sigma}_j^2=\dfrac{1}{2\gamma_j^2}$ |
| Softplus | $\dfrac{1}{p}\sum_{j=1}^{p}\mathcal{F}\tilde{\boldsymbol{\mu}}+\mathcal{L}\hat{\boldsymbol{\sigma}}_j^2$ | $\dfrac{1}{p}\sum_{j=1}^{p}\mathcal{F}$ | $\dfrac{1}{2\,p}\sum_{j=1}^{p}\mathcal{L}$ | $\dfrac{1}{3!\,p}\sum_{j=1}^{p}-\mathcal{L}\dfrac{\tilde{\boldsymbol{\mu}}}{\hat{\boldsymbol{\sigma}}_j^2}$ | $\acute{\sigma}_j^2=\dfrac{1}{2\gamma_j^2\beta^2}$ |
| ReLU | $\mathcal{F}\tilde{\boldsymbol{\mu}}+\mathcal{L}\tilde{\boldsymbol{\sigma}}^2$ | $\mathcal{F}$ | $\dfrac{1}{2}\mathcal{L}$ | $\dfrac{1}{3!}\left(-\mathcal{L}\dfrac{\tilde{\boldsymbol{\mu}}}{\tilde{\boldsymbol{\sigma}}^2}\right)$ | $\acute{\sigma}^2=0$ |
| LeakyReLU $(\theta)$ | $\theta\tilde{\boldsymbol{\mu}}+(1-\theta)\left(\mathcal{F}\tilde{\boldsymbol{\mu}}+\mathcal{L}\tilde{\boldsymbol{\sigma}}^2\right)$ | $\theta+(1-\theta)\mathcal{F}$ | $\dfrac{1-\theta}{2}\mathcal{L}$ | $\dfrac{1-\theta}{3!}\left(-\mathcal{L}\dfrac{\tilde{\boldsymbol{\mu}}}{\tilde{\boldsymbol{\sigma}}^2}\right)$ | $\acute{\sigma}^2=0$ |
| GELU | $\mathcal{F}\tilde{\boldsymbol{\mu}}+\mathcal{L}\tilde{\boldsymbol{\sigma}}^2$ | $\mathcal{F}+\mathcal{L}\dfrac{\tilde{\boldsymbol{\mu}}}{\hat{\boldsymbol{\sigma}}^2}$ | $\dfrac{1}{2}\mathcal{L}\left(1+\dfrac{1}{\hat{\boldsymbol{\sigma}}^2}-\dfrac{\tilde{\boldsymbol{\mu}}^2}{\hat{\boldsymbol{\sigma}}^4}\right)$ | $\dfrac{1}{3!}\mathcal{L}\left(-\dfrac{\tilde{\boldsymbol{\mu}}}{\hat{\boldsymbol{\sigma}}^2}-\dfrac{3\tilde{\boldsymbol{\mu}}}{\hat{\boldsymbol{\sigma}}^4}+\dfrac{\tilde{\boldsymbol{\mu}}^3}{\hat{\boldsymbol{\sigma}}^6}\right)$ | $\acute{\sigma}^2=1$ |
| SiLU | $\dfrac{1}{p}\sum_{j=1}^{p}\mathcal{F}\tilde{\boldsymbol{\mu}}+\mathcal{L}\tilde{\boldsymbol{\sigma}}^2$ | $\dfrac{1}{p}\sum_{j=1}^{p}\mathcal{F}+\mathcal{L}\dfrac{\tilde{\boldsymbol{\mu}}\acute{\sigma}_j^2}{\hat{\boldsymbol{\sigma}}_j^2}$ | $\dfrac{1}{2\,p}\sum_{j=1}^{p}\mathcal{L}\left(1+\dfrac{\acute{\sigma}_j^2}{\hat{\boldsymbol{\sigma}}_j^2}-\dfrac{\tilde{\boldsymbol{\mu}}^2\acute{\sigma}_j^2}{\hat{\boldsymbol{\sigma}}_j^4}\right)$ | $\dfrac{1}{3!\,p}\sum_{j=1}^{p}\mathcal{L}\left(-\dfrac{\tilde{\boldsymbol{\mu}}}{\hat{\boldsymbol{\sigma}}_j^2}-\dfrac{3\tilde{\boldsymbol{\mu}}\acute{\sigma}_j^2}{\hat{\boldsymbol{\sigma}}_j^4}+\dfrac{\tilde{\boldsymbol{\mu}}^3\acute{\sigma}_j^2}{\hat{\boldsymbol{\sigma}}_j^6}\right)$ | $\acute{\sigma}_j^2=\dfrac{1}{2\gamma_j^2}$ |

where $\mathcal{L}=\dfrac{1}{\hat{\boldsymbol{\sigma}}_j}\,\varphi\left(\dfrac{\tilde{\boldsymbol{\mu}}}{\hat{\boldsymbol{\sigma}}_j}\right)$ and $\mathcal{F}=\Phi\left(\dfrac{\tilde{\boldsymbol{\mu}}}{\hat{\boldsymbol{\sigma}}_j}\right)=\dfrac{1}{2}\mathbf{erfc}\left(-\dfrac{\tilde{\boldsymbol{\mu}}}{\sqrt{2\hat{\boldsymbol{\sigma}}_j^2}}\right)$

## A.8 Emperical distributions of ResNets show Gaussianity

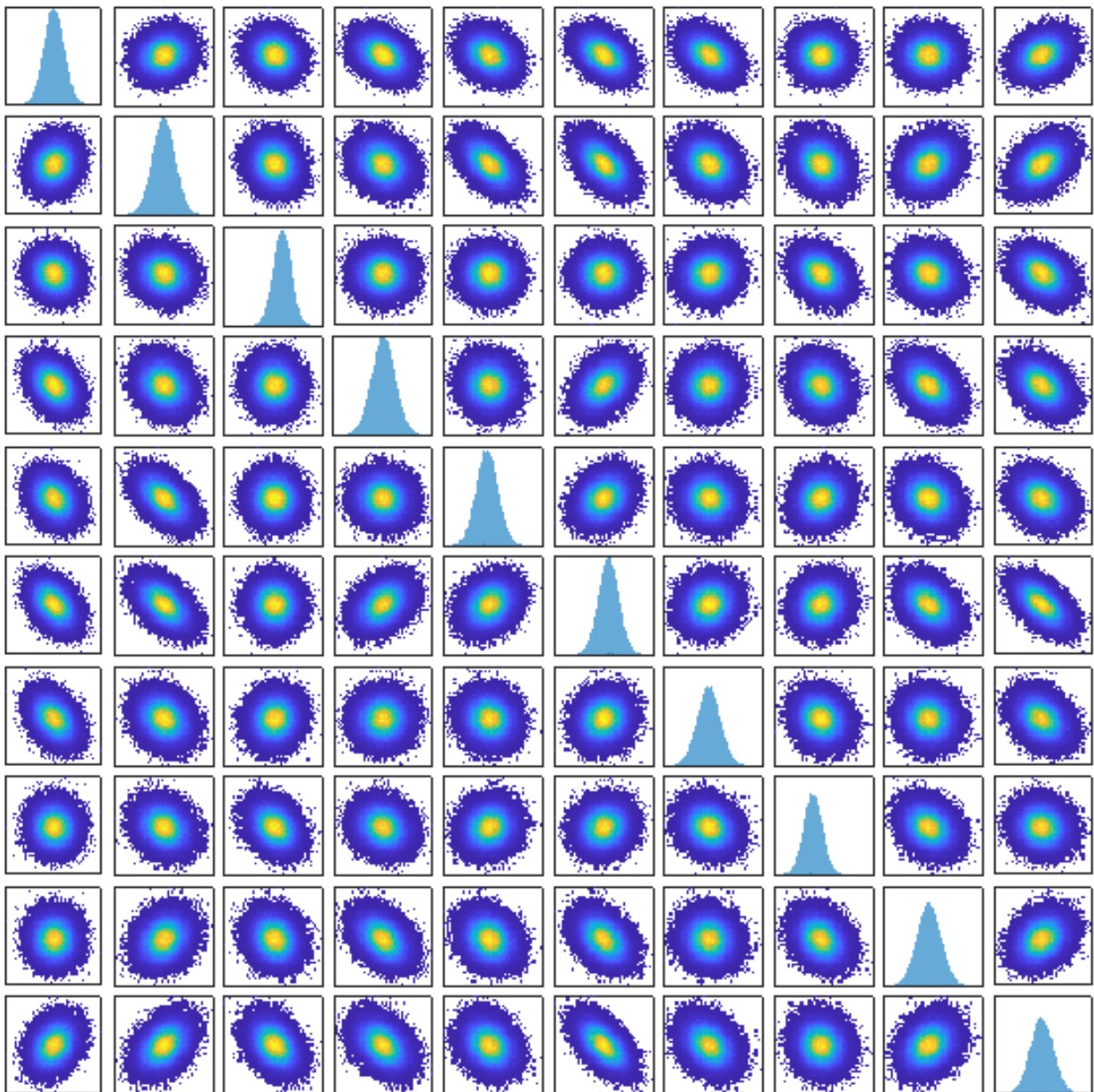

Figure 7: Empirical distributions of all units before the final softmax layer of the resnet13(ReLU).

## A.9 Approximation accuracy on other non-linearity

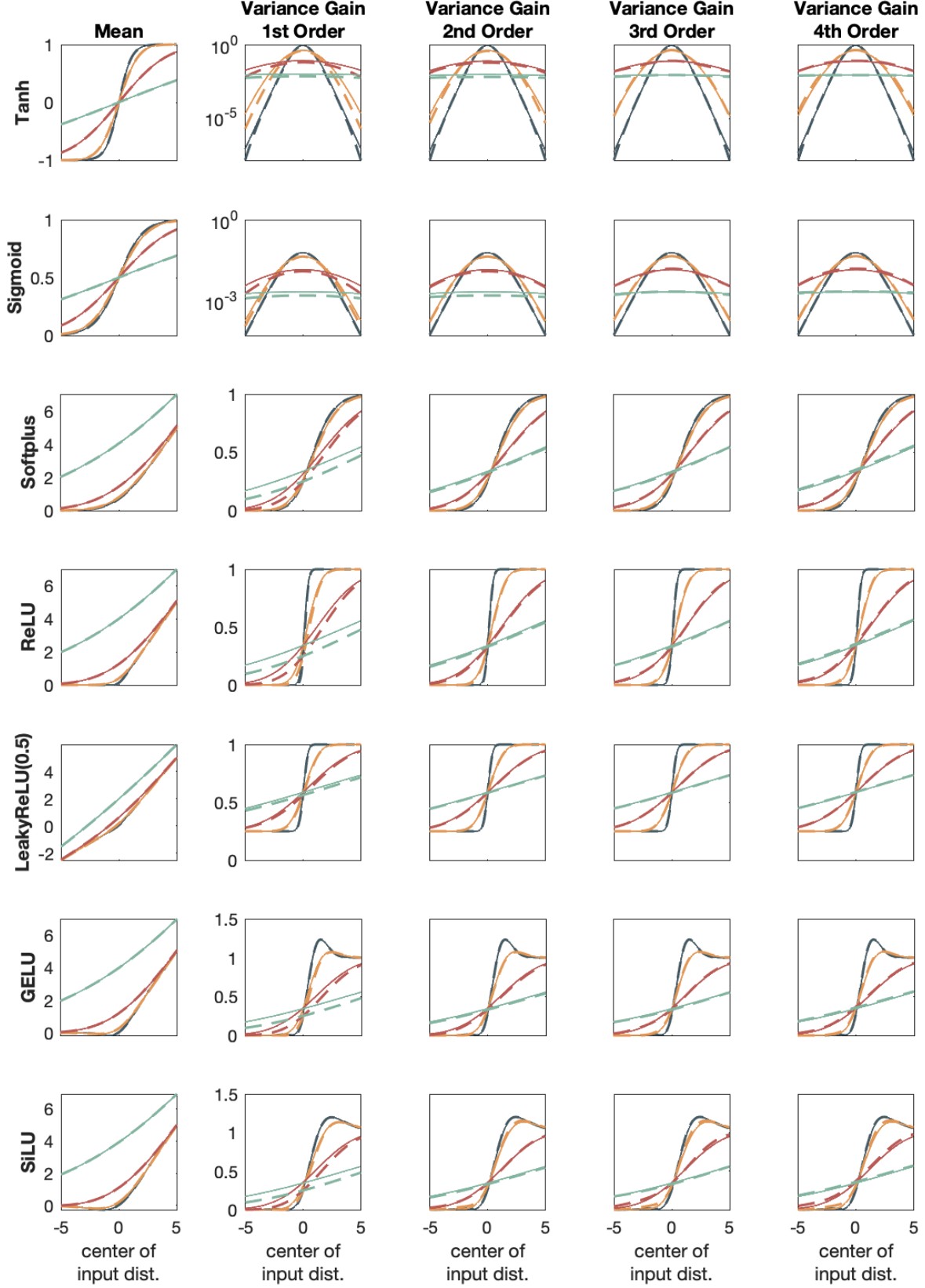

Figure 8: The meaning of different colors and line styles is the same as Fig. 1. The higher the input variance, the more significant is the benefit of using higher order Stochastic Taylor expansion.

### A.10 Pseudo Code

---

**Algorithm 1** Propagating a multi-variate Gaussian distribution through a pretrained ResNet

---

$\mu \leftarrow$ Input mean
$\Sigma \leftarrow$ Input covariance
$\mu_{res} \leftarrow$ Storage for mean of the output of residual layer
$\Sigma_{res} \leftarrow$ Storage for covariance of the output of residual layer
$\Sigma_{cross} \leftarrow$ Storage for cross-covariance between the two input of the residual layer
**for** layer in neural network **do**
  **if** layer is linear **then**
    **if** layer is addition (residual) **then**
      $\mu \leftarrow \mu + \mu_{res}$
      $\Sigma \leftarrow \Sigma + \Sigma_{res} + \Sigma_{cross} + \Sigma_{cross}^{\top}$
      empty $\mu_{res}, \Sigma_{res}, \Sigma_{cross}$
    **else**
      find effective weight $W$ and bias $b$
      $\mu \leftarrow W^{\top}\mu + b$
      $\Sigma \leftarrow W^{\top}\Sigma W$
      $\Sigma_{cross} \leftarrow W^{\top}\Sigma_{cross}$
      **if** residual connection starts from here **then**
        $\mu_{res} \leftarrow \mu$
        $\Sigma_{res}, \Sigma_{cross} \leftarrow \Sigma$
      **end if**
    **end if**
  **else**
    $\mu, \Sigma, \Sigma_{cross} \leftarrow \text{PTPE}(\text{nonlinearity}, \mu, \Sigma, \Sigma_{cross})$
    **if** residual connection starts from here **then**
      $\mu_{res} \leftarrow \mu$
      $\Sigma_{res}, \Sigma_{cross} \leftarrow \Sigma$
    **end if**
  **end if**
**end for**
$\mu_{output} \leftarrow \mu$
$\Sigma_{output} \leftarrow \Sigma$

---

## A.11 Additional results on comparing PTPE and other expectation propagation methods.

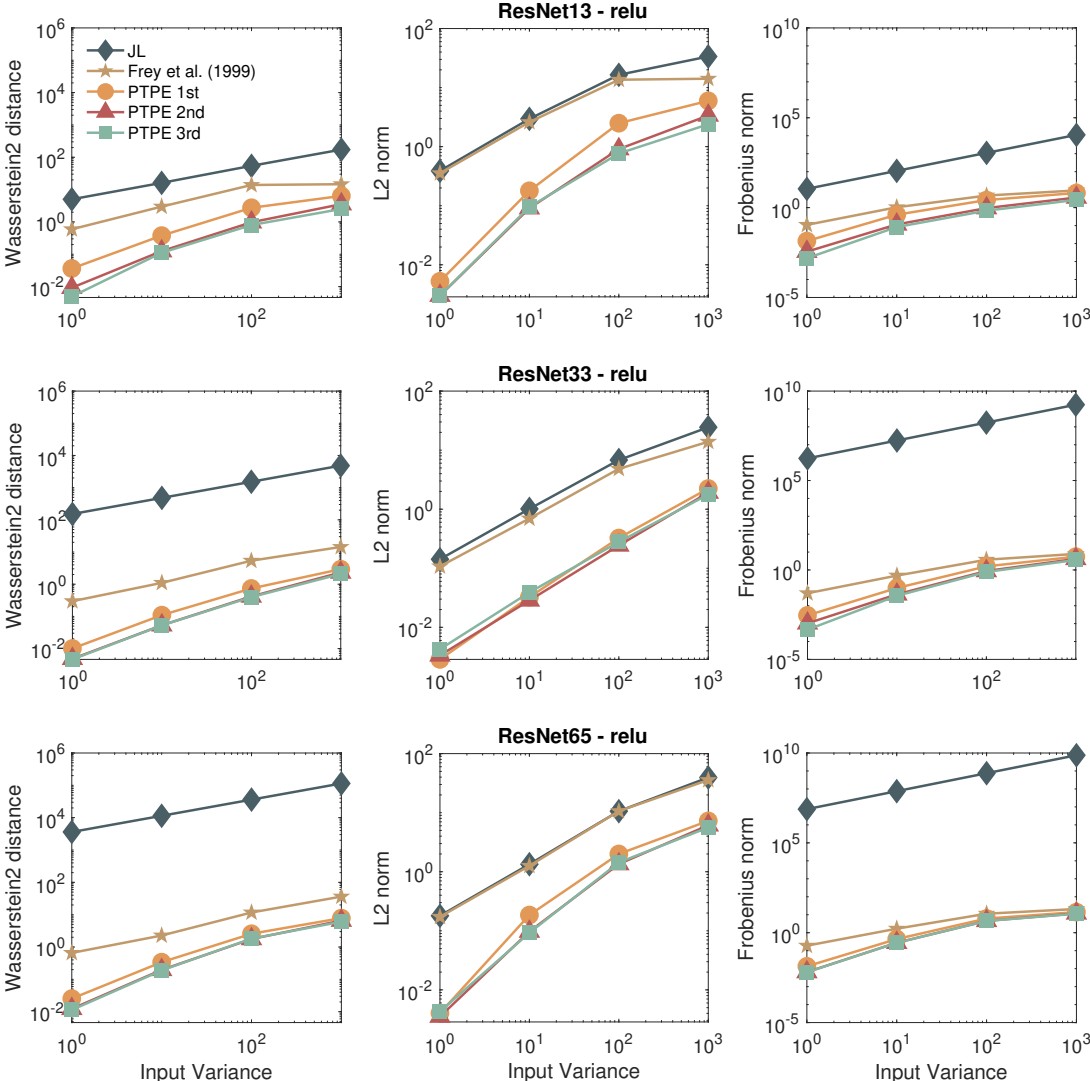

Figure 9: Similar to Figure 3, except that the models employ ReLU as the nonlinearity. The method proposed by Frey and Hinton (1999) performed poorly as it disregards correlation, a critical factor in networks with overlapping convolutional kernels and residual connections.

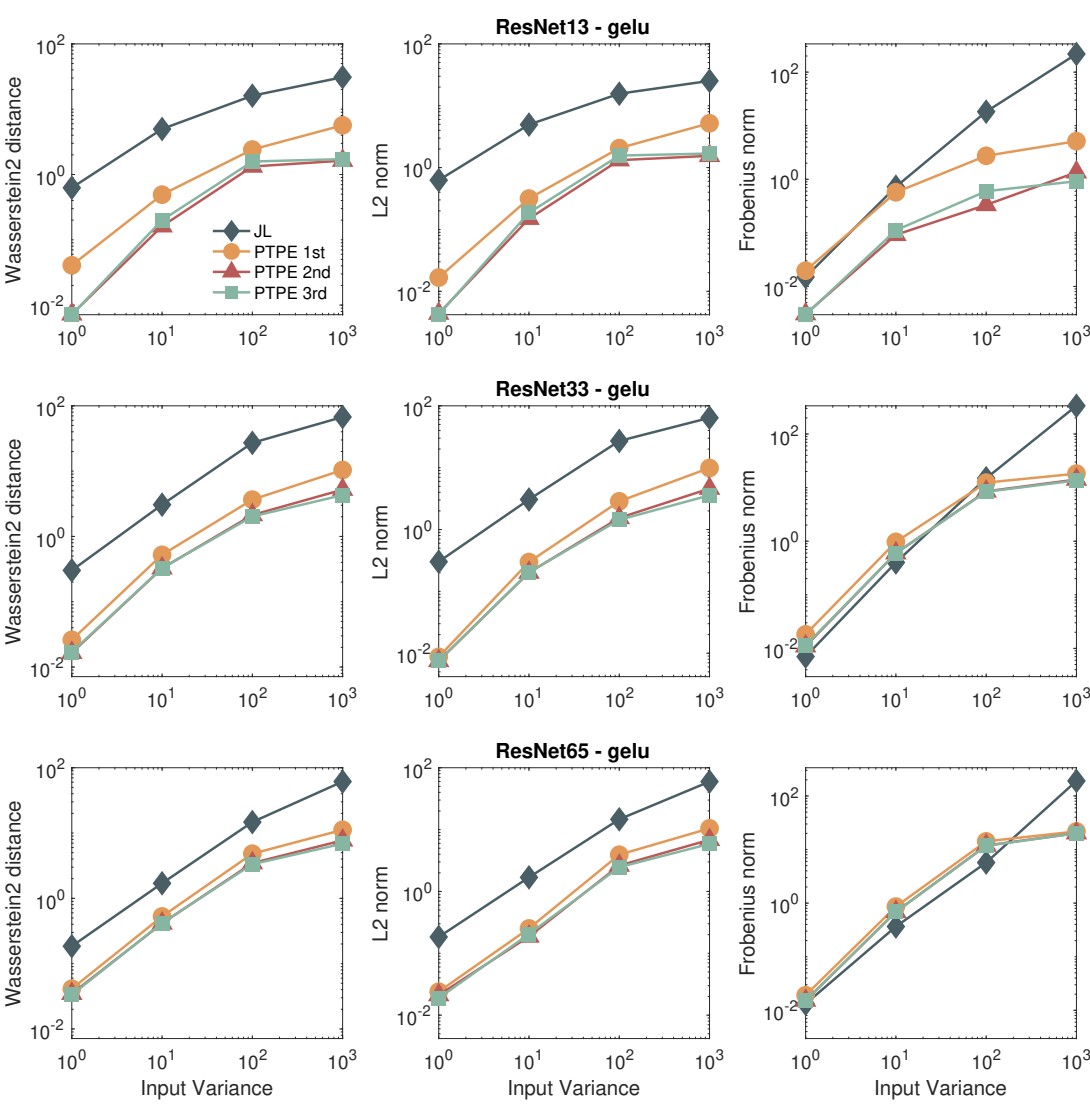

Figure 10: Similar to Figure 3, except that the models employ GELU as the nonlinearity.

**A.12 Additional results on the selection of the number of scaling factors in the reformulation of the tanh function**

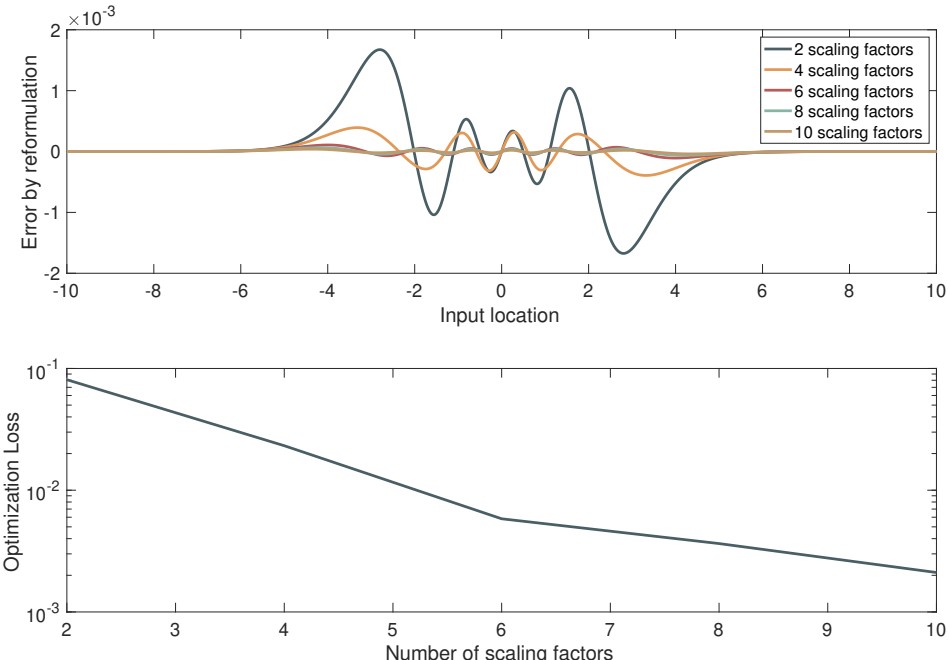

Figure 11: (Top) The error in approximating the **tanh** function using a linear combination of error functions with different scaling factors was analyzed. We selected four terms, as this configuration yields a maximum approximation error on the scale of $10^{-4}$. Importantly, increasing the number of terms enhances accuracy while only increasing computational complexity linearly, making this approach both flexible and computationally efficient. (Bottom) The optimization loss for determining the scaling factors was computed using `fmincon` in MATLAB. This loss is defined as the root mean square error (RMSE) of the approximation.

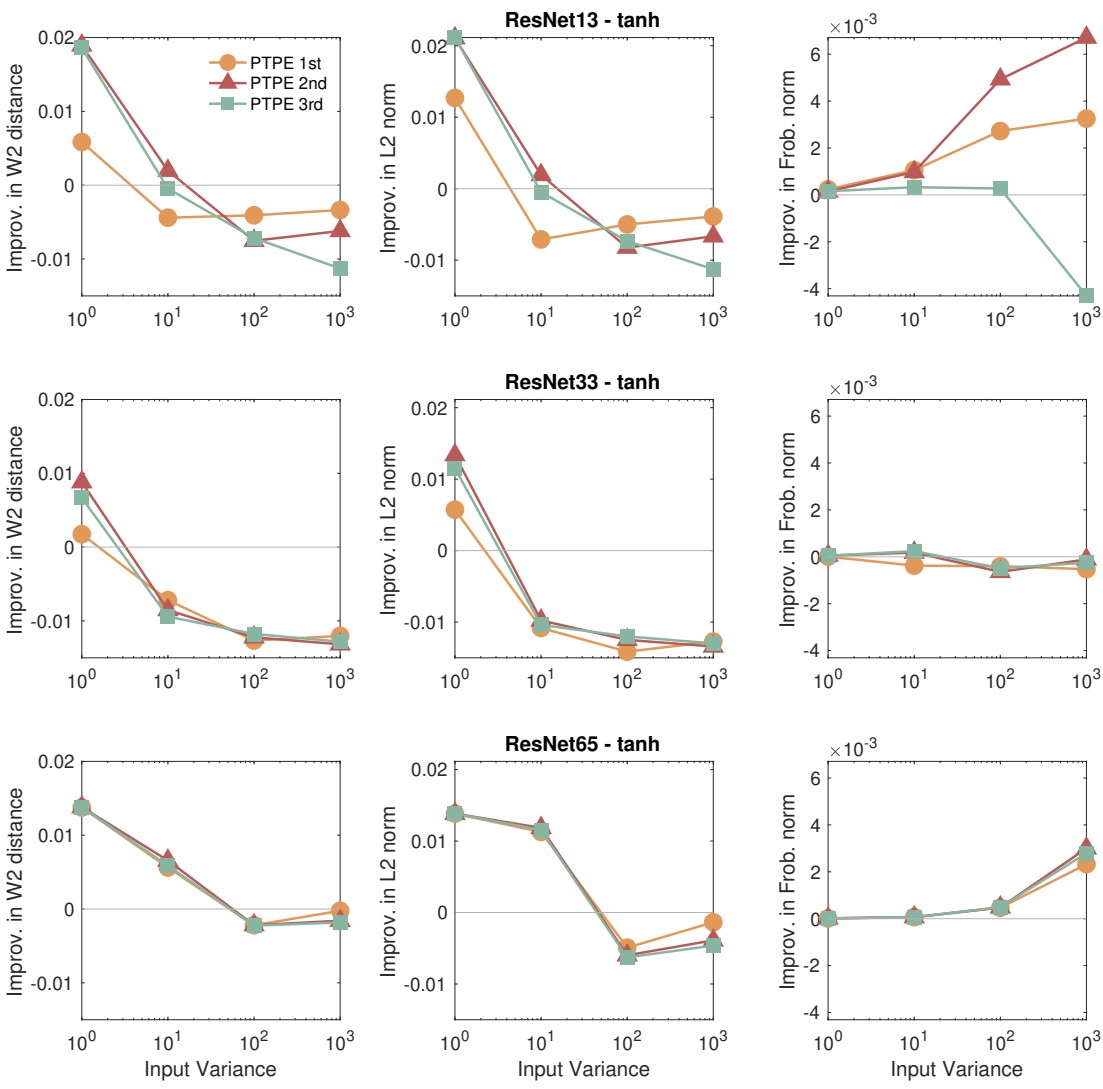

Figure 12: Improvements in the three metrics—Wasserstein-2 distance, Euclidean distance, and Frobenius norm—when using eight scaling factors $(\gamma_j)$ to approximate the **tanh** function, compared to using four, are reported. Positive values indicate improvement, while negative values denote deterioration. Notably, there is no clear dichotomy in predictive accuracy, suggesting that the approximation error of our method is not a dominant factor. Alternatively, Monte Carlo sampling with $10^6$ data points is insufficient to accurately estimate the true covariance of a 10-dimensional multivariate distribution. As a result, the reference moments may deviate from the true values. We argue This finding supports the justification for using four scaling factors rather than a larger number.

### A.13   Comparison with Gauss-Hermite Quadrature

Gauss-Hermite quadrature (GHQ) is a numerical integration method used to approximate integrals of the form:

$$\int_{-\infty}^{\infty} f(x)e^{-x^2} dx \approx \sum_{i=1}^{n} w_i f(x_i)$$

where $w_i$ are scalar weights, and $n$ is the number of quadrature points $x_i$ (also referred to as sigma points). In this context, $f(\cdot)$ can represent a nonlinearity or its square, corresponding to the mean and covariance integrals. While infinitely many linear combinations of $f(x)$ can approximate this integral, GHQ selects quadrature points and weights based on Hermite polynomials. These polynomials are orthogonal, making GHQ computationally efficient. This method also serves as the foundation of the cubature Kalman filter (CKF).

We compare the predictive accuracy of PTPE and GHQ across the first four orders (Fig. 13 to 16), using absolute error as the evaluation metric against a reference mean and variance obtained from $10^7$ Monte Carlo samples. Since Monte Carlo estimates with $10^7$ samples typically fluctuate on the order of $10^{-4}$, absolute errors of magnitude $\leq 10^{-4}$ are considered negligible.

PTPE outperforms GHQ in the first three orders. At the fourth order, PTPE provides evidently more accurate estimates than GHQ when the input variance is 1 or higher. Notably, 4th-order PTPE surpasses 4th-order GHQ on ReLU and LeakyReLU even at a variance of 0.1.

To determine how many GHQ orders are needed to surpass 4th-order PTPE, we compare 10th-, 20th-, 30th-, and 40th-order GHQ to 4th-order PTPE (Figures 17 to 20). There is no clear threshold at which GHQ consistently outperforms 4th-order PTPE, as performance depends on the type of nonlinearity. Among the seven types, PTPE is particularly effective for ReLU and LeakyReLU. However, we observe a general trend: higher-order GHQ is required to match PTPE, particularly at higher input variances.

In summary, although GHQ efficiently selects sampling points, it requires an increasing number of points for accurate estimation at higher input variances. More critically, the number of GHQ sampling points grows rapidly with dimensionality, making high-dimensional integration computationally expensive. In contrast, PTPE's complexity is at most $\mathcal{O}(n^2)$, making it more scalable. However, GHQ is expected to outperform PTPE in narrow but deep neural networks. Unlike PTPE, GHQ does not require layer-wise propagation of moments. Furthermore, in narrow networks, the Gaussianity assumption of hidden layers often breaks down, making it difficult for PTPE to provide accurate estimates.

Input Variance 0.1

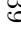

Figure 13: Comparing the absolute approximation error of PTPE and GHQ across the first four orders. The odd number of columns show the error of mean, and even number of columns show the error of variance, the smaller the better. In this panel, all input variance is 0.1.

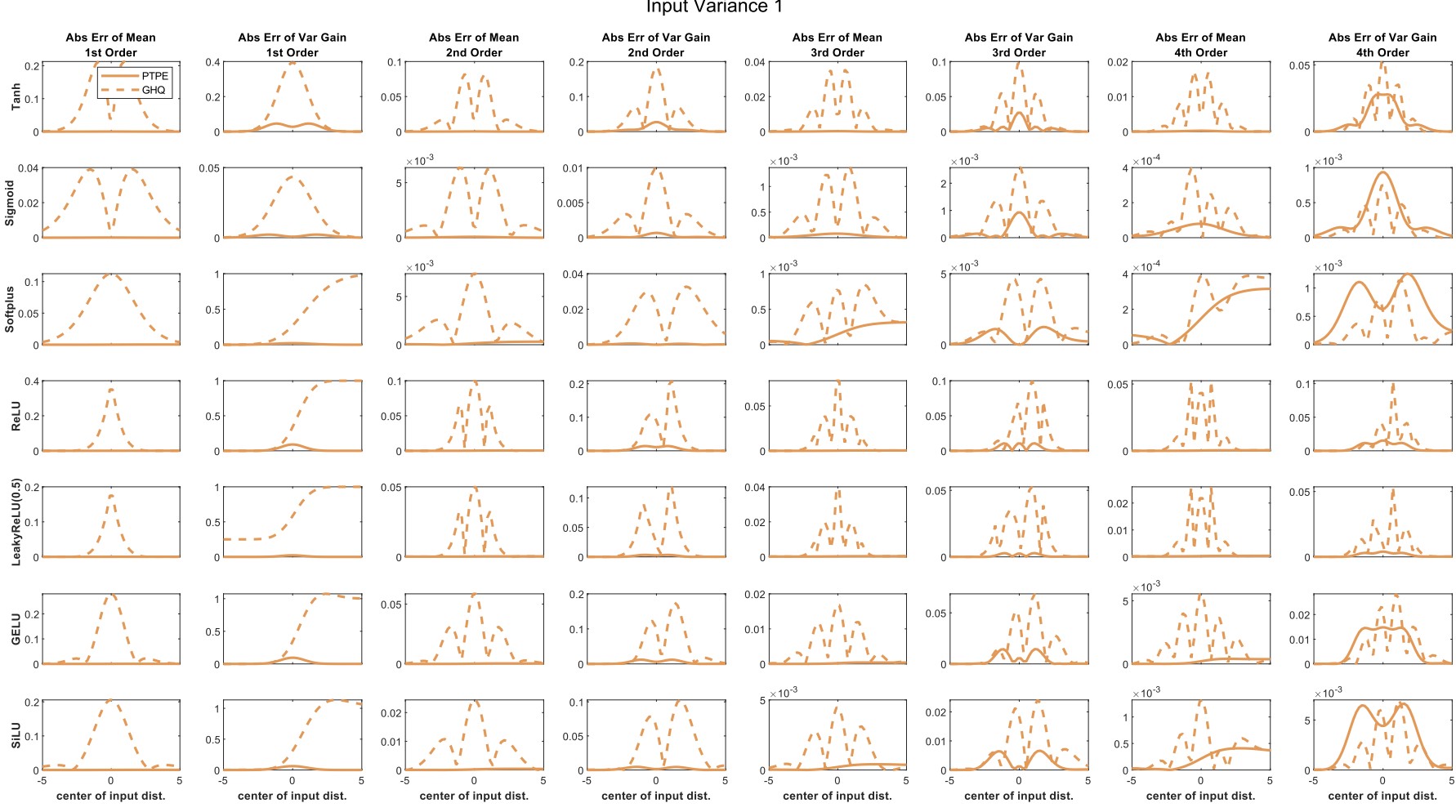

Figure 14: Comparing the absolute approximation error of PTPE and GHQ across the first four orders. The odd number of columns show the error of mean, and even number of columns show the error of variance, the smaller the better. In this panel, all input variance is 1.

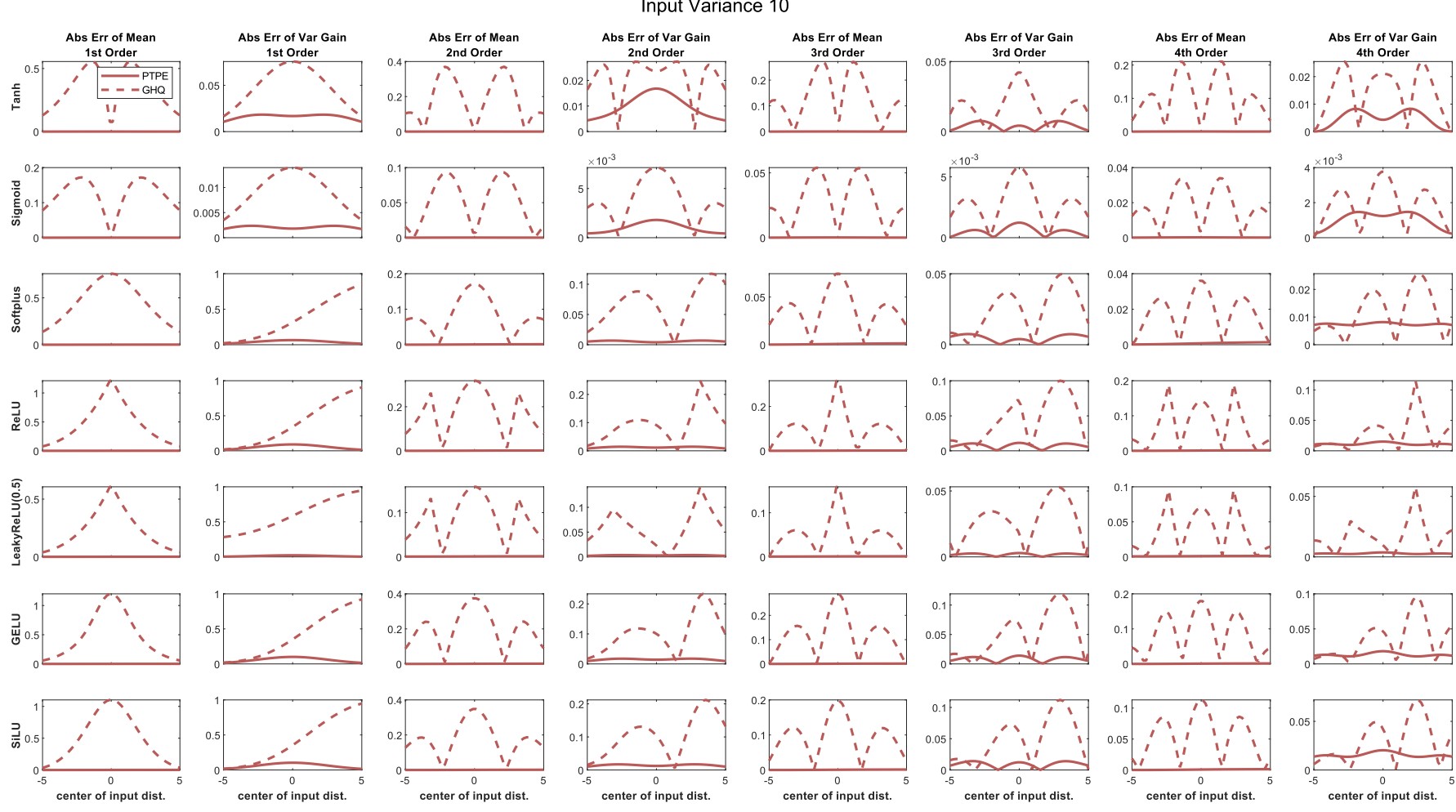

Figure 15: Comparing the absolute approximation error of PTPE and GHQ across the first four orders. The odd number of columns show the error of mean, and even number of columns show the error of variance, the smaller the better. In this panel, all input variance is 10.

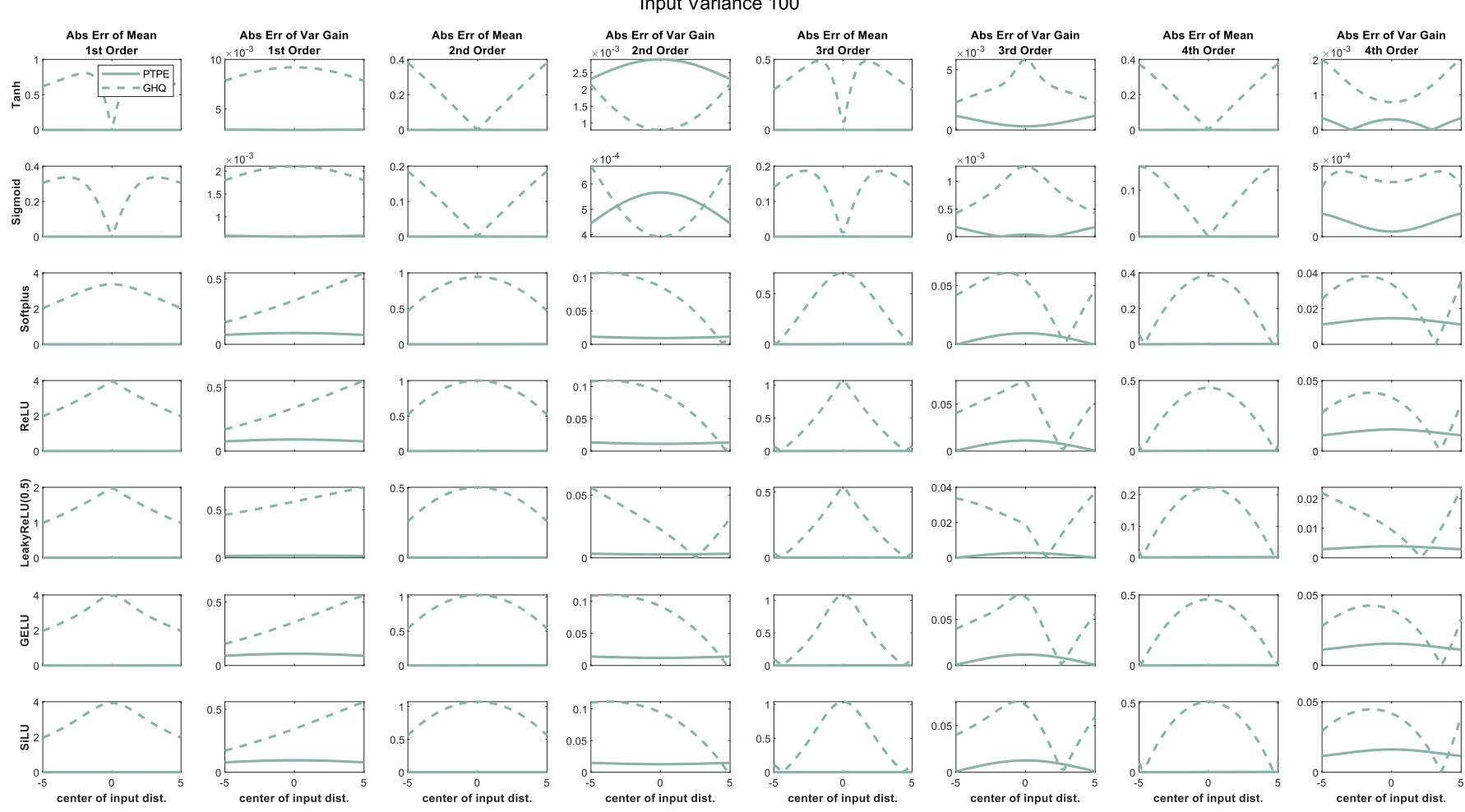

Figure 16: Comparing the absolute approximation error of PTPE and GHQ across the first four orders. The odd number of columns show the error of mean, and even number of columns show the error of variance, the smaller the better. In this panel, all input variance is 100.

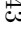

Figure 17: Comparing the absolute approximation error of 4th-order PTPE and 10th-, 20th-, 30th-, and 40th-order GHQ. The odd number of columns show the error of mean, and even number of columns show the error of variance, the smaller the better. In this panel, all input variance is 0.1.

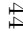

Figure 18: Comparing the absolute approximation error of 4th-order PTPE and 10th-, 20th-, 30th-, and 40th-order GHQ. The odd number of columns show the error of mean, and even number of columns show the error of variance, the smaller the better. In this panel, all input variance is 1.

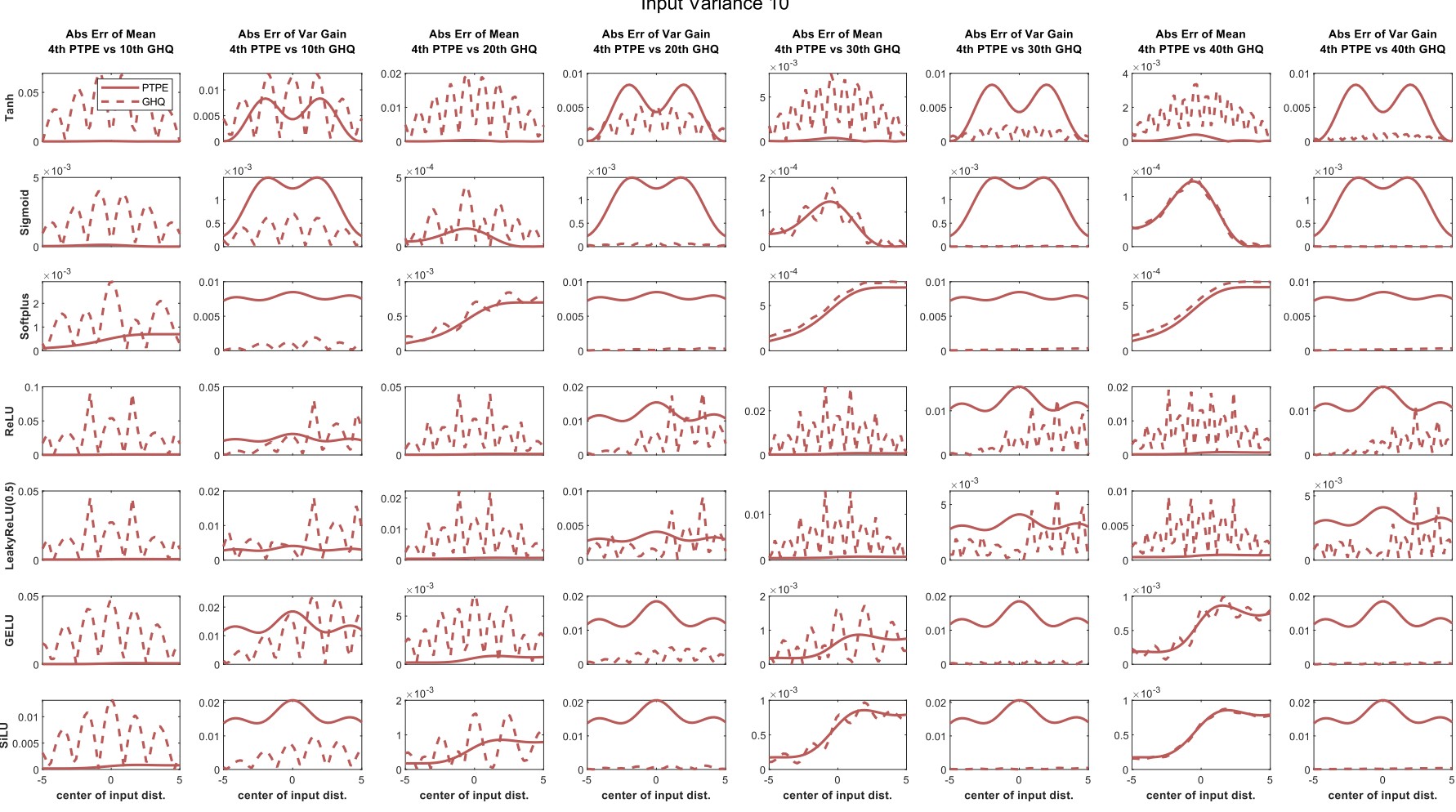

Figure 19: Comparing the absolute approximation error of 4th-order PTPE and 10th-, 20th-, 30th-, and 40th-order GHQ. The odd number of columns show the error of mean, and even number of columns show the error of variance, the smaller the better. In this panel, all input variance is 10.

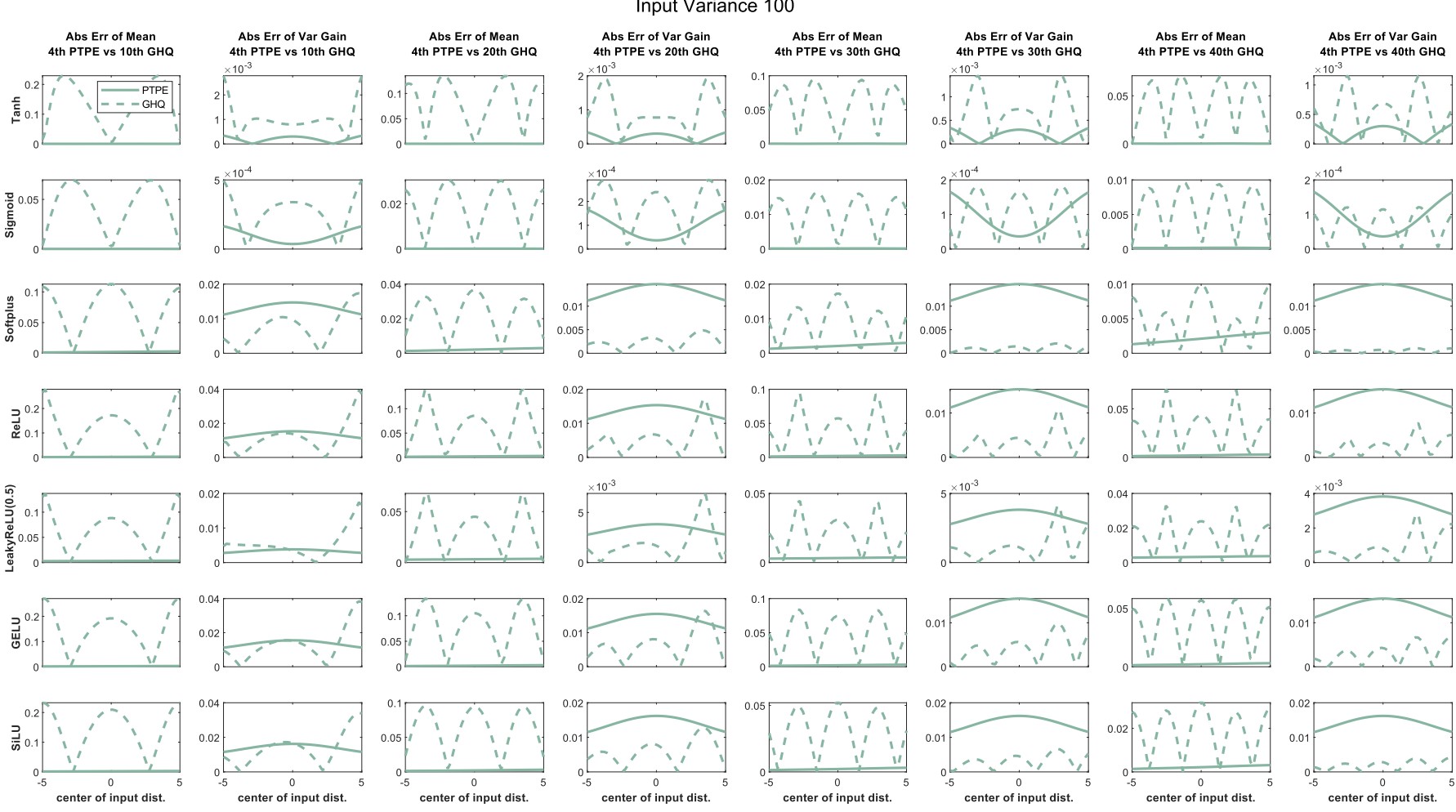

Figure 20: Comparing the absolute approximation error of 4th-order PTPE and 10th-, 20th-, 30th-, and 40th-order GHQ. The odd number of columns show the error of mean, and even number of columns show the error of variance, the smaller the better. In this panel, all input variance is 100.

