# OpenReview forum: "A Stochastic Polynomial Expansion for Uncertainty Propagation through Networks"
_TMLR — Accepted by TMLR_

### Review · Reviewer_WbCR · 2025-03-07

**Summary Of Contributions:**

This work addresses the uncertainty in neural network outputs by examining the propagation of Gaussian inputs through non-linear activation functions. The authors derive Taylor expansions for seven widely-used activation functions (Tanh, Sigmoid, Softplus, ReLU, LeakyReLU, GELU, and SiLU) and present experiments that demonstrate the effectiveness of higher order terms (up to third-order) in these expansions.

**Audience:**

Yes

**Claims And Evidence:**

Yes

**Requested Changes:**

__Major.__

- In CIFAR-10 experiments, is there any specific reason for choosing ResNet depths of 13, 33, and 65? The standard CIFAR-10 settings, originally proposed by He et al. (2016) and widely followed, are 20, 32, 44, and 56. Deviating without a clear reason may make comparisons with existing and future work more difficult, so I recommend following the common convention if possible.

- Section 3.2 demonstrates how well the proposed PTPE method predicts the mean and variance of the output. It would be helpful to also clarify how the method performs with respect to common uncertainty estimation metrics in the machine learning literature, such as negative log-likelihood, Brier score, and expected calibration error, when evaluating ResNet on CIFAR-10. The current results do not provide any indication of whether the trained neural network is practically useful, that is, how well it performs in terms of classification accuracy and how well-calibrated its prediction is.

- In the UCI experiments, was a fair comparison made with the baseline methods? Tables 2 and 3 show results for PTPE using three activations: ReLU, GELU, and Tanh. The choice of activation function should also be considered for the baselines. Therefore, for a fair comparison, comparative results between PTPE and the baseline methods should be provided for each activation function. However, it appears that the reported baseline values were taken directly from the original papers without considering such experimental factors.

__Minor.__

- On page 4, there is an unknown reference; “the number of scaling factors is discussed in ??.”

- On page 5, there is a formatting issue; “the reformulation oftanhand sigmoid.”

- It would be helpful to include the meaning of each column in Figure 3, 8, and 9 either in the subplots or in the captions; while this information is provided in the main text of Section 3.2, adding it would improve readability.

- I could not find the code at the provided anonymous GitHub link in the paper; only the DVI+PTPE, VAE+PTPE folders and LICENSE file are visible.

**Strengths And Weaknesses:**

__Strenghts.__

- This work provides comprehensive derivations for a range of activation functions, including GELU and Swish, which are commonly used in modern architectures.

- The aim of ensuring tractability through polynomial approximations is well-motivated.

__Weaknesses.__

- There seems to be insufficient discussion on the errors introduced by the proposed approximations. For instance, activations like tanh, sigmoid, and SiLU are first approximated using erf, and then the resulting approximations are further approximated using polynomial expansions. Isn’t there any concern with this two-step approximation process?

- As there is no theoretical analysis regarding how accurate the proposed polynomial expansions, experimental validation is necessary. However, the current experimental results appear to be insufficient; more details are in the Requested Changes.

- Currently, the contributions of this work appear to be relatively modest. However, the final sentence of the conclusion outlines a clear direction: "It can potentially be effectively applied in various domains, including adversarial training, Bayesian inference, generative models, and safety-critical applications, offering a versatile tool for enhancing the reliability and robustness of neural networks." These ideas should not be left merely as future work; for the paper to be more substantial, it should either address them comprehensively or delve deeply into one of them.

---

> ### Author Response · Authors · 2025-04-09
>
> > This work addresses the uncertainty in neural network outputs by examining the propagation of Gaussian inputs through non-linear activation functions. The authors derive Taylor expansions for seven widely-used activation functions (Tanh, Sigmoid, Softplus, ReLU, LeakyReLU, GELU, and SiLU) and present experiments that demonstrate the effectiveness of higher order terms (up to third-order) in these expansions.
>
> We sincerely appreciate the reviewer’s time and effort in evaluating our work and summarizing its contributions. We are pleased that the reviewer recognizes the importance of analyzing uncertainty propagation in neural networks. However, we would like to clarify a key aspect of our contribution, as the current summary does not fully capture the novelty of our approach.
>
> While PTPE draws inspiration from Taylor expansions, it is a fundamentally new method rather than a standard Taylor expansion. Traditional Taylor expansions operate in deterministic settings, whereas PTPE is specifically designed to handle uncertainty propagation. The derivation of PTPE is not that trivial.
>
> To elaborate, if the input $X$ of a function $f$ carries uncertainty, then the derivatives of $f(X)$ also inherit uncertainty. Attempting to compute the covariance of $f(X)$ using a conventional Taylor expansion leads to an integral of the form
>
> $$
> \int f^2(x) p(x) dx = \int_{(x,a)} \left[ f(a) + \sum \frac{f^{(i)} (a)}{i!} (x-a)^i \right]^2 p(x,a) d(x,a)
> $$
> which quickly becomes intractable.
>
> This work provides an alternative solution to a core problem in approximate inference [1]: finding an accurate approximating distribution (e.g., Gaussian) for a posterior distribution of unknown form and using it to efficiently compute integrals in downstream tasks. PTPE achieves this by reformulating the propagation of uncertainty through nonlinear activation functions in a computationally efficient manner, making it a practical tool for a wide range of applications.
>
> [1] Li, Yingzhen. "Topics in Approximate Inference." 2020. Retrieved from https://yingzhenli.net/home/pdf/topics_approx_infer.pdf
>
> > There seems to be insufficient discussion on the errors introduced by the proposed approximations. For instance, activations like tanh, sigmoid, and SiLU are first approximated using erf, and then the resulting approximations are further approximated using polynomial expansions. Isn’t there any concern with this two-step approximation process?
>
> We appreciate the reviewer’s concern. However, the accuracy of this two-step approximation has been thoroughly validated in Figure 1 and Figure 7 (Appendix), where we compare it against extensive Monte Carlo simulations. The results demonstrate that even with this two-step approximation, the accuracy remains high for a wide range of input mean and variance. Notably, this high accuracy is achieved using only a third-order polynomial expansion, whereas Gauss-Hermite quadrature —- a well-established alternative —- requires more terms to achieve comparable accuracy (see Section A.13).
>
> If further reduction of the approximation error is desired, the number of scaling factors can be increased. With six scaling factors, the maximum absolute error is already on the order of 1e-4, and it decreases further with additional scaling factors. Moreover, since the computational complexity scales linearly with the number of scaling factors, this approach remains efficient and is well-suited for parallel computation (as discussed in Section A.12).
>
> Additionally, we would like to highlight that using the error function to approximate the sigmoid function has a long history in the literature. The earliest reference we could find is D. J. Spiegelhalter and S. L. Lauritzen (1990). Our approach improves upon this by employing a linear combination for better accuracy.

---

> > ### Author Response · Authors · 2025-04-09
> >
> > > Currently, the contributions of this work appear to be relatively modest. However, the final sentence of the conclusion outlines a clear direction: "It can potentially be effectively applied in various domains, including adversarial training, Bayesian inference, generative models, and safety-critical applications, offering a versatile tool for enhancing the reliability and robustness of neural networks." These ideas should not be left merely as future work; for the paper to be more substantial, it should either address them comprehensively or delve deeply into one of them.
> >
> > We respectfully disagree with the reviewer’s characterization of our contributions as modest. Our work addresses a fundamental challenge in approximate inference by introducing a novel method for estimating the moments of nonlinearly transformed Gaussian distributions. Existing moment-matching approaches can only be derived for piecewise linear functions such as ReLU and step functions, leaving a significant gap in handling more general nonlinearities. Our method fills this gap, providing a solution where none previously existed.
> >
> > To substantiate our contributions, we conducted three diverse experiments demonstrating PTPE's generality and effectiveness across different settings. In response to another reviewer, we added OOD detection experiments. The results are summarized in the figure rotation_mnist_ood/results/uncertainty_estimation_results.png, available in our repository: https://anonymous.4open.science/r/Stochastic_Polynomial_Expansion-0BEB. These results illustrate not only the robustness of our approach but also its potential impact on a wide range of applications. Given these contributions, we believe our work represents a meaningful advancement and provides valuable tools for the research community.
> >
> > Regarding the discussion section, we appreciate the reviewer’s suggestion to expand on the mentioned applications. However, we believe that outlining future directions serves an important purpose—encouraging further exploration and adoption of our method in domains beyond those explicitly demonstrated. The propagation of uncertainty through nonlinearities is a ubiquitous challenge, and we highlighted adversarial training and safety-critical applications as promising areas where our method could have a significant impact. Additionally, our approach’s applicability to Bayesian inference and generative models has already been demonstrated in Sections 3.3 and 3.4.
> >
> > We hope this clarification conveys both the significance of our contributions and the rationale behind our discussion.

---

> > > ### Author Response · Authors · 2025-04-09
> > >
> > > > In CIFAR-10 experiments, is there any specific reason for choosing ResNet depths of 13, 33, and 65? The standard CIFAR-10 settings, originally proposed by He et al. (2016) and widely followed, are 20, 32, 44, and 56. Deviating without a clear reason may make comparisons with existing and future work more difficult, so I recommend following the common convention if possible.
> > > > Section 3.2 demonstrates how well the proposed PTPE method predicts the mean and variance of the output. It would be helpful to also clarify how the method performs with respect to common uncertainty estimation metrics in the machine learning literature, such as negative log-likelihood, Brier score, and expected calibration error, when evaluating ResNet on CIFAR-10. The current results do not provide any indication of whether the trained neural network is practically useful, that is, how well it performs in terms of classification accuracy and how well-calibrated its prediction is.
> > >
> > > We appreciate the reviewer’s suggestion. However, we believe there may be a misunderstanding. We would like to clarify that this experiment involved deterministic networks, not BNNs trained with PTPE or Jacobian linearization. The objective of this experiment was to evaluate PTPE's ability to approximate moments in the multivariate setting, where estimating the covariance structure is nontrivial. This complements Figure 1, which focuses on the univariate case, where covariance and cross-covariance terms are not present. Additionally, this experiment highlights the importance of accounting for cross-covariance—validating Equation 3—which is often overlooked in prior work.
> > >
> > > Regarding the choice of ResNet depths (13, 33, and 65), we believe this question partially stems from the same misunderstanding about the deterministic nature of our experiment. We were not specifically following the conventions set by He et al., as our focus was not on benchmarking but on analyzing PTPE’s performance across different network depths. While we recognize that certain ResNet depths are commonly used in prior work, we were unaware of any strict standard dictating their selection. Given the wide variety of ResNet-based architectures beyond the canonical depths, we view network depth as a design choice rather than a fixed convention —- much like the flexibility in choosing MLP sizes.
> > >
> > > The key reason for varying the depth in our experiment is to address a potential concern that approximation errors might accumulate layer by layer, ultimately becoming uncontrollable. Our results demonstrate that the accuracy of PTPE remains robust even in deep architectures, alleviating such concerns.
> > >
> > > Furthermore, our findings highlight the limitations of the commonly used first-order Taylor expansion, which is often insufficient for capturing nonlinear transformations in deep networks. We believe that many related works relying on first-order Taylor approximations could benefit from incorporating PTPE.

---

> > > > ### Author Response · Authors · 2025-04-09
> > > >
> > > > > In the UCI experiments, was a fair comparison made with the baseline methods? Tables 2 and 3 show results for PTPE using three activations: ReLU, GELU, and Tanh. The choice of activation function should also be considered for the baselines. Therefore, for a fair comparison, comparative results between PTPE and the baseline methods should be provided for each activation function. However, it appears that the reported baseline values were taken directly from the original papers without considering such experimental factors.
> > > >
> > > > We appreciate the reviewer’s concern about ensuring a fair comparison in the UCI regression experiments. However, the baseline models were evaluated using their standard implementations, as reported in prior work. This ensures that our comparisons are consistent with existing literature and avoids introducing additional confounding factors. Reimplementing these baselines with different activation functions would require additional tuning and hyperparameter adjustments, making it difficult to isolate the effect of PTPE itself.
> > > >
> > > > Our goal in this experiment is to demonstrate that PTPE enables accurate moment estimation across different activation functions, including non-piecewise-linear ones, in a scalable way—something that has not been achieved in prior work. To achieve this, we implemented PTPE within DVI, a strong and scalable Bayesian learning method that has already been shown to outperform the other baselines listed in the tables under the standard ReLU activation. Since DVI is already a superior approach, demonstrating PTPE’s effectiveness within DVI suffices to show that our method works in a state-of-the-art setting. Reimplementing older baseline methods with different activation functions would not provide additional insights, as our study is not focused on benchmarking every possible activation choice but rather on demonstrating PTPE’s generality and effectiveness in Bayesian learning.
> > > >
> > > > For a fair comparison, we have ensured that all models are evaluated under the same conditions using their established configurations. However, if the concern is that comparing MLPs with GELU and Tanh to MLPs with ReLU is unfair, we direct the reviewer’s attention to the results of PTPE with ReLU, which remain competitive against the baseline methods.
> > > >
> > > > Thus, we believe our evaluation is both fair and in line with standard practice in the field.
> > > >
> > > > > On page 4, there is an unknown reference; “the number of scaling factors is discussed in ??.”
> > > >
> > > > We appreciate the reviewer’s attention to detail and apologize for the oversight. The reference was intended to point to Section A.12 in the Appendix. We have corrected this in the revised manuscript.
> > > >
> > > > > On page 5, there is a formatting issue; “the reformulation oftanhand sigmoid.”
> > > >
> > > > Thank you for pointing this out. We have corrected the formatting issue in the revised manuscript and will ensure it is properly reflected in the camera-ready version.
> > > >
> > > > > It would be helpful to include the meaning of each column in Figure 3, 8, and 9 either in the subplots or in the captions; while this information is provided in the main text of Section 3.2, adding it would improve readability.
> > > >
> > > > That's a thoughtful comment. We will add the following sentence to the caption: The three columns correspond to W2 distance, Euclidean distance, and Frobenius norm, respectively, while the three rows correspond to different network depths.
> > > >
> > > > > I could not find the code at the provided anonymous GitHub link in the paper; only the DVI+PTPE, VAE+PTPE folders and LICENSE file are visible.
> > > >
> > > > Thank you for bringing this to our attention. The full code is included in the repository, but due to the anonymous GitHub's interface, only a subset of files is visible in the browser. To access all files, please click on "Download Repository" in the top-right corner of the page.

---

### Review · Reviewer_VBVu · 2025-03-10

**Summary Of Contributions:**

This paper proposed a novel method for propagating uncertainty from the input of a neural network to its output, based on a "stochastic polynomial expansion" of the nonlinearities in the network. The authors provide closed-form expansions for the first four polynomial coefficients for seven commonly used activation functions and provide a general methodology for computing additional coefficients for general non-linearities. The authors provide experimental evidence for the accuracy of their approximations based on the UCI regression datasets, CIFAR-10 image classification, and MNIST digit generation with a VAE.

**Audience:**

Yes

**Claims And Evidence:**

Yes

**Requested Changes:**

I reviewed a previous version of the paper and I am pleased to see that many of my concerns have been addressed to some extent.  However, as mentioned above I still feel that the experiments in this paper do not reflect realistic settings. Thus, I have several recommendations to strengthen the paper. In my opinion, this is currently a borderline submission, so while none of the following suggestions are critical on their own, making even a few of these changes would be enough for me to recommend acceptance.

1.[strengthen] Add a wider range of (more realistic) uncertainty quantification benchmarks:
    * E.g., rotated-MNIST, corrupted-CIFAR, and OOD detection tasks (see "Can you trust your model's uncertainty? Evaluating predictive uncertainty under dataset shift" by Ovadia et al.).
    * In particular, provide classification accuracy for the CIFAR-100 experiment, and add some "more realistic" corruptions from Hendrycks and Dietterich (2019).
2. [strengthen] Add deep ensembles as a baseline in tables 2 and 3.
3. [strengthen] Add an ablation for comparing the number of approximating terms (*and* MC estimation with different numbers of samples) based on a realistic uncertainty quantification benchmark (rather than the toy Gaussian setting shown in Fig 11.)

**Strengths And Weaknesses:**

## Strengths
* The paper tackles an interesting and challenging problem.
* The paper provides a novel solution to this problem and approximated moments for several commonly used activation functions. These moments could be useful outside of this line of work.

## Weaknesses
* The experimental validation is skewed towards toy settings that don't necessarily reflect the potential real-world usage of the method. For example, the comparisons to existing uncertainty qualification methods are primarily based on UCI regression datasets rather than large-scale image classification tasks, and the Gaussian noise applied to the CIFAR datasets is only one of several corruptions suggested by  Hendrycks and Dietterich (2019).
* Deep ensembles, which are a key uncertainty quantification baseline, have not been included.

---

> ### Author Response · Authors · 2025-04-09
>
> We deeply regret that we missed the previous revision deadline and fully acknowledge that our delay was unprofessional. We were in the process of addressing the reviewer comments, particularly those regarding the experiments on real-work datasets. While attempting to reproduce some results from the literature, we encountered unexpected challenges that required additional time to resolve. Please accept our sincere apologies for any inconvenience this has caused.
>
> ---
>
>
> We thank the reviewer for the helpful feedback and understand that the reviewer's main concern pertains to the scale of the experiments.
>
> It is worth noting that many seminal papers in approximate inference have primarily used UCI regression benchmarks as their main experiments. Regression is a task of substantial practical relevance, with applications in medicine, economics, geophysics, and other domains where data is often scarce or expensive to collect. For this reason, we do not consider such benchmarks to be “toy” problems, but rather meaningful and legitimate testbeds for evaluating the performance of inference methods. Our results on these tasks demonstrate that PTPE enables exact Bayesian inference in settings where traditional methods remain limited.
>
> Despite that, we acknowledge the challenges of applying exact Bayesian inference in more complex neural architectures. Two primary obstacles make this especially difficult.
>
> First, non-differentiable nonlinearities—such as max-pooling—pose substantial barriers to moment propagation.
>
> Second, 2D convolutional layers present significant computational challenges. Calculating the exact covariance of the output of a convolution layer is challenging because each output element depends on heavily overlapping (and thus correlated) input patches. This overlap results in a high-dimensional (Height x Width x Channel by Height x Width x Channel), block-structured covariance matrix that is computationally expensive to construct and manipulate. Without low-level optimization, constructing and updating this matrix during training is computationally prohibitive.
>
> Due to these limitations, we have not included experiments on datasets such as corrupted CIFAR. We agree that methods like dropout and deep ensembles are currently more scalable and effective in such contexts. Nevertheless, we believe our results are a valuable step toward extending exact Bayesian methods to broader domains, and we hope they will motivate further development and adaptation of PTPE.
>
> Finally, we would like to emphasize that TMLR prioritizes technical correctness over subjective notions of significance, thereby fostering scientific discourse on ideas that may not yet be mainstream but could prove influential in the future. While our current method is not yet applicable to vision-based uncertainty estimation, the fact that no reviewer has questioned the correctness of our derivations supports the soundness of our contributions and their appropriateness for publication.
>
>
> > Add a wider range of (more realistic) uncertainty quantification benchmarks: * E.g., rotated-MNIST, corrupted-CIFAR, and OOD detection tasks (see "Can you trust your model's uncertainty? Evaluating predictive uncertainty under dataset shift" by Ovadia et al.).
>
> In response to this suggestion, we have added experiments on MNIST categorization, rotated-MNIST, and out-of-distribution (OOD) detection using Fashion MNIST, following the setup in Figure 1 of [1]. The results are summarized in the figure rotation_mnist_ood/results/uncertainty_estimation_results.png, available in our repository: https://anonymous.4open.science/r/Stochastic_Polynomial_Expansion-0BEB.
>
> We used a two-layer MLP architecture and trained vanilla, dropout, ensemble, and DVI+PTPE models using standard training protocols. For the DVI+PTPE model, we applied cyclic annealing during training. We then evaluated model predictions on increasingly rotated versions of MNIST and analyzed the behavior of their predictive distributions. The results show that the DVI+PTPE model is significantly more robust to rotational shifts, exhibiting more stable uncertainty estimates compared to the baseline methods.
>
> For OOD detection, we used Fashion MNIST as the out-of-distribution dataset. Aside from fewer false positives at near-zero entropy for DVI+PTPE, no clear advantage emerged in this setting.

---

> > ### Author Response · Authors · 2025-04-09
> >
> > > Add deep ensembles as a baseline in tables 2 and 3.
> >
> > We have added deep ensemble [2] as a baseline.
> >
> >
> >
> > **RMSE**
> > | **Dataset**              | **Concrete Strength** | **Energy Efficiency** | **Kin8nm**  | **Naval Propulsion** | **Power Plant** | **Protein Structure** | **Wine Quality (Red)** | **Yacht Hydrodynamics** |
> > |--------------------------|-----------------------|-----------------------|-------------|----------------------|----------------|-----------------------|------------------------|-------------------------|
> > | **Data points / Dimensions** | 1030 / 8              | 768 / 8               | 8192 / 8    | 11934 / 16           | 9568 / 4       | 45730 / 9             | 1599 / 11              | 308 / 6                 |
> > | **MCVI**                 | 7.128 ± 0.123         | 2.646 ± 0.081         | 0.099 ± 0.001| 0.005 ± 0.001        | 4.327 ± 0.035  | 4.842 ± 0.031         | 0.646 ± 0.008          | 6.887 ± 0.675           |
> > | **PBP**                  | 5.667 ± 0.093         | 1.804 ± 0.048         | 0.098 ± 0.001| 0.006 ± 0.000        | 4.124 ± 0.035  | 4.732 ± 0.013         | 0.635 ± 0.008          | 1.015 ± 0.054           |
> > | **Dropout**              | 5.23 ± 0.12           | 1.66 ± 0.04           | 0.10 ± 0.00 | 0.01 ± 0.00          | 4.02 ± 0.04    | 4.36 ± 0.01           | **0.62 ± 0.01**        | 1.11 ± 0.09             |
> > | **Ensemble**             | 6.03 ± 0.58           | 2.09 ± 0.29           | 0.09 ± 0.00  | **0.00 ± 0.00**     | 4.11 ± 0.17    | 4.71 ± 0.06           | 0.64 ± 0.04            |  1.58 ± 0.48            |
> > | **PTPE ReLU**            | 5.196 ± 0.206         | 0.615 ± 0.024         | 0.072 ± 0.001| **0.003 ± 0.000**    | 3.925 ± 0.025  | 4.445 ± 0.042         | 0.633 ± 0.010          | 0.640 ± 0.057           |
> > | **PTPE GELU**            | **5.068 ± 0.153**     | **0.570 ± 0.021**     | **0.071 ± 0.000** | 0.004 ± 0.000    | **3.915 ± 0.024**| 4.415 ± 0.043        | 0.634 ± 0.010          | **0.623 ± 0.049**       |
> > | **PTPE Tanh**            | 5.574 ± 0.148         | 0.580 ± 0.022         | 0.076 ± 0.001| 0.005 ± 0.000        | 4.073 ± 0.028  | **4.364 ± 0.036**     | 0.628 ± 0.010          | 1.678 ± 0.193           |
> >
> >
> >
> >
> > **Log Likelihood**
> > | **Dataset**         | **Concrete Strength** | **Energy Efficiency** | **Kin8nm**  | **Naval Propulsion** | **Power Plant** | **Protein Structure** | **Wine Quality (Red)** | **Yacht Hydrodynamics** |
> > |---------------------|----------------------|-----------------------|-------------|----------------------|-----------------|-----------------------|------------------------|-------------------------|
> > | **MCVI**            | -3.391 ± 0.017       | -2.391 ± 0.029        | 0.897 ± 0.010 | 3.734 ± 0.116       | -2.890 ± 0.010   | -2.992 ± 0.006        | -0.980 ± 0.013         | -3.439 ± 0.163          |
> > | **PBP**             | -3.161 ± 0.019       | -2.042 ± 0.019        | 0.896 ± 0.006 | 3.731 ± 0.006       | -2.837 ± 0.009   | -2.973 ± 0.003        | -0.968 ± 0.014         | -1.634 ± 0.016          |
> > | **Dropout**         | -3.04 ± 0.02       | -1.99 ± 0.02          | 0.95 ± 0.01  | 3.80 ± 0.01         | -2.80 ± 0.01     | -2.89 ± 0.01          | -0.93 ± 0.01           | -1.55 ± 0.03            |
> > | **Ensemble**        | -3.06 ± 0.18         | -1.38 ± 0.22          | 1.20 ± 0.02  | 5.63 ± 0.05         | -2.79 ± 0.04     | -2.83 ± 0.02        | -0.94 ± 0.12           | -1.18 ± 0.21            |
> > | **DVI**             | -3.06 ± 0.01         | -1.01 ± 0.06          | 1.13 ± 0.00  | **6.29 ± 0.04**     | -2.80 ± 0.00     | -2.85 ± 0.01          | **-0.90 ± 0.01**       | -0.47 ± 0.03            |
> > | **PTPE ReLU**       | **-3.010 ± 0.037**   | -1.045 ± 0.044        | 1.251 ± 0.009| 5.751 ± 0.086       | -2.789 ± 0.007 | -2.821 ± 0.024        | -0.966 ± 0.029         | -0.910 ± 0.044          |
> > | **PTPE GELU**       | -3.092 ± 0.056       | -0.789 ± 0.039      | 1.278 ± 0.007 | 5.858 ± 0.135    | **-2.780 ± 0.006** | -2.801 ± 0.019    | -0.982 ± 0.029         | **-0.236 ± 0.052**      |
> > | **PTPE Tanh**       | -3.159 ± 0.039       | -0.827 ± 0.043        | 1.234 ± 0.008| 6.050 ± 0.028     | -2.825 ± 0.006   | -2.802 ± 0.013      | -0.939 ± 0.016         | -0.699 ± 0.067        |
> > | **LL Tanh**         | -3.07 ± 0.07         | **-0.65 ± 0.05**      | **1.29 ± 0.01**| **6.29 ± 0.19**    | -2.79 ± 0.01   | **-2.79 ± 0.00**     | -0.98 ± 0.01           | -0.92 ± 0.03            |

---

> > > ### Author Response · Authors · 2025-04-09
> > >
> > > > In particular, provide classification accuracy for the CIFAR-100 experiment, and add some "more realistic" corruptions from Hendrycks and Dietterich (2019).
> > >
> > > We would like to clarify that this experiment involved deterministic networks, not BNNs trained with PTPE or Jacobian linearization. The objective of this experiment was to evaluate PTPE's ability to approximate moments in the multivariate setting, where estimating the covariance structure is nontrivial. This complements Figure 1, which focuses on the univariate case, where covariance and cross-covariance terms are not present.
> > >
> > > If classification accuracy is desired, we note that the networks used in this experiment achieved over 90% accuracy in most cases, with the exception of those using the tanh activation, which reached over 80%. While these results are not competitive with state-of-the-art classifiers, we emphasize that the primary focus of this experiment is moment estimation, not model performance. In the context of BNN training, the accuracy of moment approximation should remain consistent across epochs, regardless of the underlying model’s classification accuracy. That is, a reliable moment propagation method should function independently of model performance.
> > >
> > > > Add an ablation for comparing the number of approximating terms (and MC estimation with different numbers of samples) based on a realistic uncertainty quantification benchmark (rather than the toy Gaussian setting shown in Fig 11.)
> > >
> > > As mentioned earlier, we are currently unable to provide BNN results using PTPE on larger-scale uncertainty quantification benchmarks due to the technical limitations discussed in the main response. However, we agree that this is an important direction, and we hope our work will motivate follow-up studies that apply PTPE to broader settings.
> > >
> > >
> > > [1] Ovadia, Y., Fertig, E., Ren, J., Nado, Z., Sculley, D., Nowozin, S., Dillon, J. V., Lakshminarayanan, B., & Snoek, J. (2019). Can you trust your model's uncertainty? Evaluating predictive uncertainty under dataset shift. Advances in Neural Information Processing Systems, 32.
> > >
> > > [2] Lakshminarayanan, B., Pritzel, A., & Blundell, C. (2017). Simple and Scalable Predictive Uncertainty Estimation using Deep Ensembles. In Proceedings of the 31st Conference on Neural Information Processing Systems (NeurIPS 2017), Long Beach, CA, USA.
> > >
> > > [3] Fu, H., Li, C., Liu, X., Gao, J., Celikyilmaz, A., & Carin, L. (2019). Cyclical annealing schedule: A simple approach to mitigating KL vanishing. In Proceedings of the 2019 Conference of the North American Chapter of the Association for Computational Linguistics (NAACL).

---

### Review · Reviewer_LDk4 · 2025-04-17

**Summary Of Contributions:**

This work proposes a method for propagating Gaussian inputs through a non-linearity $f$ by local polynomial approximation. The authors focus on the setting where each co-ordinate of $f$ depends only on the corresponding co-ordinate of $X$, as in the non-linearity in a neural network, so that the derivatives of $f$ are all diagonal. They then use a truncated Taylor series approximation, at each step using the expectation of the derivative, and due to the diagonality, the cost is only $𝒪︀(n^2)$ for length $n$ inputs.

They provide approximations to permit calculation of the expected derivatives of various widely used non-linearities, and explicit expressions for the expansion.

Experimentally, they show that the third order expansion performs empirically much better than existing first order expansions (local linearisations), in the sense that the propagated distribution is much closer to the true distribution, at least when the input variance is fairly large.

**Audience:**

Yes

**Claims And Evidence:**

Yes

**Requested Changes:**

Some minor points:

* Citations in sentences (as opposed to in parenthesis) sometimes have years and sometime don't. I think they should always have years.
* Near end of section 2: 'v.s.' -> 'versus', 'vs', or 'against'
* Figure 3, etc: It's fairly easy to guess what the legend is supposed to say, but it's both too small and overlapping with the lines - it would be good to tidy this up. You could put the legend at the bottom (in 1-2 rows; probably easiest), or elsewhere outside of the subplots.

**Strengths And Weaknesses:**

Generally, the weaknesses from the previous submission are well addressed, and the paper is overall strong.

Strengths:

* (major) The empirical evaluation is quite thorough, and the authors are generally quite clear about the contributions
* (major) The idea is clearly communicated, and there are helpful comparisons with similar methods

Weaknesses:
* (minor) It remains the case that the experiments look at multiple different orders of approximation which are sometimes quite similar - in Figures 3, 8, 9. It's not clear if the differences here are random (error bars?) or not - in particular, sometimes higher order approximations perform worse. Is that predictable? The final figures in the appendix help the reader to get a bit more insight, but it would be better if the text drew these together and gave some recommendations for the reader.

---

> ### Author Response · Authors · 2025-04-21
>
> We thank the reviewers for their time and constructive feedback. We have carefully considered each comment and provide our detailed responses below.
>
> > It remains the case that the experiments look at multiple different orders of approximation which are sometimes quite similar - in Figures 3, 8, 9. It's not clear if the differences here are random (error bars?) or not.
>
> We agree that the approximations appear similar, particularly when viewed on a logarithmic scale. This is partly because batch normalization tends to reduce the variance of activations across layers, thereby limiting the contribution of higher-order correction terms. To aid in interpretation, we drew connecting lines between scatter points, which help visually differentiate the methods by indicating which performs better.
>
> To assess variability, we reran the experiments 10 times and computed error bars. However, the resulting variances were very small—too small to be meaningfully visible on the current scale, so the answer is: no, the differences here are not random. For transparency and reproducibility, we have made the underlying data and code used to generate these figures publicly available in the anonymous repository https://anonymous.4open.science/r/Stochastic_Polynomial_Expansion-0BEB.
>
> A side note: In the course of revisiting our experiments, we identified that using single precision introduced non-negligible errors when calculating the sample mean and covariance for the Monte Carlo (MC) estimates. To address this, we recomputed the mean and covariance using double precision and replotted Figures 3, 8, 9, and 11 accordingly. While the overall conclusions remain unchanged, we observed that the measured values of evaluation metrics (Wasserstein-2, L2, and Frobenius distances) dropped by approximately an order of magnitude. The updated figures are available in the repository under cifar10experiment/figures/.
>
> > In particular, sometimes higher order approximations perform worse. Is that predictable?
>
> Thank you for raising this important point. While higher-order approximations generally perform better in theory, they can sometimes underperform in practice due to the following factors:
>
> 1. Approximate Gaussianity: The method relies on the assumption that the transformed distributions are approximately Gaussian. However, this assumption is only an approximation, and deviations from Gaussianity can cause higher-order terms to be less effective or even detrimental in certain cases.
>
> 2. Numerical Instability at Low Variance: When the input variance is small (e.g., due to batch normalization), higher-order terms become more sensitive to numerical instability. This instability can introduce errors that outweigh the potential benefits of higher-order corrections.
>
> 3. Imprecise Reference Distributions: The evaluation of our method relies on comparing to a reference distribution estimated using 1e6 samples. However, this is actually a very limited sample size for estimating the statistics (e.g., mean and covariance) of a high-dimensional distribution (e.g., 10D or more). Ideally, we would require a number of samples on the order of (1e6)^10 to obtain accurate estimates—but this is computationally infeasible. Even increasing the sample size to 1e7 already requires substantial runtime and memory, and moving beyond that (e.g., to 1e8) becomes prohibitively expensive. That said, we found that increasing the sample size from 1e6 to 1e7 leads to reduction in metrics Wasserstein-2, L2, and Frobenius distances. This suggests that the observed underperformance of higher-order terms may be partly due to imperfections in the estimated reference.
>
> While the first and third issues are inherent to the problem setup and cannot be fully eliminated, the second issue can be addressed. In practice, we recommend adaptively choosing the order of approximation based on input variance. For example, use second-order PTPE when the variance is low to avoid instability, and use higher orders when the input variance is larger.

---

> > ### Author Response · Authors · 2025-04-21
> >
> > > The final figures in the appendix help the reader to get a bit more insight, but it would be better if the text drew these together and gave some recommendations for the reader.
> >
> > Thank you for the suggestion. To improve clarity and accessibility, we have moved the explanatory footnote previously on page 7 into the introduction, where it now provides an early overview of how the different figures relate to one another.
> >
> > > Citations in sentences (as opposed to in parenthesis) sometimes have years and sometime don't. I think they should always have years.
> >
> > We appreciate this stylistic observation. In our manuscript, we used \citeauthor{} when referring to authors as the subject of a sentence (which omits the year), and \cite{} when referring to a paper (which includes the year). This is a common stylistic distinction in academic writing, but we are happy to revise this for consistency if preferred by the journal.
> >
> > > Near end of section 2: 'v.s.' -> 'versus', 'vs', or 'against'
> >
> > Thank you for catching this. We have updated the text accordingly.
> >
> > > Figure 3, etc: It's fairly easy to guess what the legend is supposed to say, but it's both too small and overlapping with the lines - it would be good to tidy this up. You could put the legend at the bottom (in 1-2 rows; probably easiest), or elsewhere outside of the subplots.
> >
> > Thank you for the attention to detail. We have adjusted the figure layout and updated the legends to improve clarity and avoid overlap. The revised figures are available in the repository under cifar10experiment/figures/.

---

### Decision · Action_Editor_Syhe · 2025-06-02

**Recommendation:** Accept with minor revision

**Additional Comments:**

This is a resubmission to TMLR and I quote previous reviews from the last round as follows:

Previously, reviewers' major questions for this submission were:
- Clarification on the method regarding related work (e.g., approximations done in Kalman filtering), and not enough literature review.
- Clarity on the math presentations.
- Experiments on real-world datasets are missing -- only with experiments on neural networks showing approximation error to Monte Carlo estimates.

I tried to recruit the same set of reviewers with part success. This round, while reviewers felt that the major concerns from previous submission are mostly addressed, the remaining major comments from the reviewers are:
- Experiment on real-world datasets are still relatively small scale (as compared with large-scale deep learning).
- Theoretical analyses regarding the approximation quality for the activation functions are missing.

In revision please address the reviewers' comments and include in the additional experiments provided in author feedback phase.

**Audience:**

Yes

**Audience Explanation:**

This paper is for machine learning researchers interested in uncertainty quantification. The audience size should be big enough.

**Claims And Evidence:**

Yes

**Claims Explanation:**

This work considers uncertainty quantification in neural network by propagating Gaussian distributions through the network. The main contribution is the truncated polynomial approximations to a list of commonly used activation functions to enable accurate approximation of the mean and variance of transformed Gaussian. Experiments have validated the efficacy of the approximations on simulated benchmarks as well as small scale real world datasets such as UCI regression benchmark and CIFAR-10.

The paper's strength lies in the comprehensive efforts in providing polynomial approximations to many commonly used activation functions, which are validated with extensive synthetic datasets.